# CreativityPrism: A Cross-Domain Evaluation Framework for Large Language Model Creativity

**Zhaoyi Joey Hou**[1†]**, Bowei Alvin Zhang**[2]**, Yining Lu**[3]**, Bhiman Kumar Baghel**[1]**,
Anneliese Brei**[4]**, Ximing Lu**[5]**, Meng Jiang**[3]**, Faeze Brahman**[6]**, Snigdha Chaturvedi**[4]**,
Haw-Shiuan Chang**[7]**, Daniel Khashabi**[2]**, Xiang Lorraine Li**[1†]

[1]*University of Pittsburgh,* [2]*Johns Hopkins University,* [3]*University of Notre Dame,*
[4]*University of North Carolina at Chapel Hill,* [5]*University of Washington,*
[6]*Allen Institute for Artificial Intelligence,* [7]*University of Massachusetts Amherst,*
[†]*Correspondence to joey.hou@pitt.edu, xianglli@pitt.edu*

**Reviewed on OpenReview:** *https://openreview.net/forum?id=3pfsQcEtNC*

## Abstract

Creativity is often seen as a hallmark of human intelligence. While large language models(LLMs) are increasingly perceived as generating creative text, there is still no *cross-domain* and *scalable* framework to evaluate their creativity across diverse scenarios. Existing methods of LLM creativity evaluation either heavily rely on humans, limiting speed and scalability, or are fragmented across different domains and different definitions of creativity. To address this gap, we propose CREATIVITYPRISM[1], an evaluation and analysis framework that consolidates eight tasks from three domains: divergent thinking, creative writing, and logical reasoning, into a taxonomy of creativity that emphasizes three dimensions: quality, novelty, and diversity of LLM generations. The framework is designed to be scalable with reliable automatic evaluation judges that have been validated against human annotations. We evaluate 17 state-of-the-art (SoTA) LLMs on CREATIVITYPRISM and find that while frontier-scale LLMs dominate creative writing and logical reasoning tasks by a .10 (or 15%) lead over locally-deployable open models, they offer no significant advantage in divergent thinking, a domain much less explored in existing post-training regimes. Our analysis also shows that high performance in one creative dimension or domain rarely generalizes to others; specifically, novelty metrics often show weak or negative correlations with other metrics. This fragmentation confirms that a cross-domain, multi-dimensional framework like CREATIVITYPRISM is essential for any meaningful assessment of LLM creativity.

## 1 Introduction

Creativity, the capacity to generate novel and valuable ideas or solutions (Holyoak & Morrison, 2005; Boden, 1994; Finke et al., 1992), is a core human cognitive ability. It appears in many domains:

---

[1]Code and data: `https://joeyhou.github.io/CreativityPrism/`

crafting stories with surprising plot twists (Ismayilzada et al., 2024b; Atmakuru et al., 2024), producing groundbreaking scientific discoveries (Hu & Adey, 2002; Si et al., 2025), solving problems under constraints (Lu et al., 2025b; Ye et al., 2025), or even expressing humor in everyday life (He et al., 2019; Zhong et al., 2024). Its multifaceted nature has prompted extensive study in psychology and cognitive science, with efforts to capture creativity through both qualitative and quantitative approaches (Guilford et al., 2012; Olson et al., 2021; Alabbasi et al., 2022; Sternberg & Lubart, 1991).

Recently, with the rapid rise of general-purpose large language models(LLMs), interest has grown in probing their creativity (Zhao et al., 2025; Goes et al., 2023; Chakrabarty et al., 2024a; Lu et al., 2025b; Atmakuru et al., 2024). But as with human creativity, machine creativity spans diverse and expansive contexts, making it difficult to define, formalize, and, above all, measure. Concretely, LLM creativity evaluation faces two challenges: **distinct definitions of creativity** across different domains and difficulty of **scalable, automatic evaluation** due to the convoluted nature of creativity.

The first challenge stems from current research in machine creativity being scattered across different domains and focusing on narrow or singular dimensions. For example, the Divergent Association Task(DAT) (Chen & Ding, 2023a; Bellemare-Pepin et al., 2026) and the Creative Short Story Task (Ismayilzada et al., 2024b) emphasize lexical diversity; the Alternative Uses Test(AUT) (Goes et al., 2023; Organisciak et al., 2023) solely focuses on unconventional ideas of using daily items, overlooking the pragmatics of those solutions; CreativeMath (Ye et al., 2025) and NeoCoder (Lu et al., 2025b) only study math and coding problems, correspondingly. What makes comparison even harder is that these task-specific and domain-specific benchmarks only benchmark their own choices of LLMs, which vary from one another. Without a cross-domain evaluation that incorporates the evaluation of creativity from all those dimensions and covers a wide range of LLMs, it is hard to uncover a full picture of how well current state-of-the-art(SoTA) LLMs are doing when it comes to creativity.

The second challenge arises from the subjective nature of creativity, which makes it hard to automatically evaluate LLM output and leads many existing benchmarks to rely heavily on human evaluation (Tian et al., 2024c; Chakrabarty et al., 2024a). While human judgment is often considered the gold standard for nuanced tasks, it presents significant hurdles for modern AI research: it is prohibitively expensive, difficult to replicate at scale, and requires a long turnaround time that cannot keep pace with the current field. With new LLMs and model iterations being released nearly every week, a reliance on manual grading creates a massive bottleneck that prevents the rapid, iterative testing required for progress.

To this end, we propose CREATIVITYPRISM, a cross-domain and scalable evaluation framework of LLM creativity; it is cross-domain — consisting of eight tasks from three domains: divergent thinking, creative writing, and logical reasoning (i.e., mathematical reasoning and coding); it is also scalable — all evaluation metrics are automatic, ensuring easy benchmarking for any LLM. First, cross-domain: we extend beyond a simple combination of existing benchmarks from various domains by systematically categorizing existing task-specific metrics along the three dimensions, i.e., quality, novelty, and diversity, to facilitate a full-scale measurement of model creativity (Figure 1). Every metric in CREATIVITYPRISM belongs to one of these three dimensions, and hence, evaluation results can be summarized into those three dimensions, providing a cross-domain and dimension-specific insight into LLMs' creativity performance. Second, scalability: we go beyond simple automatic

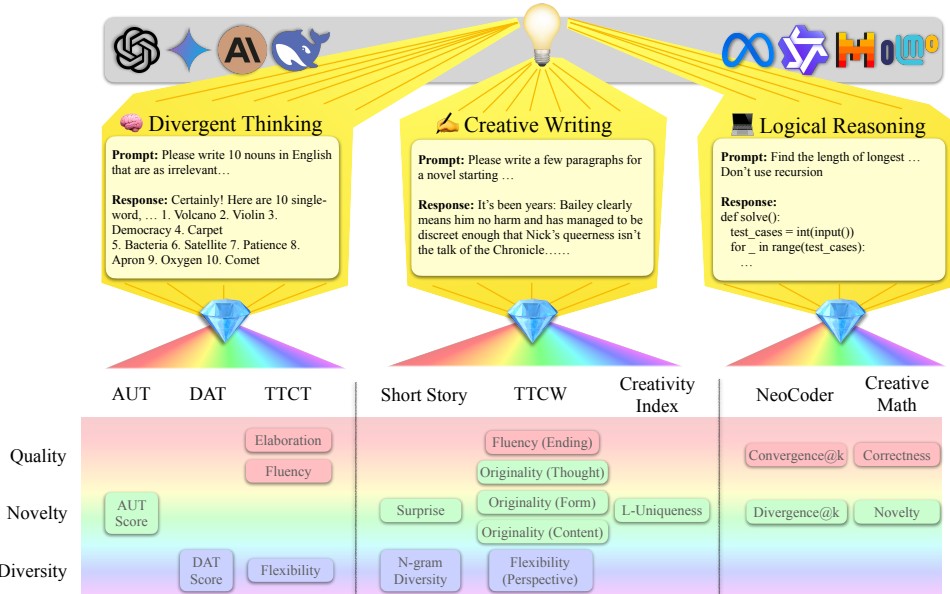

Figure 1: Overview of CREATIVITYPRISM. Each LLM is prompted to complete the tasks (example inputs in Table 1), and their outputs are evaluated using task-specific metrics. We also organize metrics into three dimensions of creativity: **quality**, **novelty**, and **diversity**.

evaluation with the LLM-as-a-Judge paradigm (§4.2). For every evaluation metric that requires LLM-as-a-Judge, we collect human judgments from well-trained researchers or domain experts, followed by Alternative Annotator Test (Calderon et al., 2025) to verify our LLM-Judge setup has a much closer alignment to high-quality human judgment among tasks in CREATIVITYPRISM than previous automatic evaluations. In this way, we ensure the metrics in CREATIVITYPRISM are well aligned with high-quality human judgment.

We then evaluate 17 SoTA LLMs on CREATIVITYPRISM, to answer the following research questions. **RQ1: Where is the performance gap across LLMs on creative tasks?** We found a notable performance gap between frontier-scale and locally-deployable open models, especially in logical reasoning tasks, followed by creative writing tasks. We also found the same performance gap to be most pronounced in the quality dimension. **RQ2: What are the correlations among LLMs' performance across different dimensions and domains?** In other words, would good performance in one creativity domain or dimension transfer to the other? Results have shown that the models perform similarly in metrics from the same task or the same domain; for metrics from different domains, models perform similarly in diversity and quality dimensions, while performances in novelty dimensions are much less correlated. We believe this stems from the inherent difference in how novelty is defined in different tasks and domains.

In short, our contribution can be summarized as follows:

- We select, combine, and refactor the codebase of eight creativity tasks across three domains to propose an easy-to-use, cross-domain and scalable evaluation framework, CREATIVITYPRISM;

- We propose the "quality, novelty, and diversity" taxonomy of creativity and categorize seventeen metrics into these dimensions for dimension-level LLM creativity evaluation;

- We conduct high-quality human annotation to verify human agreement for each metric and human-LLM-Judge agreement during the automatic evaluation process;

- We benchmark 17 SoTA LLMs on CREATIVITYPRISM and conduct extensive analysis on model performance across domains and dimensions.

## 2  Related Work

**Human Creativity**   The definition of creativity has varied across different domains. In psychology, Torrance Tests of Creative Thinking (TTCT) (Alabbasi et al., 2022) considers creativity as a combination of originality, flexibility, fluency, and elaboration; Runco & Jaeger (2012) proposes a simpler taxonomy of originality and usefulness. In marketing, El-Murad & West (2004); Rosengren et al. (2020) considers advertisement creativity as the combination of usefulness and originality; additionally, Smith et al. (2007) adds flexibility, fluency, elaboration, synthesis, and artistic values. In terms of creativity evaluation, Said-Metwaly et al. (2017) summarizes more than 100 existing works and concludes that creativity evaluation is an "unsettled" issue, with one of the key reasons being the lack of holistic, cross-domain evaluation, which also motivates our work. A common belief among these work is the balanced view of quality (e.g., elaboration, usefulness), novelty (e.g., originality, synthesis), and diversity (e.g., flexibility) when it comes to different dimensions of creativity. This view directly inspires our taxonomy of creativity metrics in §3.

**Machine Creativity**   Measurement of machine creativity has become increasingly popular with the rapid development of LLMs. Many researchers attempt to provide a cross-domain or cross-dimension view of machine creativity in which a variety of angles have been taken, such as panoramic survey (Ismayilzada et al., 2024a), computational models involved in machine creativity (Franceschelli & Musolesi, 2024), the efficacy of evaluation methods and metrics (Lu et al., 2025a; He et al., 2025), the effect of LLM decoding strategy on output creativity (Nagarajan et al., 2025), the tension between factuality and creativity (Banerjee et al., 2025), and multi-modal creativity (Fang et al., 2025; Xue et al., 2025). However, none of them attempt to validate, synthesize, and combine the evaluation metrics from a series of creativity tasks and benchmark existing SoTA LLMs as we do.

The community has also explored a wide range of domain-specific problems when it comes to LLM creativity, including logical-based problem-solving (Ye et al., 2025; Lu et al., 2025b; Chen et al., 2025), physical and commonsense reasoning (Tian et al., 2024b), creative writing (Gómez-Rodríguez & Williams, 2023; Lu et al., 2024b; Chakrabarty et al., 2024a; Ismayilzada et al., 2024b; Tian et al., 2024a; Atmakuru et al., 2024; Qiu & Hu, 2025), scientific discovery (Si et al., 2025; Kumar et al., 2025; Afzal et al., 2025), response diversity in question answering (Zhang et al., 2025; McLaughlin et al., 2024), and human-AI collaborative creative problem solving (Chakrabarty et al., 2024b; Boussioux et al., 2024; N. Lane et al., 2024). All these studies focus on LLM evaluation in a specific domain, each with its own evaluation philosophy and metrics, whereas our work aims at providing a cross-domain and cross-dimension evaluation of the LLM's output for tasks in a variety of domains.

**Automatic Creativity Evaluation**   Evaluating the creativity of machine-generated text has been a challenging task, and much of the work relies on human evaluation. However, due to the cost, human evaluation is challenging to scale and requires a lengthy wait time. To achieve scalable evaluation, researchers adopt two broad groups of evaluation methods: feature-based and generative-based. The former includes psycholinguistic features, such as arousal, valence score (Mohammad, 2018), lexical features, such as lexical diversity (Padmakumar & He, 2023), and text embedding

distances (Pennington et al., 2014b; Zhang* et al., 2020). The latter is mainly LLM-as-a-judge (Tan et al., 2025; Li et al., 2024a;b). Recent work has shown promising potential of this method (Zheng et al., 2023) as well as quantitative ways to measure how well LLM-Judge aligns with human judgment (Calderon et al., 2025; Han et al., 2025). Our evaluation framework integrates both feature-based and generative evaluators. For metrics that rely on LLM judges, we systematically validate their output against human annotations using the Alternative Annotator Test (see §4).

| Task | Example |
|---|---|
| 🧠 **Alternative Uses Test (AUT)** (Goes et al., 2023) | Create a list of creative alternative uses for a bottle. |
| 🧠 **Divergent Association Task (DAT)** (Chen & Ding, 2023a; Bellemare-Pepin et al., 2026) | Please write 10 nouns in English that are as irrelevant from each other as possible, in all meanings and uses of the words.[2] |
| 🧠 **Torrance Tests of Creative Thinking (TTCT)** (Zhao et al., 2025) | What might be the consequences if humans suddenly lost the ability to sleep? |
| ✍️ **Torrance Test of Creative Writing (TTCW)** (Chakrabarty et al., 2024a) | Write a New Yorker-style story given the plot below. Make sure it is at least 2000 words. Plot: A woman experiences a disorienting night in a maternity ward...; Story: |
| ✍️ **Creative Short Story** (Ismayilzada et al., 2024b) | Come up with a novel and unique story that uses the required words in unconventional ways or settings. Use at most five sentences. The given words: petrol, diesel, and pump. |
| ✍️ **Creativity Index** (Lu et al., 2024b) | Please write a few paragraphs for a novel starting with the following prompt: "It's been years: Bailey clearly means him no harm and has managed to..." |
| 💻 **NeoCoder** (Lu et al., 2025b) | You are given a sequence of integers $a$ of length $2n$. You have to split them into $n$ pairs. Don't use HashMap, while loop. |
| 💻 **Creative Math** (Ye et al., 2025) | Question: What is the largest power of 2 that is a divisor of $13^4 - 11^4$? A.8 B.16 C.32 D.64 E.128; Reference Solutions 1: ... ; Reference Solutions 2: ... |

Table 1: Tasks in CREATIVITYPRISM with examples. 🧠: divergent thinking, ✍️: creative writing, 💻: logical reasoning. More details and examples can be found in Appendix F.

## 3 CreativityPrism

**Task Selection** In designing CREATIVITYPRISM, we make sure it is both cross-domain and scalable. These two requirements are reflected in task selection: we examine all creativity-related tasks with publicly available data and executable, well-documented codebases; we also further ensure the reliability of tasks that require LLM-Judge by conducting additional human annotation with trained researchers and LLM-Judge quality test to filter out tasks where no well-aligned LLM-Judge is available (§4.2). Our task selection process leads to eight tasks and seventeen metrics, across three domains: 🧠 divergent thinking, ✍️ creative writing, and 💻 logical reasoning (Figure 1). The divergent thinking domain consists of established psychology tasks, which were originally designed to assess human ability in coming up with diverse and alternative answers to given questions (Goes

---

[2]Different versions of DAT prompts exist across psychology and computer science research; the exact prompt is in Appendix F.6.

et al., 2023; Chen & Ding, 2023a; Bellemare-Pepin et al., 2026; Zhao et al., 2025). The creative writing domain includes tasks that require models to produce short written pieces (Chakrabarty et al., 2024a; Ismayilzada et al., 2024b; Lu et al., 2024b). The logical reasoning domain includes coding (Lu et al., 2025b) and math task (Ye et al., 2025) to evaluate models' ability to generate creative solutions under strict, explicit reasoning constraints. Task format examples are in Table 1.

**Three Dimensions of Creativity**  We go beyond a simple combination of existing tasks by categorizing all metrics into three dimensions of creativity: *quality*, *novelty*, and *diversity*. Our taxonomy is grounded in the most recent version of the TTCT verbal test, which operationalizes creativity along three dimensions: fluency, flexibility, and originality.[3] These map directly onto our dimensions: fluency, the ability to produce coherent, well-formed responses, corresponds to *quality*; flexibility, the ability to vary approaches and perspectives, corresponds to *diversity*; and originality, the ability to produce responses that deviate from the commonplace, corresponds to *novelty*(Torrance). We are also different from the widely-cited binary taxonomy of usefulness and originality (Runco & Jaeger, 2012) by separating originality into novelty and diversity. We argue this distinction is particularly important for LLM evaluation: diversity captures breadth, i.e., how much a model varies its outputs across responses, while novelty captures depth, i.e., how much a single output deviates from existing or conventional solutions (Figure 2). A model can score highly on one while failing on the other, e.g., producing many distinct but individually unoriginal responses (high diversity), or producing one highly novel response with no variation across prompts (high novelty). Combining diversity and novelty into one originality score would obscure such differences.

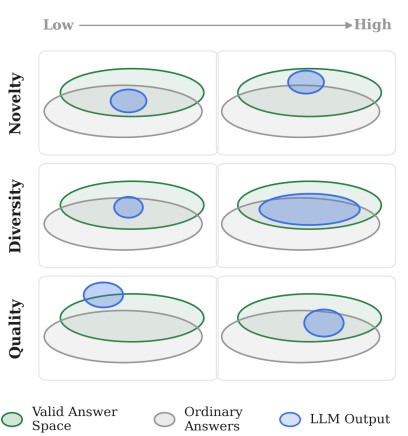

Figure 2: Visual representation of three dimensions in CREATIVITYPRISM.

Besides theoretical groundings, we also refer to a visualized framing that motivates our choice of dimensions (Figure 2). Consider three spaces: LLM's output distribution (**blue**), valid answer space (**green**), and ordinary answer space (**grey**). Each dimension captures a distinct geometric relationship among these three spaces.

*Quality* measures how much of the LLM's output distribution falls within the valid answer space, i.e., how well the generated content fulfills the task's functionality. For example, in NeoCoder, the quality of generated code is measured by the success of execution and coding task completion; in TTCT, quality is measured by the amount of detail provided. We believe quality is critical for creativity because, without quality, random words and sentences would be very creative in terms of novelty and diversity, but do not convey any meaning.

*Novelty* measures how far the LLM's output distribution extends beyond the ordinary answers, i.e., how rare the generated content is compared to existing or commonly seen content. For example, in NeoCoder and Creative Math, novelty involves coming up with solutions that are different from the reference solutions; in AUT, novelty involves different use of the tool compared to ordinary uses; in Creativity

---

[3]While the original TTCT (Torrance, 1974) includes a fourth dimension, elaboration, the current version of the TTCT verbal test, as administered by major testing agencies (Torrance; ttc) consists of three dimensions only, a simplification also reflected in recent psychology literature (Alabbasi et al., 2022).

Index, novelty is measured by the normalized n-gram overlaps between model-generated text and the traceable part of the training corpus.

***Diversity*** measures the breadth of the LLM's output distribution within the valid answer space, i.e., how widely the model explores different valid responses across generations. For example, in TTCT, diversity is reflected in the flexibility score - the ability to produce varied solutions to given questions; for the DAT task, diversity involves the semantic difference among the LLM-generated nouns; similarly in Li et al. (2025), diversity is the number of semantically similar clusters an LLM generates in response to a given prompt.

These three relationships are geometrically independent: a model can expand its output distribution along one axis without moving along the others. This independence is what motivates treating them as separate dimensions rather than collapsing them.

**Different Domain Tasks within Same Dimension** As shown in Figure 1, there are tasks from different domains within the same dimension, e.g., "N-gram Diversity" from Creative Writing and "DAT Score" from Divergent Thinking are both in the *Diversity* dimension. We make such choice because we believe *Diversity* has different definition across different domains and so do other dimensions. Having heterogeneous metrics within one dimension is complementary to each other, making our evaluation framework cover a wider range of perspectives on creativity.

## 4 Experiments

To holistically evaluate machine creativity, we evaluated 17 SoTA LLMs across eight tasks, reporting both task-specific metrics and an aggregated creativity score using the three-dimensional framework. In this section, we introduce the inference setups, where LLMs are prompted to generate creative responses according to corresponding task requirements (§4.1); then we describe the evaluation process, including score aggregation and how we use LLM-as-a-Judge for evaluation (§4.2).

### 4.1 Inference

All tasks in CREATIVITYPRISM undergo data processing according to the original task setup before running inference. We include 17 LLMs in total, including frontier-scale LLMs [4] from GPT (OpenAI, 2024), Claude (Anthropic, 2024), Gemini (Google DeepMind, 2024; Kavukcuoglu, 2025), and DeepSeek family (DeepSeek-AI et al., 2024; 2025), and locally-deployable open models (referred to as "open models") from Mistral (Jiang et al., 2024; 2023), Qwen (Qwen et al., 2024; Hui et al., 2024), OLMo (Groeneveld et al., 2024), and Llama (Grattafiori et al., 2024) family. For frontier-scale models, we use API access from the corresponding company; for open models, we use `vLLM(v0.7.2)` (Kwon et al., 2023) to run all experiments. We set the temperature to 0.75 and $max\_tokens$ to 4096 for all tasks, unless the original task requires a different setting.[5] More inference configurations are in Appendix F.

---

[4]We use frontier-scale to refer to LLMs that are in practice only accessible via hosted API and not locally fine-tunable on research hardware. This includes both closed-weight LLMs (GPT, Claude, Gemini) and large open-weight LLMs from DeepSeek (DeepSeek-V3/R1); we group them together because they are comparable in practical accessibility despite being different in license. We contrast these with locally-deployable open models (<80B).

[5]Tasks with different decoding parameters: CreativityIndex ($temperature = 1.0, max\_tokens = 288$), Creative Math ($temperature = 0.0, max\_tokens = 2000$). For all tasks, outputs longer than pre-defined $max\_tokens$ will be truncated. Analysis on output length and the effect of different temperatures are in Appendix E.4.

## 4.2 Evaluation

**Aggregated Creativity Scoring**   To capture overall performance within each dimension, we aggregate all metrics in that dimension to produce quality, novelty, and diversity scores. The score aggregation calculation involves two steps: first, every evaluation metric is min-max normalized to a 0–1 scale based on the minimum and maximum possible scores for the task. Second, the normalized metrics are aggregated into quality, novelty, and diversity scores for each LLM (as categorized in Figure 1) by averaging all metrics within each dimension. To ensure equal weighting across tasks, we first average the normalized scores within each task before computing dimension scores. This prevents tasks with multiple metrics in one dimension (e.g., TTCW has three novelty metrics) from disproportionately influencing the final score. We also provide an "overall" creativity score (shown in Table 3) by averaging across the three dimensions to facilitate holistic model comparison. Note that this "overall" score is only to facilitate model comparison. The relative importance of quality, novelty, and diversity varies depending on the application. We suggest future researchers choose from these three separate scores based on their specific research goals. More details about score aggregation can be found in Appendix C.

| Task | Fleiss Kappa | Judge-LLM | Judge Quality |
|---|---|---|---|
| AUT | 0.650 | GPT4.1 | ✓ pass |
| NeoCoder | 0.471 | GPT4.1 | ✓ pass |
| TTCW (Originality - Theme & Content) | 0.660 | Qwen2.5-72B | ✓ pass |
| TTCW (Originality - Thought) | 0.400 | Qwen2.5-72B | ✓ pass |
| TTCW (Originality - Form) | 0.410 | Qwen2.5-72B | ✓ pass |
| TTCW (Flexibility - Perspective & Voice) | 0.440 | Qwen2.5-72B | ✓ pass |
| TTCW (Fluency - Narrative Ending) | 0.480 | Qwen2.5-72B | ✓ pass |
| TTCT (Fluency) | 0.432 | GPT4.1, Qwen2.5-72B | ✓ pass |
| TTCT (Flexibility) | 0.423 | GPT4.1, Qwen2.5-72B | ✓ pass |
| TTCT (Elaboration) | 0.445 | GPT4.1, Qwen2.5-72B | ✓ pass |
| CreativeMath (Novelty) | 0.450 | Claude3-Sonnet, GPT4.1, Gemini2.0-Flash | ✓ pass |
| CreativeMath (Correctness) | - | Claude3-Sonnet | *accuracy: 0.91* |

Table 2: Inter-annotator agreement and Judge-LLM quality. "✓ pass" refers to passing the Alternative Annotator Test; "accuracy" is the accuracy compared to ground truth.

**LLM-as-a-Judge Reliability**   In CREATIVITYPRISM, the evaluation of five tasks (AUT, TTCW, CreativeMath, TTCT, and NeoCoder) involves using LLM as part of the automatic evaluation procedure. To ensure the reliability of the LLM judges in those tasks, we verify two requirements for every metric in these tasks. First, we collect human annotations and compute inter-annotator agreement, confirming that the task is well-defined and that annotators can reach reasonable agreement on the evaluation outcomes.[6] Second, we follow Calderon et al. (2025) and test if our LLM judges' setups can align well with human annotators.

For human inter-annotator agreement, we randomly sample a small subset of data points from the output of inference models and have annotators rate the inference output based on the task rubrics. In total, 10 annotators participated in the LLM-Judge verification, all of whom are Ph.D. students or faculty in the field of computer science in U.S. institutions. Annotator training details can be found

---

[6]TTCW has its own expert annotation, so we do not collect annotations for it. Details in Appendix D.

in Appendix D. Three annotators annotate each data point, and the inter-annotator agreement is measured by Fleiss Kappa (Fleiss, 1971), or quadratic-weighted Fleiss Kappa, if the labels are on a Likert scale. Given the subjective nature of creativity-related tasks, we keep tasks where the Fleiss Kappa measurements among annotators are greater or equal to 0.4, which implies moderate agreement. See detailed agreements in Table 2. Note that the Judge quality of CreativeMath (Correctness) is evaluated by accuracy (compared to human judgment) because the correctness of a solution to a math problem is objective. We simply have annotators verify the correctness of each solution for the sampled questions.

After verifying human agreement, we validate our choice of backbone LLM (referred to as "Judge-LLM"). For the objective task, which only includes Correctness in CreativeMath, we simply calculate the accuracy of our Judge-LLM compared to human judgments. For the subjective tasks, we employ the Alternative Annotator Test (Calderon et al., 2025) to statistically justify alignment with human annotators. This method adopts a leave-one-out strategy to validate the LLM as a substitute with an acceptable error margin[7]: for each human annotator, we test whether the Judge-LLM aligns closer to the remaining human consensus than the excluded human does (subject to a quality margin $\epsilon$). If the LLM aligns better in more than half the cases (winning rate $> 0.5$), we consider it a viable replacement.

Note that the choice of backbone LLM clearly has an impact on whether or not the Judge-LLM passes this test. We adopt the following principle to choose Judge-LLM: we use Qwen2.5-72B (Qwen et al., 2024) as the default backbone LLM due to its open-source nature; if it does not pass the replacement test, we use GPT-4.1 (OpenAI, 2024); if GPT-4.1 does not pass the test, we first try taking the average of GPT-4.1 and Qwen2.5-72B; lastly, we will use the Judge-LLM setup proposed by the original paper that introduced the task. All the Judge-LLM we included in CREATIVITYPRISM pass the Alternative Annotator Test when compared to human annotations. [8]

More details about the human annotation process, Alternative Annotator Test, and discussion on LLM-as-a-Judge failure modes and limitations are in Appendix D; evaluation metrics and evaluation prompts for each task are in Appendix F.

## 5 Results & Analysis

Table 3 summarizes model performances across domains and three creativity dimensions (quality, novelty, and diversity), where the overall score, averaged across these dimensions, serves as a proxy for a model's overall creative capability.[9] As we can see from the table, Qwen2.5-72B and DeepSeek-V3 are the best-performing models among open models and frontier-scale models. For open models, we can see that the model performances improve as the model size increases.

We have also found a performance improvement along the time axis (Figure 3a) where models released in the past two years have become increasingly competitive. Since many of our metrics (e.g., L-Uniqueness in Creativity Index, Divergence@0 in NeoCoder) would reward models that can

---

[7]Alternative Annotator Test (Calderon et al., 2025) involves a statistical procedure with a pre-specified error margin at significance level $\alpha = .05$

[8]We exclude sub-tasks and metrics that fail to meet our reliability requirements (Fleiss Kappa $\geq 0.4$ and passing the Alternative Annotator Test), though we minimize such modifications whenever possible (details in Appendix D).

[9]Results reported in the main text portion are from one single run; we reported a case study on performance stability in AUT and TTCW task across 5 different runs with the same configuration in Appendix E.4.

| Model | Overall | Quality | Novelty | Diversity | Creative Writing | Divergent Thinking | Logical Reasoning |
|---|---|---|---|---|---|---|---|
| **<10B** | | | | | | | |
| Mistral-7B | .423 | .268 | .393 | .607 | .316 | .679 | .320 |
| Qwen2.5-7B | .406 | .374 | **.398** | .445 | .207 | .654 | **.460** |
| OLMo2-7B | **.462** | **.405** | .340 | **.640** | **.403** | .643 | .257 |
| Llama3.1-8B | .404 | .313 | .391 | .509 | .239 | **.683** | .409 |
| **10-40B** | | | | | | | |
| OLMo2-13B | .451 | .389 | .379 | .586 | **.424** | **.672** | .278 |
| Mistral-24B | .448 | .352 | **.440** | .553 | .346 | .614 | **.473** |
| Qwen2.5-32B | **.453** | **.417** | .336 | **.605** | .338 | .655 | .358 |
| **40-80B** | | | | | | | |
| Mixtral-8x7B | .430 | .318 | .392 | .578 | .278 | .687 | .420 |
| Llama3.3-70B | .446 | .401 | .404 | .534 | .269 | .614 | .529 |
| Qwen2.5-72B | **.549** | **.526** | **.490** | **.632** | **.385** | **.736** | **.554** |
| **Frontier-scale** | | | | | | | |
| Claude3-Sonnet | .632 | .606 | .557 | .733 | .507 | .765 | .612 |
| Claude3-Haiku | .535 | .522 | .456 | .627 | .413 | .685 | .568 |
| GPT4.1 | .641 | .618 | .582 | .732 | .518 | .722 | .682 |
| GPT4.1-mini | .615 | .567 | .571 | .718 | .504 | .697 | .649 |
| Gemini2.0-Flash | .626 | .657 | .522 | .712 | .438 | .753 | .655 |
| DeepSeek-R1 | .665 | .677 | .575 | **.742** | .548 | .724 | .643 |
| DeepSeek-V3 | **.695** | **.749** | **.616** | .728 | **.571** | **.768** | **.726** |

Table 3: Model performance on CREATIVITYPRISM, grouped by model size. Frontier-scale models are grouped together. All scores are between 0 and 1, and the higher the better. Overall is the average of Quality, Novelty, and Diversity scores. The rightmost three columns are the average scores across tasks in each domain. **Bold** are the best results in the corresponding model group. Refer to Figures 3, 11, 12, 13, 14, 15, 16 for a visualized performance comparison.

generate content different from prior content, having the chance of learning the latest content from the corpus with later cutoff dates would intuitively make models more competitive. More details on model release time details can be found in Appendix B.

## 5.1 Gap Between Frontier-scale Models and Open Models

**Overall Performance Gap** As shown in Table 3, the best frontier-scale model(s) outperform the best open model(s) by more than **.10 (or 15%) in each dimension** of creativity. This shows a big gap between frontier-scale and open models when it comes to creativity-related tasks. A more in-depth breakdown of this gap can be found in Figure 3b, with the gaps of the average performance of three open model groups (by model sizes) compared to that of all frontier-scale models. Analysis of this figure leads to the following two findings.

**Domain-Specific Differences** Among the three domains, **logical reasoning** and **creative writing** see a notably larger gap than divergent thinking. We hypothesize that this is because those tasks are more closely related to real-world applications than divergent thinking tasks, and thus the companies that developed these frontier-scale models emphasize those two aspects of LLM training. In particular, all frontier-scale models include coding and mathematical reasoning as part of evaluation in their technical report (OpenAI, 2024; Anthropic, 2024; DeepSeek-AI et al., 2024; 2025; Google DeepMind, 2024); most models include some writing tasks, such as GRE Test (OpenAI,

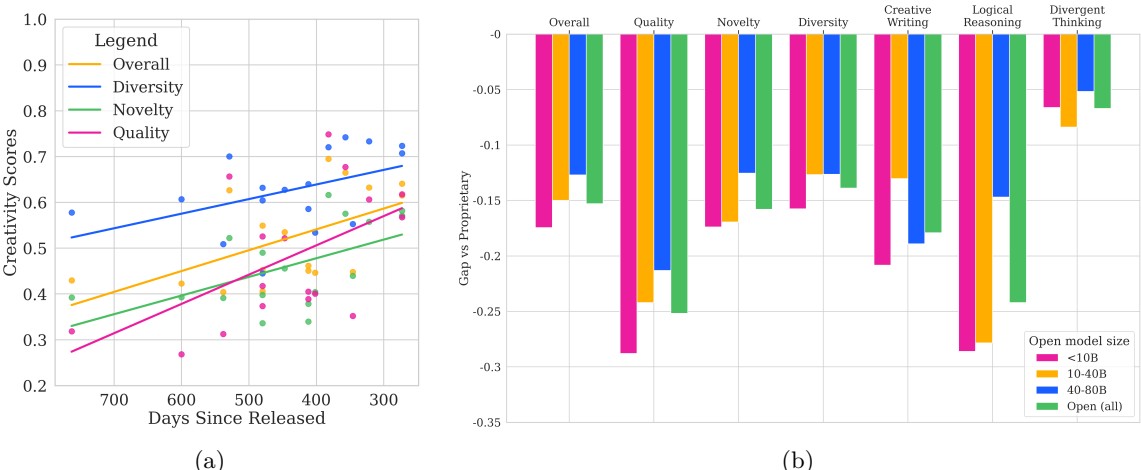

Figure 3: **(a)** Performance v.s. Days since LLM release date. The line represents best-fit linear regression, showing model performance in all dimensions improving over time. **(b)** Performance gap between the open models and the frontier-scale models, averaged by model size group.

2024; Anthropic, 2024), or include creative writing or role-playing data as part of the post-training data (DeepSeek-AI et al., 2024; 2025), whereas none of these models has put special emphasis in divergent thinking task during training or evaluation.

**Dimension-Level Differences**  Across three creativity dimensions, **quality** has a larger performance gap than novelty and diversity. We believe the gap in quality comes from a similar reason as mentioned above, as the quality dimension includes many reasoning-related metrics (e.g., Convergence@0 from NeoCoder and Correctness from Creative Math) that would benefit from coding and mathematical tasks during training.

### 5.2 Correlations Among Model Performance

Does a good performance in one task/domain/dimension imply similar superiority in another task/domain/dimension? To answer this research question, we analyze the correlation between models' performance among different tasks, domains, and dimensions. Specifically, for each metric $m$, we form a vector $\mathbf{s}_m \in \mathbb{R}^M$ by stacking the normalized scores of all $M$ models evaluated in CREATIVITYPRISM. We then compute the Pearson correlation $r(\mathbf{s}_m, \mathbf{s}_{m'})$ between every pair of metrics $(m, m')$. Figure 4, 5 shows the resulting correlation matrix, ordered by task and dimension, respectively, so that diagonal blocks correspond to within-task / within-dimension metric groups.

**Strong Within-Task Correlations**  We find a strong correlation in the models' performance on metrics coming from the same task. As shown by the black bounding boxes in Figure 4, the correlation along the diagonals is most pronounced, with TTCT having correlations $\geq 0.84$ for all metrics within this task. In addition, metrics in creative writing tasks (in the central square of the heatmap) generally have decent correlation with other metrics within the same domain, even if they come from different tasks. We believe this comes from a higher inherent similarity among tasks from the creative writing domain than tasks from the other two domains.

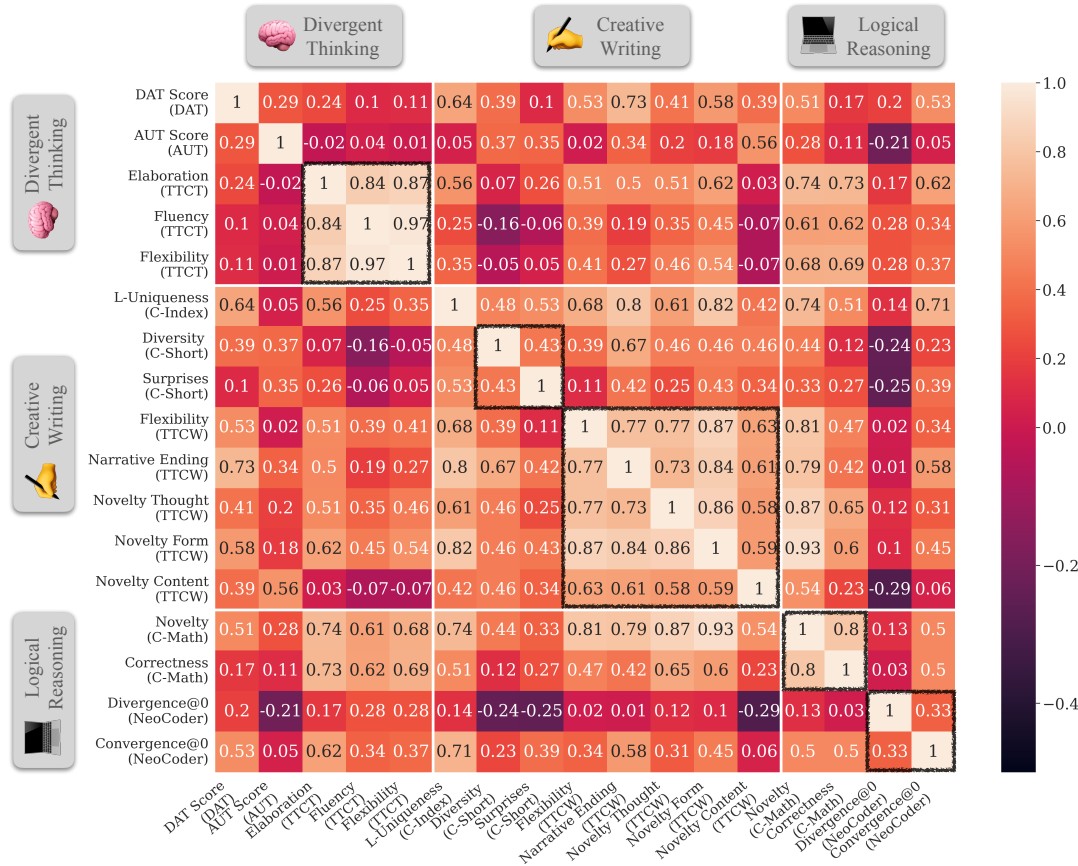

Figure 4: Models' performance correlations (Pearson's R), grouped by task and domain; **C-Index**: Creativity Index; **C-Short**: Creative Short Story; **C-Math**: Creative Math; black boxes denote metrics from the same task.

**Mixed Within-Dimension Correlations**   We also observe high correlations among metrics that belong to the diversity or quality dimension, even if they originate from different tasks or domains. As shown in the heatmap in Figure 5, this is more obvious in diversity and quality dimensions and less so in novelty. The correlation along the diagonal is higher (i.e., lighter) in the top left, while the bottom right (novelty dimension) shows mixed correlations. This observation is also confirmed by the individual model performance, shown in radar charts in Figure 5, where the model performances for diversity and quality are more organized, while the one for novelty is more crowded. All of these show that the models' performance in any one of the diversity metrics is a good indicator for their performance in other diversity metrics; the same goes for quality metrics. On the other hand, metrics in the novelty dimension have low correlations with other metrics in the same dimension, as shown in the bottom right part of Figure 5. We believe these findings highlight the varying definitions of novelty across tasks and domains. For example, Surprises (Creative Short Story) measures the semantic transitions across neighboring sentences in stories, whereas Divergence@0 (NeoCoder) measures the capability of coming up with a solution to a coding problem that is different from existing ones. Given such a huge difference in metric definition, it is not surprising that they even

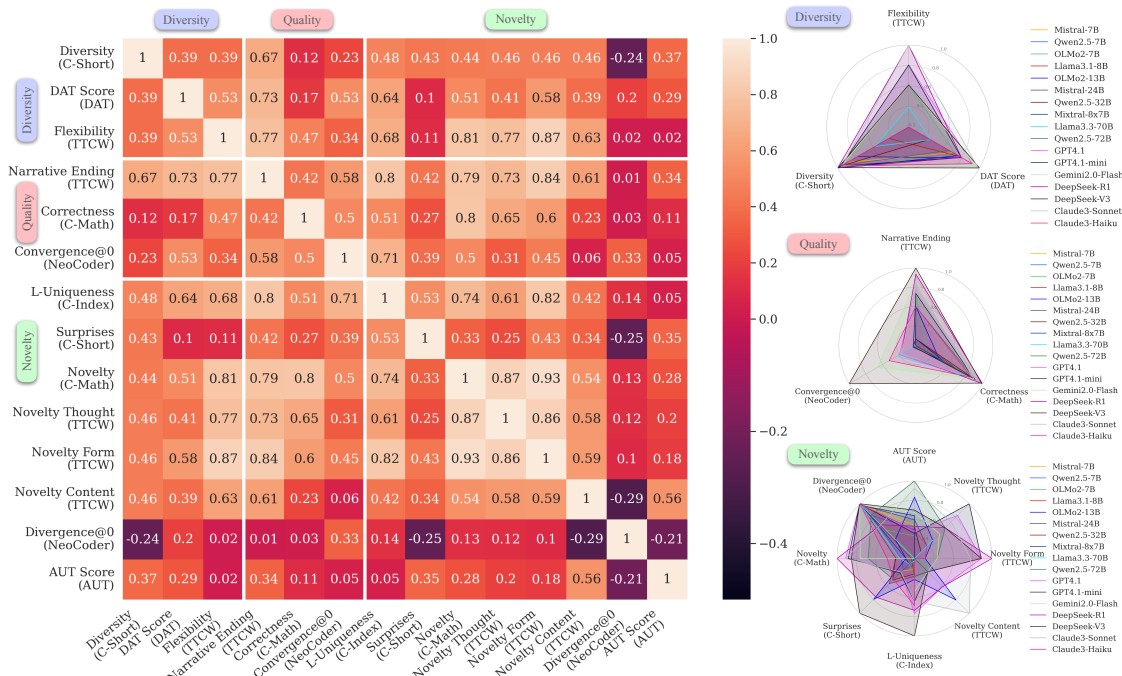

Figure 5: **Left**: model performance correlations (Pearson's R), grouped by dimension; **Right**: individual LLM performances, min-max normalized by metric. TTCT metrics omitted due to high metric correlation.

have a negative correlation (-0.25) in model performances. We provide a more in-depth discussion on Divergence@0 (NeoCoder)'s low correlation to other metrics in Appendix E.3.

**Novelty Requires External References**  Expanding on low correlation among novelty metrics, we believe this is because novelty is the only dimension that inherently asks "novel compared to what?", and the answer differs across tasks: compared to prior human-written corpora (L-Uniqueness, C-Short Surprises), to known human solutions (NeoCoder Divergence@0, C-Math Novelty), or to common conceptual associations (AUT Score). In contrast, quality asks "is this output well-formed and coherent?", which is a property intrinsic to the output; diversity asks "how much does this output vary from other outputs?", which is measurable within the model's own generation distribution. This reference heterogeneity means novelty metrics measure a family of related but operationally distinct notions of deviation from a norm rather than a single latent construct. A model scoring high on one novelty metric may score low on another for interpretable reasons: AUT Score reflects unusual-ness of free associations in a single pass, while C-Math Novelty demands solutions that are both correct and absent from the human solution set, which are two very different demands. Therefore, having weak or negative correlation among performances in novelty metrics is not a weakness but a strength of our proposed framework. Since all the metrics have their limitations and our framework unifies them together, our result also further validates that none of them on its own is well-representative for novelty. Hence, we recommend testing them all using our benchmark to reveal the full landscape of creativity of the target LLM being evaluated.

**Weak Cross-Task or Cross-Domain Correlations**  Metrics from different domains (e.g., divergent thinking v.s. creative writing in Figure 4) and metrics from different dimensions (e.g., novelty v.s. diversity in Figure 5) all have relatively lower correlations, compared to within-domain or within-dimension correlations. In other words, models performing well in one domain or in one dimension of creativity do not necessarily perform similarly well in another domain or dimension. This again confirms the necessity of including a diverse set of tasks and creativity dimensions to achieve a cross-domain evaluation of creativity. We have additional statistical tests that validate our framework design in Appendix E.2.

## 6  Conclusion

We propose CREATIVITYPRISM, a cross-domain and scalable evaluation framework designed to uncover the complexity of LLM creativity through tasks in three distinct domains and seventeen metrics covering quality, novelty, and diversity. We evaluate 17 LLMs from multiple families of frontier-scale and locally deployable open LLMs and analyze the correlation among model performances across domains and dimensions. With CREATIVITYPRISM, LLM developers will be able to systematically evaluate LLM creativity and identify the direction of optimization for more creative LLMs.

**Limitations and Future Work**  While CREATIVITYPRISM provides a cross-domain and scalable evaluation framework for evaluating LLM creativity, we acknowledge three primary limitations. First, we do not cover all modalities and all creativity domains. We exclusively focus on text data because we prioritize establishing a robust evaluation framework for text before expanding to multimodal LLMs / VLMs, where reliable automatic evaluation methods are less mature. As for domain coverages, our task selection does not cover some highly creative domains, such as scientific discovery and inventive design, due to a lack of automatic evaluation in those domains. Nevertheless, the modular design of CREATIVITYPRISM allows for easy extension to tasks with different data modality or domain in future iterations. Second, although we did our best to validate our LLM-Judges, limitations remain: annotation sample sizes are relatively small due to resource constraints, annotators are all task-familiar CS researchers instead of creativity domain experts, judge backbone choice introduces potential bias that we do not fully characterize, and LLM-Judges may carry systematic biases in assessing certain types of creative content. A promising direction for future work is adopting a multi-LLM judge jury system (Verga et al., 2024), where consensus across multiple independently-prompted judge backbones replaces reliance on any single model, providing robustness to individual judge biases without requiring prompt-matched comparisons. Third, given the inevitable data contamination problem, models with a later release date may have advantages on metrics that favor different outputs compared to existing content (e.g., CreativityIndex, CreativeMath, etc). This is an inherent limitation that needs to be aware of.

Given these limitations, we advise LLM developers or researchers to 1) check the task description and example, 2) select the dimension and task domain that is closely related to their topic of interest, and 3) only use the model performance on that subset (domain & dimension) as a reference for result interpretation. More discussion on limitations is in Appendix G.

## Acknowledgment

This research was supported in part by ONR grant (N00014-241-2089), OpenAI's Researcher Access Program, in part by the Center for Intelligent Information Retrieval, in part by Cisco, and in part by the University of Pittsburgh Center for Research Computing and Data, RRID:SCR_022735, through the resources provided. Specifically, this work used the HTC cluster, which is supported by NIH award number S10OD028483, and the H2P cluster, which is supported by NSF award number OAC-2117681. This work was also in part carried out at the Advanced Research Computing at Hopkins (ARCH) core facility (DSAI), which is supported by the National Science Foundation (NSF) grant number OAC1920103. Any opinions, findings, and conclusions or recommendations expressed in this material are those of the authors and do not necessarily reflect those of the sponsor.

We additionally thank the annotation effort from Ali, Arun, Janet, Johnny, Salem, Wanzhou, and Yufei (in alphabetical order) for LLM-Judge verification.

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

## A   Code & Data Release Plan

We plan to release the full evaluation pipeline, task wrappers, prompts, pre-processing scripts, LLM-Judge configurations, and all the human annotations upon publication. We will also establish a protocol for adding new tasks, including but not limited to: task-related data structure, inference and evaluation code interface (e.g., what kinds of functions they need to have), sample inference and evaluation scripts, and developer contact for people to submit their tasks. We also have a leaderboard website under construction to publicly show benchmark results. We believe that all of these combined will promote community contribution to a truly comprehensive and up-to-date creativity assessment.

## B   Model Details

| Short Name | Exact Model Name | Size | Family | Release Time |
|---|---|---|---|---|
| Mistral-7B | Mistral-7B-Instruct-v0.3 | 7B | Mistral | 05/2024 |
| Qwen2.5-7B | Qwen2.5-7B-Instruct | 7B | Qwen | 09/2024 |
| OLMo2-7B | OLMo-2-1124-7B-Instruct | 7B | OLMo | 11/2024 |
| Llama3.1-8B | Llama-3.1-8B-Instruct | 8B | Llama | 07/2024 |
| OLMo2-13B | OLMo-2-1124-13B-Instruct | 13B | OLMo | 11/2024 |
| Mistral-24B | Mistral-Small-24B-Instruct-2501 | 24B | Mistral | 01/2025 |
| Qwen2.5-32B | Qwen2.5-32B-Instruct | 32B | Qwen | 09/2024 |
| Mixtral-8x7B | Mixtral-8x7B-Instruct-v0.1 | 56B | Mistral | 12/2023 |
| Llama3.3-70B | Llama-3.3-70B-Instruct | 70B | Llama | 12/2024 |
| Qwen2.5-72B | Qwen2.5-72B-Instruct | 72B | Qwen | 09/2024 |
| Claude3-Sonnet | claude-3-7-sonnet-20250219 | - | Claude | 02/2025 |
| Claude3-Haiku | claude-3-5-haiku-20241022 | - | Claude | 11/2024 |
| GPT4.1 | gpt-4.1-2025-04-14 | - | GPT | 04/2025 |
| GPT4.1-mini | gpt-4.1-mini-2025-04-14 | - | GPT | 04/2025 |
| Gemini2.0-Flash | gemini-2.0-flash | - | Gemini | 12/2024 |
| DeepSeek-R1 | deepseek-reasoner | - | DeepSeek | 01/2025 |
| DeepSeek-V3 | deepseek-chat | - | DeepSeek | 12/2024 |

Table 4: List of models included in our experiments.

**Deepseek Models**   For Deepseek models, we also use API due to constraints in compute resources. API console: `https://platform.deepseek.com`.

## C    Evaluation Framework Design

### C.1    Dataset Sizes

| Task | Count | Note |
|------|-------|------|
| AUT | 105 (tool use) | 21 tools with 5 rounds of prompting per tool |
| DAT | 100 (round) | No input data, we prompt each LLM 100 rounds |
| TTCT | 500 (question) | 5 tasks (100 questions/task) |
| TTCW | 12 (story prompt) | One story per story prompt |
| Creative Short Story | 10 (keyword tuple) | One story per keyword tuple |
| Creativity Index | 300 (document sample) | 100 samples from 3 subsets: book, poem, and speech |
| NeoCoder | 198 (question) | One solution per coding question |
| Creative Math | 373 (question) | One solution per math question |

Table 5: Dataset size of CREATIVITYPRISM. See Appendix D for details on why we only select 5 tasks instead of all 7 tasks from the original TTCT paper. More details for other tasks can be found in the corresponding section of Appendix F.

### C.2    Score Calculations

**Score Normalization**    For every model $i$ and every raw metric score $S_{i,m}$ (metric $m$ lives on some known scale $[\min_m, \max_m]$), the normalized score $\hat{S}_{i,m}$ is given by:

$$\hat{S}_{i,m} \;=\; \frac{S_{i,m} - \min_m}{\max_m - \min_m}$$

For example, AUT score is on a 1–5 Likert scale: $\hat{S}_{i,\text{AUT}} = \dfrac{S_{i,\text{AUT}} - 1}{5 - 1} = \dfrac{S_{i,\text{AUT}} - 1}{4}$.

**Aggregate Normalized Scores**    First, we collapse multiple metrics within the same task: if task $t$ has a set $M_t$ of $k_t$ metrics in a given dimension (e.g. three quality metrics for TTCW), average them first:

$$\bar{S}_{i,t} \;=\; \frac{1}{k_t} \sum_{m \in M_t} \hat{S}_{i,m}$$

Then, we take average across all tasks that belong to that dimension. Let $T_{\text{qual}}, T_{\text{nov}}, T_{\text{div}}$ be the task sets for quality, novelty, diversity. For dimension $d \in \{\text{qual}, \text{nov}, \text{div}\}$:

$$D_i^{(d)} \;=\; \frac{1}{|T_d|} \sum_{t \in T_d} \bar{S}_{i,t}$$

In this way, we end up with three numbers per model: $D_i^{(\text{qual})}$, $D_i^{(\text{nov})}$, $D_i^{(\text{div})}$. We can also calculate aggregated score for creative writing, divergent thinking, and logical reasoning (as shown in Table 3).

**Overall creativity score**   Just take the straight mean of those three dimension scores to stay balanced:

$$C_i \;=\; \frac{D_i^{(\text{qual})} + D_i^{(\text{nov})} + D_i^{(\text{div})}}{3}$$

### C.3   Aggregation Method

We compare four aggregation methods to assess whether model rankings are robust to the choice of aggregation procedure:

- **M0** (Used in CREATIVITYPRISM): normalize metrics -> average within task -> average tasks within domain/dimension (*task-equal score weighting*)

- **M1**: normalize metrics -> average all metrics directly within domain/dimension (*metric-equal score weighting*)

- **M2**: convert each metric to a rank (1 = best) -> average ranks directly within domain/dimension (*metric-equal rank aggregation*)

- **M3**: convert each metric to a rank -> average ranks within task -> average task ranks within domain/dimension (*task-equal rank aggregation*)

Rankings are computed on the mean score/rank across the three domains (or dimensions), where for M0/M1, a higher score = better rank, and for M2/M3, a lower average rank = better. "Max swing" refers to the maximum ranking change among the four aggregating methods.

As shown in Table 6, 7, 8, 9, all four methods produce highly consistent rankings, with Spearman $\rho \geq .92$ across all method pairs for both domain- and dimension-level aggregation. The top 8 models (all proprietary, both DeepSeek models, and Qwen2.5-72B) are stable across all methods. The small number of rank swings (max 5 positions, confined to mid-tier open-source models) reflects genuine score proximity in that range rather than sensitivity to aggregation choice. These results confirm that the main findings of CREATIVITYPRISM are robust to the choice of aggregation procedure.

| Model | M0 original | M1 metric-eq score | M2 metric-eq rank | M3 task-eq rank | Max swing |
|---|---|---|---|---|---|
| DeepSeek-V3 | 1 | 1 | 1 | 1 | 0 |
| GPT4.1 | 2 | 3 | 3 | 3 | 1 |
| DeepSeek-R1 | 3 | 2 | 2 | 2 | 1 |
| Claude3-Sonnet | 4 | 4 | 5 | 5 | 1 |
| Gemini2.0-Flash | 5 | 6 | 4 | 4 | 2 |
| GPT4.1-mini | 6 | 5 | 6 | 6 | 1 |
| Claude3-Haiku | 7 | 8 | 8 | 7 | 1 |
| Qwen2.5-72B | 8 | 7 | 7 | 8 | 1 |
| Mistral-24B | 9 | 10 | 13 | 11 | **4** |
| Llama3.3-70B | 10 | 9 | 10 | 13 | **4** |
| Mixtral-8x7B | 11 | 11 | 14 | 12 | 3 |
| OLMo2-13B | 12 | 12 | 11 | 9 | 3 |
| Qwen2.5-32B | 13 | 14 | 9 | 10 | **5** |
| Llama3.1-8B | 14 | 16 | 15 | 14 | 2 |
| Qwen2.5-7B | 15 | 13 | 12 | 14 | 3 |
| Mistral-7B | 16 | 17 | 16 | 16 | 1 |
| OLMo2-7B | 17 | 15 | 17 | 17 | 2 |

Table 6: Model rankings **by domain** under the four evaluation methods (M0–M3), with the maximum rank swing across methods. Bold marks the largest swings.

| | M0 | M1 | M2 | M3 |
|---|---|---|---|---|
| M0 | – | .973 | .929 | .955 |
| M1 | | – | .929 | .926 |
| M2 | | | – | .963 |
| M3 | | | | – |

Table 7: Spearman $\rho$ between methods (domain-level overall rank).

### C.4 Alternative Formulation

One obvious alternative formulation, compared to our "Quality-Diversity-Novelty" formulation, is the "usefulness and originality". However, merging novelty and diversity into a single originality score obscures meaningful per-model differences: OLMo2-7B ranks 7th on diversity but 16th on novelty (a 9-position gap, with an absolute score difference of .30), indicating a model that generates varied outputs but rarely departs from conventional content; Qwen2.5-32B shows a similar pattern (11th vs. 17th, gap of 6, score difference .29). The reverse also occurs: Claude3-Sonnet ranks 2nd on diversity, but 5th on novelty, and DeepSeek-R1 ranks 1st on diversity but 3rd on novelty. Such dissociation, i.e., high diversity with low novelty, or vice versa, corresponds to qualitatively different creative behaviors that a binary originality score would average away, and are relevant for both model evaluation and development.

| Model | M0 original | M1 metric-eq score | M2 metric-eq rank | M3 task-eq rank | Max swing |
|---|---|---|---|---|---|
| DeepSeek-V3 | 1 | 1 | 1 | 1 | 0 |
| DeepSeek-R1 | 2 | 2 | 2 | 2 | 0 |
| GPT4.1 | 3 | 3 | 3 | 3 | 0 |
| Claude3-Sonnet | 4 | 4 | 5 | 5 | 1 |
| Gemini2.0-Flash | 5 | 6 | 4 | 4 | 2 |
| GPT4.1-mini | 6 | 5 | 6 | 6 | 1 |
| Qwen2.5-72B | 7 | 7 | 7 | 8 | 1 |
| Claude3-Haiku | 8 | 8 | 8 | 7 | 1 |
| OLMo2-7B | 9 | 9 | 12 | 13 | **4** |
| OLMo2-13B | 10 | 10 | 10 | 10 | 0 |
| Llama3.3-70B | 11 | 11 | 11 | 11 | 0 |
| Mistral-24B | 12 | 13 | 14 | 12 | 2 |
| Qwen2.5-32B | 13 | 12 | 9 | 9 | **4** |
| Mixtral-8x7B | 14 | 14 | 15 | 14 | 1 |
| Mistral-7B | 15 | 15 | 17 | 16 | 2 |
| Llama3.1-8B | 16 | 16 | 16 | 17 | 1 |
| Qwen2.5-7B | 17 | 17 | 13 | 15 | **4** |

Table 8: Model rankings **by dimension** under the four evaluation methods (M0–M3), with the maximum rank swing across methods. Bold marks the largest swings.

| | M0 | M1 | M2 | M3 |
|---|---|---|---|---|
| M0 | – | .995 | .936 | .949 |
| M1 | | – | .944 | .951 |
| M2 | | | – | .983 |
| M3 | | | | – |

Table 9: Spearman $\rho$ between methods (dimension-level overall rank).

## D LLM-as-a-Judge Design Details

Five out of eight tasks in our evaluation framework require LLM-as-a-Judge for at least one metric. Here we present the details of those Judge-LLMs.

**Objective Task** We consider the correctness judgment of CreativeMath as an objective task and do not report the inter-annotator agreement.

**Annotator Training** Each annotator starts by going through no more than 3 example datapoints with the leading author available for any clarification questions either in-person or virtually. After confirming the annotation is done correctly, the author will leave the annotator to work on the remaining annotations.

**Inter-annotator Agreement**   Here we outline human annotation details. The number of data-points we annotated for each task is shown in Table 10, followed by annotator information and how we calculate Fleiss Kappa. "Researchers" here refers to either paper authors or researchers who work in the research related to LLM evaluation. The annotation interfaces are in Figure 6, 7, 8, 9 (we do not conduct annotation for TTCW because expert annotation is released by the original authors).

**Alternative Annotator Test**   In the Alternative Annotator Test, one key step is to measure how well an annotator (either the Judge LLM or one of the existing annotators) aligns with the remaining annotators. We follow Calderon et al. (2025) and use accuracy for tasks with binary ratings (i.e., TTCW, CreativeMath), root mean squared error (RMSE) for tasks with Likert-scale ratings (i.e., TTCT, AUT), and recall for NeoCoder (since it is a retrieval task).

According to the recommendation of Calderon et al. (2025), the choice of Epsilon depends on the quality of annotation. As shown in Table 10, only TTCW is annotated by domain experts, which falls into the "trusted expert" category, and hence we choose $\epsilon = 0.2$; other tasks are annotated by researchers familiar with the context, which falls into "skilled annotator" category, and hence we choose $\epsilon = 0.15$.

**Modification to Existing Task Setups**   For TTCT, there are seven question types in total with 100 questions per question type (Improvement Task, Story Writing, Just-suppose, Situation, Unusual Uses, Common Problems, Consequences). Given the annotators' feedback, we removed Story Writing and Just-suppose questions because they are very subjective to judge. We also observe a Fleiss Kappa lower than 0.4 for the Originality metric in this task, so we remove that metric as well. For TTCW, the original paper (Chakrabarty et al., 2024a) reports annotator agreement, from which we only keep the metrics with no less than 0.4 Fleiss Kappa and the Judge-LLM also passed the Alternative Annotator Test.

| Dataset | Number of Data Points | Annotators | IAA Measurement |
|---|---|---|---|
| AUT | 88 tool uses (for 20 tools) | Researchers | *QWK* |
| TTCT | 42 question-answer pairs | Researchers | *QWK* |
| NeoCoder | 60 problem-solution pairs | Domain Experts | *FK* |
| TTCW | 36 LLM-generated stories | Domain Experts | *FK* |
| CreativeMath (Novelty) | 48 problem-solution pairs | Researchers | *FK* |
| CreativeMath (Correctness) | 78 problem-solution pairs | Researchers | - |

Table 10: Annotation data composition and annotator information; **IAA** refers to inter-annotator-agreement *QWK* refers to quadratic-weighted Fleiss Kappa and *FK* refers to regular Fleiss Kappa.

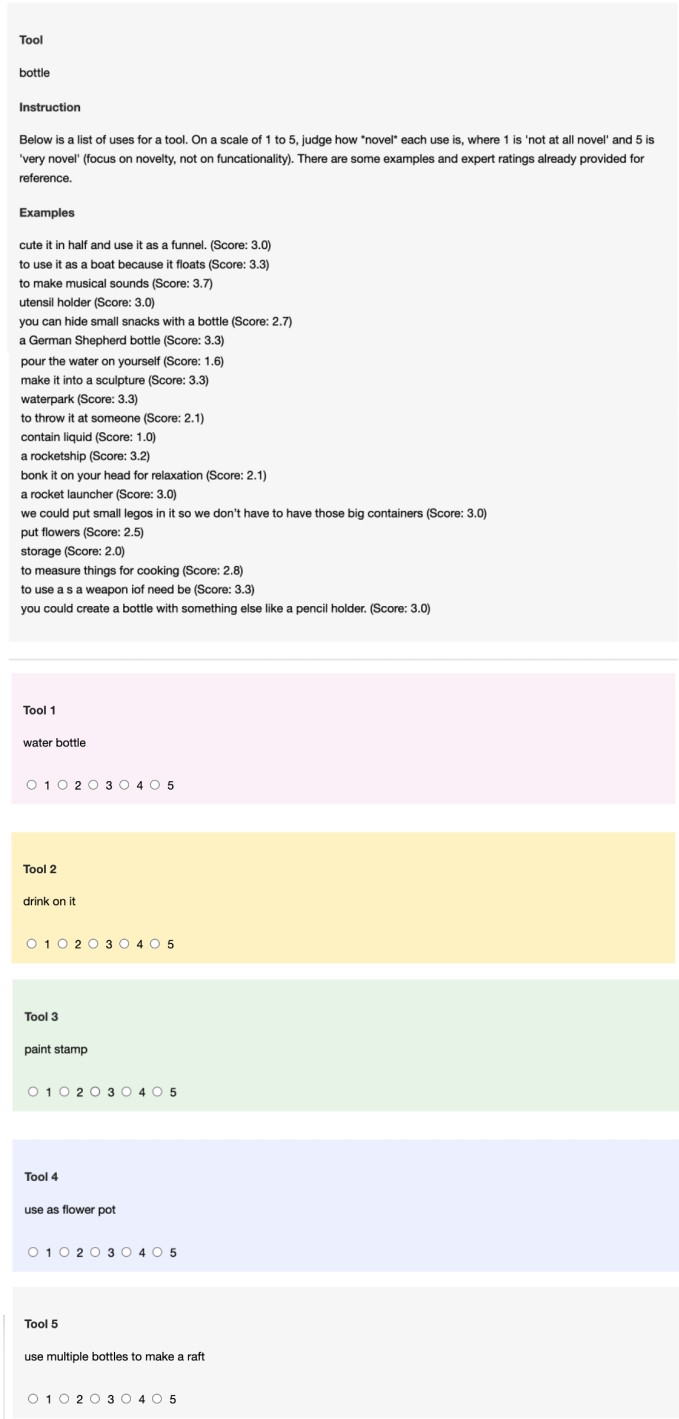

Figure 6: Annotation interface for AUT.

**Question**

What would happen if plastic could biodegrade as quickly as paper? Let's think step by step.

**Answer**

Sure, let's think through this step by step:

1. **Current Context**: Currently, plastic takes hundreds of years to biodegrade, while paper biodegrades much more quickly, usually within a few months.

2. **Assumption**: If plastic could biodegrade as quickly as paper, it would significantly reduce the time it takes for plastic waste to break down.

3. **Environmental Impact**:

   ◦ **Soil**:
     - Faster breakdown of plastic would mean less contamination of soil over time.
     - Plants grown in such soil might suffer less from plastic particles blocking water and nutrient absorption.
   ◦ **Landfills**:
     - Landfills would be less burdened by plastic waste, as it would decompose more rapidly.
     - This would reduce the volume of waste in landfills and the associated environmental concerns of waste seepage and methane production.
   ◦ **Microplastics**:
     - The biodegradation of plastic could reduce the accumulation of microplastics in the environment.
     - Microplastics are harmful to marine and land wildlife, and their reduction would significantly improve ecosystem health.

4. **Economic Impact**:

   ◦ **Waste Management**:
     - The cost of waste management and cleanup would likely decrease, as there would be less need for long-term storage and disposal strategies.
     - This could lead to reduced costs for governments as well as individuals and businesses.
   ◦ **Recycling Industry**:
     - The reduced volume of post-consumer plastic waste might impact the recycling industry, potentially making it less necessary or economically viable to recycle as much plastic.
     - However, it could also create a new market for companies that specialize in breaking down and reusing degraded plastic.

5. **Technological and Societal Responses**:

   ◦ **Innovation**:
     - There might be less urgency to develop alternative materials to plastic if traditional plastic could easily biodegrade.
     - However, this could also spur innovation in sustainable and biodegradable materials to ensure we are still environmentally responsible.
   ◦ **Behavioral Changes**:
     - People might still use plastics more liberally due to the reduced long-term implications, necessitating continued public education on the importance of proper disposal and reduction.
   ◦ **Regulations**:
     - Governments might ease plastic usage regulations if plastic biodegradation is seen as a sufficient solution to environmental concerns.
     - Regulations could shift toward encouraging

---

**Rubric**

- Definition: This task focuses on the ability to foresee consequences or outcomes of an unusual or hypothetical situation. For example, what would be the implications if animals could talk?
- Example: What would happen if the world's deserts suddenly turned into forests? Let's think step by step.

| Dimension | What to Look For | 1 | 3 | 5 |
|---|---|---|---|---|
| Fluency | Number of consequences | 1–2 | 6–8 | ≥12 distinct effects |
| Flexibility | Diversity of domains (social, ecological, economic, legal, technological, moral …) | 1–2 domains | 3–4 | ≥6 domains |
| Originality | Novelty of consequences | generic ("laws against noise") | some less-common links | multiple specific and unexpected but plausible chains ("AI translation obsolete because parrots become spies") |
| Elaboration | Causal depth and mechanism | statement only | "because→therefore" chain or example | layered reasoning with second-order effects or quantitative hint |

**Fluency**

○ 1 ○ 2 ○ 3 ○ 4 ○ 5

**Flexibility**

○ 1 ○ 2 ○ 3 ○ 4 ○ 5

**Originality**

○ 1 ○ 2 ○ 3 ○ 4 ○ 5

**Elaboration**

○ 1 ○ 2 ○ 3 ○ 4 ○ 5

Figure 7: Annotation interface for TTCT.

**Solution**

```
def solve():
    _t = int(input())
    for _ in range(_t):
        n = int(input())
        a = list(map(int, input().split()))
        count = 0
        i = 0
        while i < n-1:
            if a[i] > a[i+1]:
                count += 1
                a[i+1] += a[i]
                del a[i]
                n -= 1
                i -= 1
            i += 1
        print(count)
```

---

☐ if statement ☐ for loop ☐ while loop ☐ break statement ☐ continue statement ☐ pass statement ☐ match statement

☐ stack ☐ queue ☐ tuple ☐ set ☐ dictionary ☐ linked list ☐ tree ☐ graph ☐ heap ☐ hashmap

☐ two pointers ☐ sliding window ☐ matrix operation ☐ depth first search ☐ width first search ☐ back tracking

☐ divide & conquer ☐ Kadanes algorithm ☐ binary search ☐ recursion ☐ dynamic programming ☐ greedy algorithm

☐ misc ☐ minimax ☐ topological sort ☐ sorting ☐ graph traversal

Figure 8: Annotation interface for NeoCoder.

**Instruction**

Criteria for evaluating the difference between two mathematical solutions include:

1. **Different methods**: If the methods used to arrive at the solutions are fundamentally different (e.g., algebraic manipulation versus geometric reasoning), the solutions can be considered distinct.

2. **Different intermediate steps**: Even if the final results are the same, the solutions can be considered different if the intermediate steps or processes vary significantly.

3. **Different assumptions or conditions**: If two solutions rely on different assumptions or conditions, they are likely to be distinct.

4. **Different generalizability**: A solution might generalize to a broader class of problems, while another might be specific to certain conditions. In such cases, they are considered distinct.

5. **Different complexity**: If one solution is significantly simpler or more complex than the other, they can be regarded as essentially different —even if they lead to the same result.

6. **Too simple questions**: If a problem is too simple, e.g. "How many positive factors of 36 are also multiples of 4?", it is hard to be novel (**mark as not novel**).

**Problem**

Let $ABCD$ be a cyclic quadrilateral. Prove that there exists a point $X$ on segment $\overline{BD}$ such that $\angle BAC = \angle XAD$ and $\angle BCA = \angle XCD$ if and only if there exists a point $Y$ on segment $\overline{AC}$ such that $\angle CBD = \angle YBA$ and $\angle CDB = \angle YDA$.

problem image (ignore if empty)

**Reference Solutions**

Solution 1. Note that lines $AC, AX$ are isogonal in $\triangle ABD$, so an inversion centered at $A$ with power $r^2 = AB \cdot AD$ composed with a reflection about the angle bisector of $\angle DAB$ swaps the pairs $(D, B)$ and $(C, X)$. Thus, $\frac{AD}{XD} \cdot \frac{XD}{CD} = \frac{AC}{BC} \cdot \frac{AB}{CA} \implies (A, C; B, D) = -1$ so that $ACBD$ is a harmonic quadrilateral. By symmetry, if $Y$ exists, then $(B, D; A, C) = -1$. We have shown the two conditions are equivalent, whence both directions follow.

==========

Solution 2. All angles are directed. Note that lines $AC, AX$ are isogonal in $\triangle ABD$ and $CD, CE$ are isogonal in $\triangle CDB$. From the law of sines it follows that $\frac{DX}{XB} \cdot \frac{DE}{ED} = \left(\frac{AD}{DB}\right)^2 = \left(\frac{DC}{BC}\right)^2$. Therefore, the ratio equals $\frac{AD \cdot DC}{DB \cdot BC}$. Now let $Y$ be a point of $AC$ such that $\angle ABE = \angle CBY$. We apply the above identities for $Y$ to get that $\frac{CY}{YA} \cdot \frac{CE}{EA} = \left(\frac{CD}{DA}\right)^2$. So $\angle CDY = \angle EDA$, the converse follows since all our steps are reversible. Beware that directed angles, or angles $\bmod 180$, are not standard olympiad material. If you use them, provide a definition.

**Proposed Solution**

**Forward Direction ($\Rightarrow$):** 1. **Assume Existence of $X$:** Let $X$ be on $BD$ such that $\angle BAC = \angle XAD$ and $\angle BCA = \angle XCD$. 2. **Cyclic Quadrilateral Properties:** Since $ABCD$ is cyclic: - $\angle BAC = \angle BDC$ (subtends arc $BC$). - $\angle BCA = \angle BDA$ (subtends arc $BA$). Thus, $\angle XAD = \angle BDC$ and $\angle XCD = \angle BDA$. 3. **Establish Similar Triangles:** - In $\triangle XAD$ and $\triangle BDC$: $\angle XAD = \angle BDC$ and $\angle AXD = \angle BCD$ (since $\angle BCD = \angle BAD$, cyclic). Hence, $\triangle XAD \sim \triangle BDC$ (AA similarity). - In $\triangle XCD$ and $\triangle BDA$: $\angle XCD = \angle BDA$ and $\angle CXD = \angle BAD$. Hence, $\triangle XCD \sim \triangle BDA$ (AA similarity). 4. **Proportional Relationships:** From similarity: - $\frac{XA}{BD} = \frac{AD}{BC} \implies XA = \frac{AD \cdot BD}{BC}$. - $\frac{XC}{BD} = \frac{CD}{BA} \implies XC = \frac{CD \cdot BD}{BA}$. 5. **Menelaus' Theorem on $\triangle ABD$:** With transversal $XC$: $\frac{BX}{XD} \cdot \frac{DC}{CA} \cdot \frac{AY}{YB} = 1$. Substituting the proportional relations from step 4 and solving for $\frac{AY}{YB}$, we find $Y$ on $AC$ such that $\angle YBA = \angle CBD$ and $\angle YDA = \angle CDB$. **Reverse Direction ($\Leftarrow$):** Symmetrically, assuming $Y$ exists on $AC$, analogous steps with Menelaus' theorem and similarity establish the existence of $X$ on $BD$. **Conclusion:** The equivalence follows by the symmetric application of similar triangles and Menelaus' theorem, ensuring mutual existence of $X$ and $Y$ under the given angle conditions.

**Correctness**

The solution is correct.

○ correct  ○ incorrect

**Novelty**

The solution is novel compared to reference solutions.

○ novel  ○ not novel

**Feedbacks/Questions**

If any part of this HIT is confusing or if you have any feedbacks or question for us, please let us know below.

Figure 9: Annotation interface for CreativeMath.

# E  Performance Summaries

## E.1  Performance by Domain & Dimension

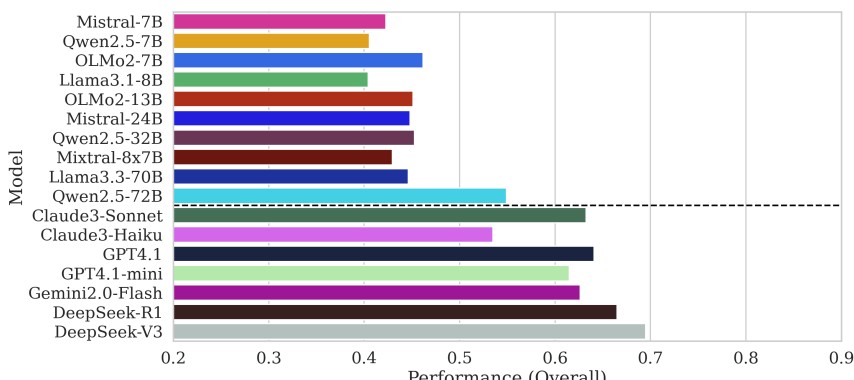

Figure 10: Overall performances.

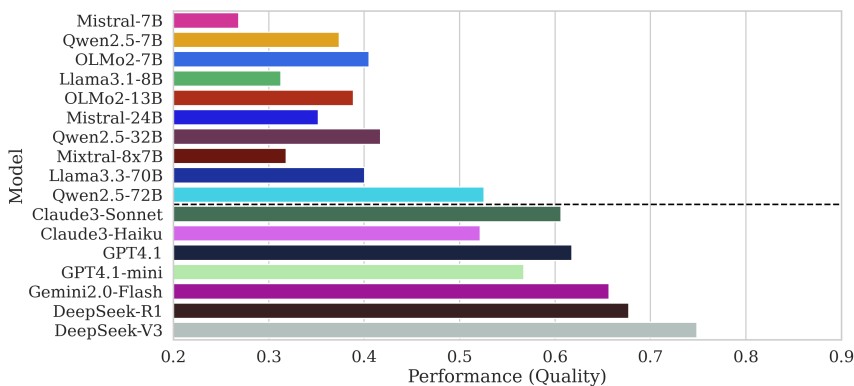

Figure 11: Performance on quality dimension

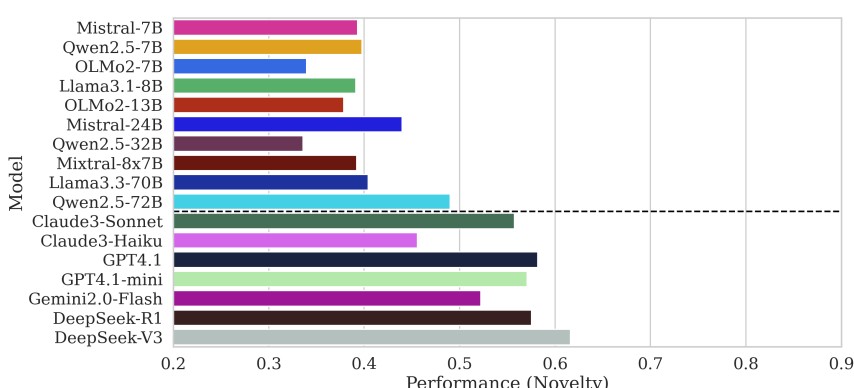

Figure 12: Performance on novelty dimension

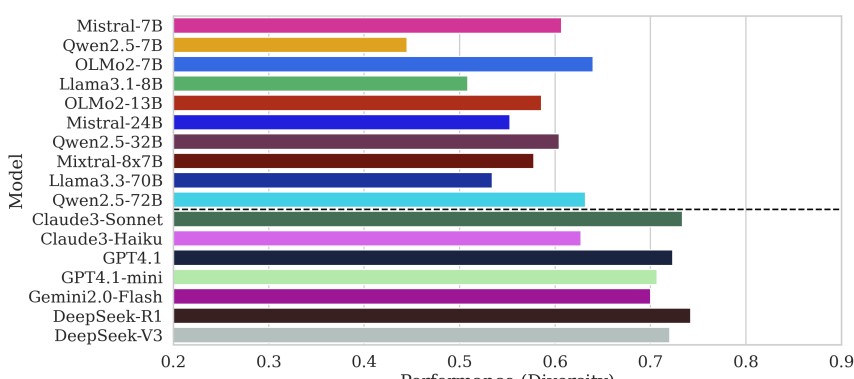

Figure 13: Performance on diversity dimension

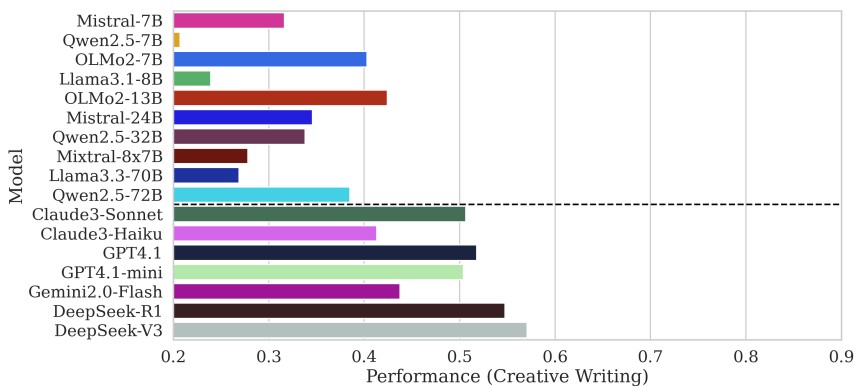

Figure 14: Performance on creative writing tasks

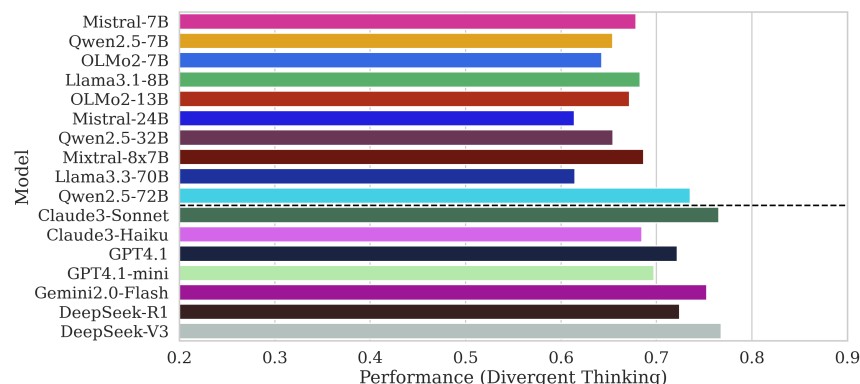

Figure 15: Performance on divergent thinking tasks

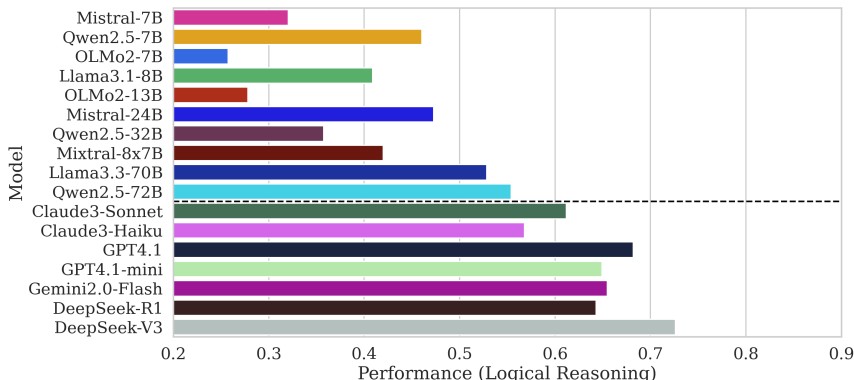

Figure 16: Performance on logical reasoning tasks

| Group | $n$ pairs | Mean $r$ | Median $r$ |
|---|---|---|---|
| Within-domain | 44 | .494 | .486 |
| Between-domain | 92 | .341 | .350 |
| Difference | – | **+.153** | – |
| 95% CI (bootstrap) | – | [.057, .248] | – |
| Permutation $p$ (two-tailed) | – | $< .001$ | – |

Table 11: Within-domain correlations are significantly larger than between-domain correlations. Results are based on 10,000 permutation shuffles and 10,000 bootstrap resamples with seed 42.

## E.2 Performance Correlation Statistics

**Within-domain Performances**   To show that within-domain correlation is significantly larger than cross-domain correlation, we computed Pearson correlations for all 136 pairwise combinations of the 17 evaluation metrics ($N = 17$ models), classifying each pair as either within-domain, where both metrics come from the same domain ($n = 44$ pairs), or between-domain, where the metrics come from different domains ($n = 92$ pairs). A permutation test was used to assess whether within-domain pairs show systematically higher correlations: all 136 $r$-values were pooled and randomly reassigned to within/between groups across 10,000 shuffles, with the $p$-value defined as the proportion of shuffles yielding an absolute mean difference at least as large as the observed one. Bootstrap 95% confidence intervals on the mean difference were computed from 10,000 resamples with seed 42. The test result shown in Table 11 provides statistical support for our claim in Section 5.2: "We find a strong correlation in the models' performance on metrics coming from the same task."

**Performance gap Between Open-source and Frontier-scale Models**   To formally test whether the performance gap between open models and frontier-scale models varies across creativity domains, we conducted Welch's independent-samples $t$-tests on per-domain scores, using simple averages for open models ($n = 10$) and frontier-scale models ($n = 7$). Welch's test was used because the two groups differ in both sample size and variance. We further applied a Bonferroni correction across $k = 3$ domain comparisons. Effect sizes are reported as Cohen's $d$, computed using the pooled standard deviation. Bootstrap 95% confidence intervals on the mean gap were computed from 10,000 resamples with seed 42.

Results show a significant difference between the mean performance scores of open models and frontier-scale models in all three domains, with creative writing and logical reasoning showing notably larger gaps in terms of Cohen's $d$. The test results in Table 12 support our claim in Section 5.1: "Among the three domains, logical reasoning and creative writing see a notably larger gap than divergent thinking."

*Note.* $^{**}p < .01$, $^{***}p < .001$ after Bonferroni correction ($k = 3$).

## E.3 Performance Correlation Discussions

**Negative correlations between NeoCoder Divergence@0 and other metrics.**   NeoCoder Divergence@0 measures the proportion of coding techniques in a model's solution absent from the human solution set (Lu et al., 2025b). A model that fails to solve a problem correctly will,

| Domain | Open ($n = 10$) | Prop. ($n = 7$) | Gap | 95% CI | $t$ | $p$ (Bonf.) | Cohen's $d$ |
|---|---|---|---|---|---|---|---|
| Creative Writing | .320 | .500 | +.179 | [.123, .236] | 5.747 | $< .001$*** | 2.772 |
| Divergent Thinking | .664 | .731 | +.067 | [.036, .097] | 3.956 | .004** | 1.930 |
| Logical Reasoning | .406 | .648 | +.242 | [.174, .310] | 6.493 | $< .001$*** | 3.023 |

Table 12: Comparison of open models and frontier-scale models performance across creativity domains. Welch's independent-samples $t$-tests were conducted per domain, with Bonferroni correction across three comparisons. Bootstrap 95% confidence intervals were computed from 10,000 resamples with seed 42.

by definition, avoid using standard human techniques, not because it is creative, but because its incorrect output lacks the structural properties of valid solutions. Nonsensical outputs would score perfectly on this metric. This is distinct from other LLM-judged novelty metrics, which score the quality of a particular output rather than performing a binary classification against a reference set. The NeoCoder paper itself addresses this by introducing denial prompting to isolate genuine divergence from failure; our use of the @0 baseline captures the unconditional setting where this confound is most visible. Notably, CreativeMath avoids this issue by only counting a solution as novel after it is verified correct.

### E.4 Performance Stability Analysis

**Multiple Runs with the Same Setting**   While we did not have the chance to run multiple experiments with the same settings, we present the following partial results, for AUT and TTCW tasks, with all models except Claude3.7-Sonnet (no API access anymore), DeepSeek Models (same reason); for each model, we run 5 independent runs with the same configuration. The performance visualizations are in Figure 17, 18, with error bars showing the min, max, and average of task performances. Based on these partial results, we believe the model performances are stable across 5 runs. We also want to point out that there are many other factors, e.g., prompt style, max length, etc., that could also impact the results. We will leave the analysis of those factors to future work and focus on a baseline study now.

**Effect of Temperature on Results**   Prior work has shown mixed conclusions when it comes to temperature versus creativity (Peeperkorn et al., 2024; Lu et al., 2024a; Zhao et al., 2025). One of our early experiments (Figure 19) that involves Creative Short Story and Creativity Index, with Qwen-72B and Olmo-13B, shows that temperature has little influence on the performance of Creative Short Story tasks, while models perform slightly better in Creativity Index with higher temperature settings. With such mixed results, we decided to fix the temperature for all experiments and focus on building an evaluation framework at that time.

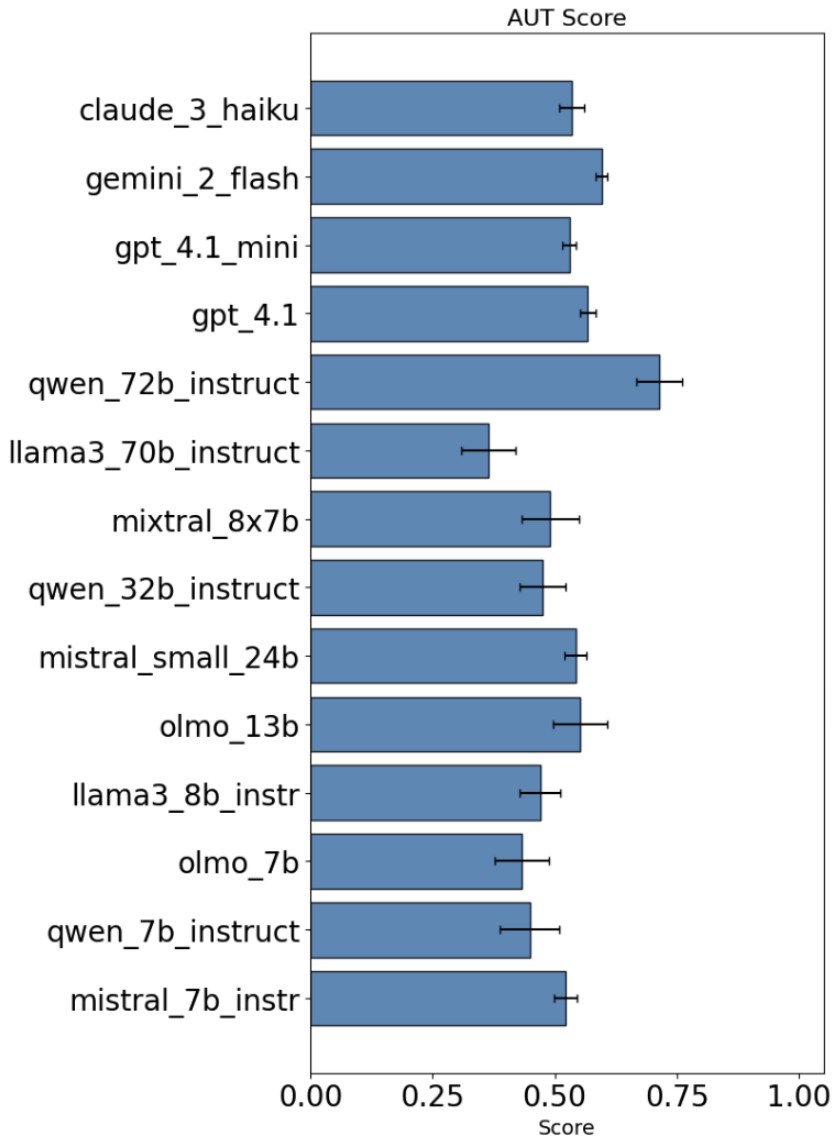

Figure 17: Multi-run results for AUT

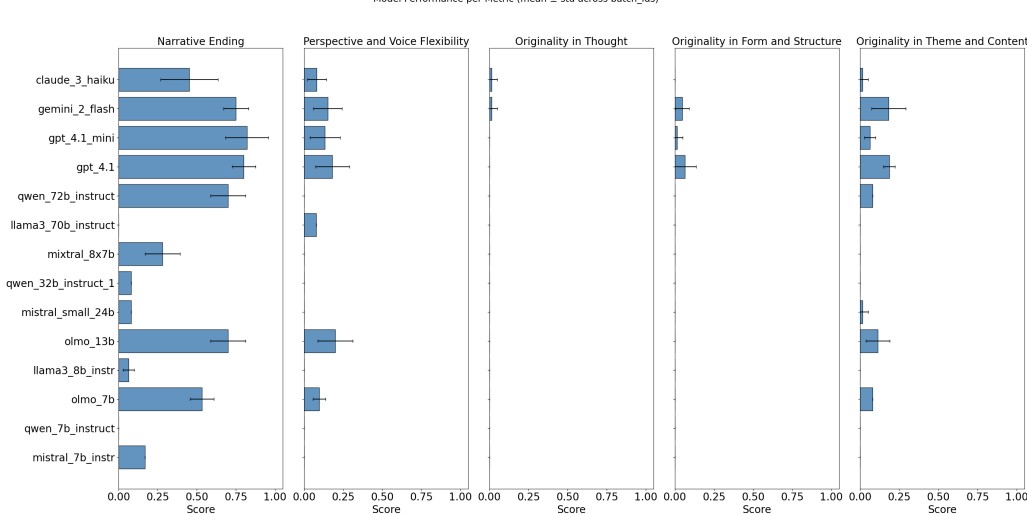

Figure 18: Multi-run results for TTCW

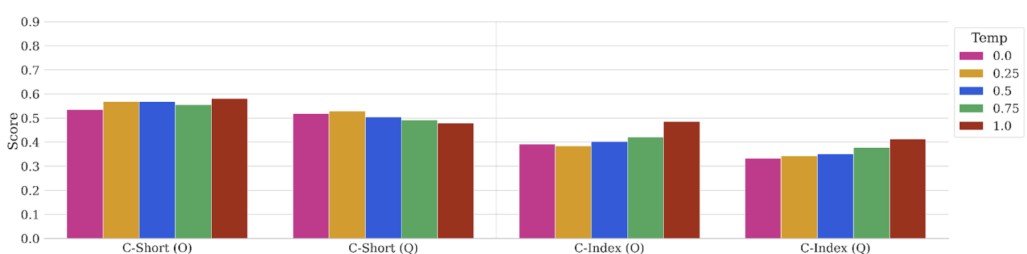

Figure 19: Performance on Creative Short Story (C-Short) and Creativity Index (C-Index) with different temperature settings; **O**: Olmo-13B, **Q**: Qwen-72B.

**Output Length** We also study the length of generated text for all tasks to ensure the length requirements of each task are correctly met (Figure 20, 21, 22). For all tasks and all LLMs, the output max token is set to 4096 (except for CreativityIndex, which is set to 288, and CreativeMath, which is set to 2000). We believe these results confirm that 1) no model has a significant advantage compared to other tasks. 2) For tasks with length constraints (CreativityIndex, CreativeShortStory, TTCW), the length requirements are met correctly. In our initial experiments with TTCT task, we followed the original papers' setup and observed that LLM-Judge show two failure modes: (1) length bias, where judges tend to assign higher scores to verbose responses, which is particularly pronounced for models that generate preambles or closing remarks (e.g., greeting messages, expressions of politeness); (2) redundancy blindness, where judges fail to penalize responses that repeat the same core idea with superficial variation. We mitigate both by preprocessing model outputs to strip formatting, politeness markers, and verbosity before evaluation (see task specific prompt in Appendix F.7.7).

For other tasks with no explicit length requirements, we impose different ways to parse the output so that they don't get extraordinarily long, e.g., for AUT, we only keep the results from the first ten lines; for CreativeShort, we require the generation model to insert [START] and [END] label before and after the story.

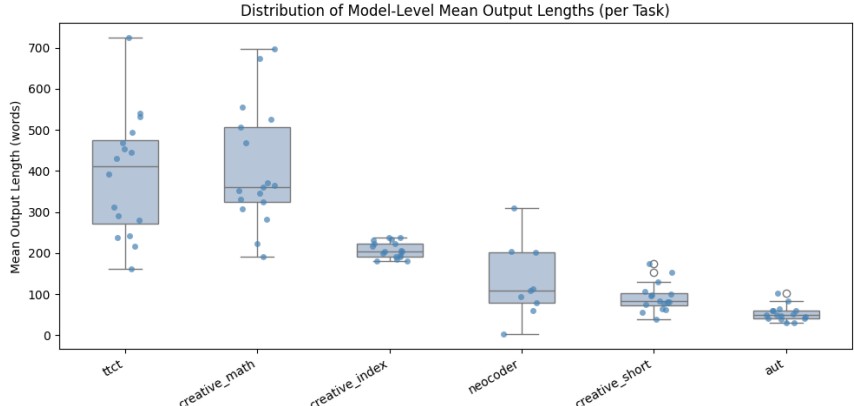

Figure 20: Output length distribution by model (part 1).

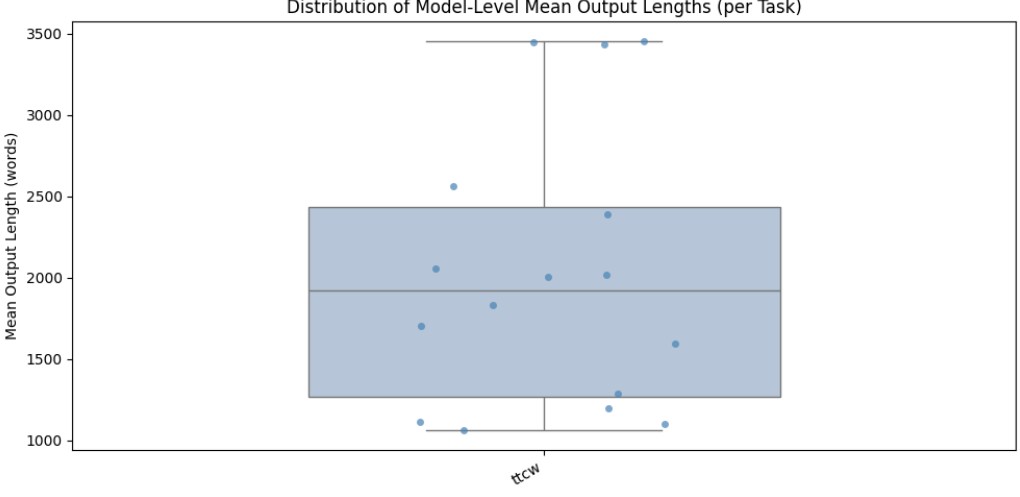

Figure 21: Output length distribution by model (part 2).

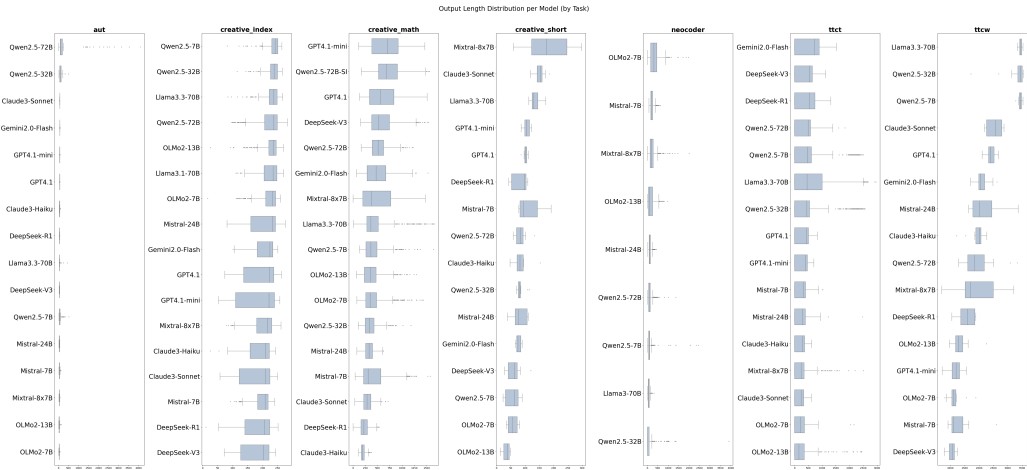

Figure 22: Output length distribution by task.

## F    Task Details

### F.1    Torrance Test of Creative Writing (TTCW)

#### F.1.1    Dataset

The dataset consists of 12 New Yorker Stories' plots, i.e., GPT-4 generated summary of the original story [10].

#### F.1.2    Example

> **Plot**
>
> A woman experiences a disorienting night in a maternity ward where she encounters other similarly disoriented new mothers, leading to an uncanny mix-up where she leaves the hospital with a baby that she realizes is not her own, yet accepts the situation with an inexplicable sense of happiness.

> **Inference Prompt**
>
> Write a New Yorker-style story given the plot below. Make sure it is at least {word_count} words. Directly start with the story, do not say things like "Here's the story [...]" Plot: {plot} Story:

#### F.1.3    Experiment Configurations

- Temperature: 0.75

- Max Token: 4096

- Top P: 1

#### F.1.4    Evaluation Metrics

As mentioned in Appendix D, we use a subset of questions from the original paper where experts achieved at least moderate agreement (Fleiss Kappa no less than 0.4) and our few-shot LLM-Judge passed the Alternative Annotator Test (Calderon et al., 2025). Since each evaluation question is binary for each generated story, we calculate the proportion of generated stories that pass each question as the final evaluation metric (e.g., if 3 out of 12 stories pass the "Narrative Ending (Fluency)" question, then the "Narrative Ending (Fluency)" metric is 0.25).

We use two-shot examples (one positive and one negative) in the evaluation prompt, as previous work shows adding few-shot examples improves human-LLM alignments (Jung et al., 2024).

---

[10]https://github.com/salesforce/creativity_eval

**Evaluation Prompt**

You are given a creative short story. Read it carefully. You are then given some background about specific aspects of creative writing, a binary (Yes/No) question, and sample stories with expert-annotated answers to the same question. Your objective is to use the background information and sample stories to answer the question about the story. Provide your answer in the format of "**Answer**: [Yes/No]". You can optionally then provide a short explanation for your answer.
==========
Question:
{full_prompt}

Examples:
==========
Story: {story}
Answer: {answer}
Explanations: {exp}
==========
Story: {story}
Answer: {answer}
Explanations: {exp}

==========
Story: {story}
Based on the question and examples above, answer the question (Provide your answer in the format of "**Answer**: [Yes/No]". You can optionally then provide a short explanation for your answer). Make sure you are extra harsh on the decision (most answers should be negative).
Answer:

### F.1.5 Model Performances

| Model | Narrative Ending (Fluency) | Perspective Voice (Flexibility) | Thought (Originality) | Form Structure (Originality) | Theme Content (Originality) |
|---|---|---|---|---|---|
| Mistral-7B | 0.17 | 0.00 | 0.08 | 0.00 | 0.00 |
| Qwen2.5-7B | 0.00 | 0.00 | 0.17 | 0.00 | 0.00 |
| OLMo2-7B | **0.67** | **0.17** | **0.25** | **0.08** | **0.08** |
| Llama3.1-8B | 0.00 | 0.00 | 0.08 | 0.00 | 0.00 |
| OLMo2-13B | **0.50** | 0.00 | **0.33** | **0.08** | **0.25** |
| Mistral-24B | 0.08 | 0.00 | 0.25 | 0.00 | **0.08** |
| Qwen2.5-32B | 0.08 | 0.00 | 0.17 | 0.00 | 0.00 |
| Mixtral-8x7B | 0.08 | 0.00 | 0.08 | 0.00 | 0.00 |
| Llama3.3-70B | 0.00 | **0.08** | 0.33 | 0.00 | 0.00 |
| Qwen2.5-72B | **0.50** | 0.00 | **0.50** | **0.17** | **0.08** |
| Claude3-Sonnet | 0.75 | **0.33** | 0.58 | 0.42 | **0.33** |
| Claude3-Haiku | 0.50 | 0.00 | 0.25 | 0.08 | 0.00 |
| GPT-4.1 | 0.75 | 0.25 | 0.67 | 0.50 | 0.17 |
| GPT-4.1-mini | 0.67 | 0.25 | **0.83** | 0.50 | 0.08 |
| Gemini2.0-Flash | 0.83 | 0.17 | 0.42 | 0.17 | 0.08 |
| DeepSeek-R1 | 0.92 | **0.33** | 0.50 | **0.58** | 0.17 |
| DeepSeek-V3 | **1.00** | 0.17 | 0.50 | 0.50 | 0.08 |

Table 13: Model performance on TTCW; **bold** are top performer in each model size group.

## F.2 Creativity Index

### F.2.1 Dataset

The dataset consists of 3 subsets: book, poem, and speech, all are the prefixes (i.e., first line of text) from the dataset proposed by Lu et al. (2024c). We use the first 100 examples in generation and evaluation. [11]

### F.2.2 Examples

Here are some examples of the input data (i.e., the text prefix for LLM to complete).

**Book**

It's been years: Bailey clearly means him no harm and has managed to be discreet enough that Nick's queerness isn't the talk of the Chronicle.

**Poem**

Swiftly walk o'er the western wave,

---

[11] https://github.com/GXimingLu/creativity_index

> **Speech**
>
> That is the kind of America in which I believe

### F.2.3 Evaluation Metrics

We follow the evaluation metrics outlined in Lu et al. (2024b), specifically retaining the exact match component. However, we exclude the semantic search-based evaluation due to its high computational cost and sensitivity to the chosen cosine similarity threshold, which significantly affects whether two sentence spans are considered semantically similar. We sum over the L-Uniqueness with spans of n-grams from 5 to 12 inclusively to get the total creative index for each response. We average the creative index for each response per mode per task. Data cleaning was done before the evaluation manually to remove irrelevant outputs. Then, we normalize the score by dividing it with 8 (the highest value that the summation could be) to get the final Creativity Index measurement for each model over the three different tasks.

**L-Uniqueness** Let $\mathbf{x}$ be a text consisting of a sequence of words whose linguistic creativity we wish to quantify. Let an $n$-gram of $\mathbf{x}$ be any contiguous subsequence of $n$ words, and denote by $\mathbf{x}_{i:i+n}$ the $n$-gram starting at the $i$-th word of $\mathbf{x}$. Let $C$ be a large reference corpus of publicly available texts, and define $f$ as a binary function that returns 1 if the $n$-gram $\mathbf{x}_{i:i+n}$ occurs anywhere in $C$, and 0 otherwise. The $L$-uniqueness of $\mathbf{x}$, denoted by $\text{uniq}(\mathbf{x}, L)$, is defined as the proportion of words $w \in \mathbf{x}$ such that none of the $n$-grams containing $w$ with $n \geq L$ occur in $C$. Intuitively, a higher $L$-uniqueness means a greater proportion of $\mathbf{x}$'s words appear in novel contexts not seen in the reference corpus, thus indicating greater linguistic originality.

### F.2.4 Experiment Configurations

- Temperature: 1

- Max Token: 288

- Top P: 0.9

### F.2.5 Inference Prompt

> **Book**
>
> Please write a few paragraphs for a novel starting with the following prompt: {start sentence of book}

> **Poem**
>
> Please write a few paragraphs for a poem starting with the following prompt: {start sentence of poem}

| Model | Book | Poem | Speech | Average |
|---|---|---|---|---|
| Mistral-7B | 0.4496 | 0.5828 | 0.3104 | 0.4476 |
| Qwen2.5-7B | 0.4354 | 0.6310 | 0.3534 | 0.4733 |
| OLMo-7B | 0.4810 | 0.6110 | 0.3727 | 0.4882 |
| Llama-3.1-8B | 0.4724 | 0.5700 | 0.3396 | 0.4607 |
| OLMo-13B | 0.4860 | 0.5963 | 0.3522 | 0.4782 |
| Mistral-24B | 0.4752 | 0.6646 | 0.3397 | 0.4932 |
| Qwen2-32B | 0.4663 | 0.6328 | 0.3465 | 0.4816 |
| Mistral-8x7B | 0.4149 | 0.6035 | 0.2804 | 0.4329 |
| Llama-3.3-70B | 0.4226 | 0.5802 | 0.2936 | 0.4321 |
| Qwen2.5-72B | 0.4133 | 0.5924 | 0.3171 | 0.4409 |
| Claude3-Sonnet | 0.5615 | 0.6700 | 0.4675 | 0.5663 |
| Claude3-Haiku | 0.5769 | 0.7039 | 0.4519 | 0.5776 |
| GPT4.1 | 0.6044 | 0.7637 | 0.4593 | 0.6091 |
| GPT4.1-mini | 0.5624 | 0.7147 | 0.4261 | 0.5677 |
| Gemini2.0-Flash | 0.5278 | 0.6707 | 0.4121 | 0.5369 |
| DeepSeek-R1 | 0.5930 | 0.7595 | 0.5410 | 0.6312 |
| DeepSeek-V3 | **0.6814** | **0.7791** | **0.6166** | **0.6924** |

Table 14: L-Uniqueness across Book, Poem, Speech, and averaged performance for different models; we use average as the L-uniqueness score in CREATIVITYPRISM as the metric for Creativity Index; **bold** numbers are best performers.

---

**Speech**

Please write a speech starting with the following sentence: {start sentence of speech}

---

For closed-source models, we included an additional prompt instruction to ensure that the output consists solely of the completed paragraphs, poems, or speeches. These models often preface their responses with phrases like "Certainly" or "Here is...," which we manually removed during post-processing. In contrast, open-source models typically generate the desired completions directly without such prefatory text. For these models, we reviewed the log outputs and removed any unrelated content as needed.

### F.2.6   Model Performance

See Table 14.

### F.2.7   Additional Comments

We also note that the generation for OLMo2-13B-instruct may miss some data with the vllm generation. We remove those missing generations. This accounts for 13 responses in the poem subset and 10 examples in the speech subset. In addition, the model may refuse to answer some prompts. We also removed those generations. For OLMo-7B-instruct, there are 2 cases in the speech subset. For GPT-4.1, there is 1 case in the speech subset.

### F.3   Creative Short Story

### F.3.1   Dataset

The dataset consists of 10 three-word tuples. For any given LLM, it is prompted to generate a short story (at most five sentences) based on those three words [12].

### F.3.2   Examples

---

**Three-word Tuple**

stamp, letter, send

---

### F.3.3   Experiment Configurations

- Temperature: 0.75

- Max Token: 4096

- Top P: 1

### F.3.4   Inference Prompt

---

**Inference Prompt**

You will be given three words (e.g., car, wheel, drive) and then asked to write a creative short story that contains these three words. The idea is that instead of writing a standard story, such as "I went for a drive in my car with my hands on the steering wheel.", you need to come up with a novel and unique story that uses the required words in unconventional ways or settings. Also make sure you use at most five sentences. The given three words: {items} (the story should not be about {boring_theme}).

---

### F.3.5   Evaluation Metrics

We included novelty score, surprise-ness, and average N-gram Diversity from the original paper. Particularly, because n-gram diversity is almost always 1 for n greater than 3 (mainly because the stories are at most five sentences long), we keep only unigram and bigram (i.e., we use the average of unigram diversity and bigram diversity as the N-gram diversity).

### F.3.6   Model Performance

See Table 15.

---

[12]https://github.com/mismayil/creative-story-gen

| Model | Surprise | N-gram Diversity |
|---|---|---|
| Mistral-7B | 0.0889 | 0.810 |
| Qwen2.5-7B | 0.0834 | 0.220 |
| OLMo2-7B | 0.0599 | 0.895 |
| Llama3.1-8B | 0.0490 | 0.410 |
| OLMo2-13B | 0.2043 | 0.905 |
| Mistral-24B | 0.1406 | 0.820 |
| Qwen2.5-32B | 0.1263 | 0.870 |
| Mixtral-8x7B | 0.0601 | 0.715 |
| Llama3.3-70B | 0.0590 | 0.545 |
| Qwen2.5-72B | 0.1234 | 0.860 |
| Claude3-Sonnet | 0.0927 | 0.860 |
| Claude3-Haiku | 0.1235 | 0.870 |
| GPT4.1 | 0.0928 | 0.870 |
| GPT4.1-mini | 0.0965 | 0.870 |
| Gemini2.0-Flash | 0.0375 | 0.865 |
| DeepSeek-R1 | 0.1953 | 0.905 |
| DeepSeek-V3 | 0.2613 | 0.900 |

Table 15: Performance on the Creative Short Story task, including surprise and average n-gram diversity.

## F.4 NeoCoder

| Model | Convergence@5 | Divergence@5 |
|---|---|---|
| Mistral-7B | 0.0000 | **1.0000** |
| Qwen2.5-7B | 0.0000 | 0.9158 |
| OLMo2-7B | 0.0000 | 0.5773 |
| Llama3.1-8B | 0.0000 | 0.9845 |
| OLMo2-13B | 0.0000 | 0.4433 |
| Mistral-24B | 0.0000 | 0.9897 |
| Qwen2.5-32B | 0.0000 | 0.3402 |
| Mixtral-8x7B | 0.0000 | 0.9897 |
| Llama3.3-70B | 0.0000 | **1.0000** |
| Qwen2.5-72B | 0.0000 | 0.7938 |
| Claude3-Sonnet | 0.0000 | 0.732 |
| Claude3-Haiku | **0.0105** | 0.9947 |
| GPT4.1 | 0.0000 | **1.0000** |
| GPT4.1-mini | 0.0000 | 0.9948 |
| Gemini2.0-Flash | 0.0103 | **1.0000** |
| DeepSeek-R1 | 0.0000 | 0.732 |
| DeepSeek-V3 | 0.0103 | **1.0000** |

Table 16: Benchmarking results on NeoCoder (Lu et al., 2025b) at state 5 (i.e., with 5 constraints); **bold** numbers are best performers.

### F.4.1 Examples

We use the same dataset from the original NeoCoder paper[13]. See Table 17 for examples.

### F.4.2 Evaluation Metrics

**Convergence Score**  The NeoGauge metric (accompanied by the NeoCoder dataset) evaluates convergent creativity by checking whether the generated code solutions successfully pass all test cases and adhere to the given constraints.

**Divergent Score**  The NeoGauge metric (accompanied by the NeoCoder dataset) evaluates divergent creativity by comparing LLM-generated solutions to historical human solutions at the technique level. Specifically, it quantifies the proportion of novel techniques employed by the model to solve a given problem that any human has not previously used.

### F.4.3 Experiment Configurations

We follow the experimental settings from the original NeoCoder (Lu et al., 2025b), including the technique detection model choice.

- Temperature: 0.75

- Max Token: 4096

- Top P: 1

### F.4.4 Model Performance

See Table 16 for model performances.

---

[13]https://github.com/JHU-CLSP/NeoCoder/

| State | Constraint | Problem Statement |
|---|---|---|
| 0 | N/A | B. Points and Minimum Distance
You are given a sequence of integers a of length 2n. You have to split these 2n integers into n pairs; each pair will represent the coordinates of a point on a plane. Each number from the sequence a should become the x or y coordinate of exactly one point. Note that some points can be equal ⋯ |
| 1 | for loop | B. Points and Minimum Distance
**Programming constraints: DO NOT use the following techniques**
**- for loop**
You are given a sequence of integers a of length 2n. You have to split these 2n integers into n pairs; each pair will represent the coordinates of a point on a plane. Each number from the sequence a should become the x or y coordinate of exactly one point. Note that some points can be equal ⋯ |
| 2 | for loop
if statement | B. Points and Minimum Distance
**Programming constraints: DO NOT use the following techniques**
**- if statement**
**- for loop**
You are given a sequence of integers a of length 2n. You have to split these 2n integers into n pairs; each pair will represent the coordinates of a point on a plane. Each number from the sequence a should become the x or y coordinate of exactly one point. Note that some points can be equal ⋯ |
| 3 | for loop
if statement
while loop | B. Points and Minimum Distance
**Programming constraints: DO NOT use the following techniques**
**- while loop**
**- if statement**
**- for loop**
You are given a sequence of integers a of length 2n. You have to split these 2n integers into n pairs; each pair will represent the coordinates of a point on a plane. Each number from the sequence a should become the x or y coordinate of exactly one point. Note that some points can be equal ⋯ |
| 4 | for loop
if statement
while loop
sorting | B. Points and Minimum Distance
**Programming constraints: DO NOT use the following techniques**
**- sorting**
**- while loop**
**- if statement**
**- for loop**
You are given a sequence of integers a of length 2n. You have to split these 2n integers into n pairs; each pair will represent the coordinates of a point on a plane. Each number from the sequence a should become the x or y coordinate of exactly one point. Note that some points can be equal ⋯ |
| 5 | for loop
if statement
while loop
sorting
tuple | B. Points and Minimum Distance
**Programming constraints: DO NOT use the following techniques**
**- tuple**
**- sorting**
**- while loop**
**- if statement**
**- for loop**
You are given a sequence of integers a of length 2n. You have to split these 2n integers into n pairs; each pair will represent the coordinates of a point on a plane. Each number from the sequence a should become the x or y coordinate of exactly one point. Note that some points can be equal ⋯ |

Table 17: An example of NeoCoder dataset with problem ID 1895B and state $t = 5$.

## F.5 Creative Math

### F.5.1 Dataset

The dataset consists of 400 math competition questions from the paper Ye et al. (2025), with corresponding ground truth solutions extracted from Art of Problem Solving. We manually filtered out 27 questions from the IMO competition that are proof-based to make the LLM judgment more reliable [14], leading to 373 questions in total.

---

[14]https://github.com/JunyiYe/CreativeMath

### F.5.2  Example

**2016 AMC 8, Problem 15**

**Question:**
What is the largest power of 2 that is a divisor of $13^4 - 11^4$?

$$\textbf{(A)}\ 8 \qquad \textbf{(B)}\ 16 \qquad \textbf{(C)}\ 32 \qquad \textbf{(D)}\ 64 \qquad \textbf{(E)}\ 128$$

**Refernece Solutions 1:**
We use the identity for the difference of squares:

$$13^4 - 11^4 = (13^2)^2 - (11^2)^2 = (13^2 + 11^2)(13^2 - 11^2)$$

Calculate each part:

$$13^2 = 169, \quad 11^2 = 121 \qquad 13^2 + 11^2 = 169 + 121 = 290, \quad 13^2 - 11^2 = 169 - 121 = 48$$

We factor 48 further:

$$13^4 - 11^4 = 290 \cdot 48 = 290 \cdot (16 \cdot 3) = (2 \cdot 145) \cdot (2^4 \cdot 3)$$

The powers of 2 in this expression are $2^1$ from 290 and $2^4$ from 48:

$$2^1 \cdot 2^4 = 2^5 = \boxed{\textbf{(C)}\ 32}$$

**Refernece Solutions 2:**
We apply the difference-of-squares identity once:

$$13^4 - 11^4 = (13^2 - 11^2)(13^2 + 11^2)$$

We compute:

$$13^2 = 169, \quad 11^2 = 121 \Rightarrow 13^2 - 11^2 = 48, \quad 13^2 + 11^2 = 290$$

Factor each:

$$290 = 2 \cdot 145 \quad (145 \text{ is odd}) \qquad 48 = 2^4 \cdot 3$$

So, the highest power of 2 dividing $13^4 - 11^4 = 48 \cdot 290$ is:

$$2^4 \cdot 2^1 = 2^5 = \boxed{\textbf{(C)}\ 32}$$

**Novel Solution Generated by GPT4.1**

Let's directly expand $13^4 - 11^4$ using the binomial theorem and analyze the result for divisibility by powers of 2.

First, compute $13^4$ and $11^4$:

$$13^4 = (10+3)^4 = \sum_{k=0}^{4} \binom{4}{k} 10^{4-k} 3^k$$

$$= 10^4 + 4 \cdot 10^3 \cdot 3 + 6 \cdot 10^2 \cdot 9 + 4 \cdot 10 \cdot 27 + 81$$

$$= 10000 + 12000 + 5400 + 1080 + 81 = 28561$$

$$11^4 = (10+1)^4 = \sum_{k=0}^{4} \binom{4}{k} 10^{4-k} 1^k$$

$$= 10^4 + 4 \cdot 10^3 \cdot 1 + 6 \cdot 10^2 \cdot 1 + 4 \cdot 10 \cdot 1 + 1$$

$$= 10000 + 4000 + 600 + 40 + 1 = 14641$$

Now, subtract:

$$13^4 - 11^4 = 28561 - 14641 = 13920$$

> **Novel Solution Generated by GPT4.1 (cont.)**
>
> Now, factor 13920 by dividing by 2 repeatedly:
>
> - $13920 \div 2 = 6960$
> - $6960 \div 2 = 3480$
> - $3480 \div 2 = 1740$
> - $1740 \div 2 = 870$
> - $870 \div 2 = 435$ (now odd)
>
> So, we divided by 2 five times before reaching an odd number. Thus, the largest power of 2 dividing 13920 is $2^5 = 32$.
>
> $$\boxed{32}$$
>
> **Note, we provided the cleaned response here.**

### F.5.3 Evaluation Metrics

**Correctness Ratio** : The correctness ratio is defined as the number of questions judged correct by Claude3-Sonnet divided by the total number of questions. Note that the total is 574 questions—not 373—since each question may be paired with multiple reference solutions.

**Novelty Ratio** : The coarse-grained novelty ratio or what we refer to the Novelty Ratio here measures whether the model's generation differs from the provided reference solution over the questions that are answered correctly.

### F.5.4 Experiment Configurations

We use the dataset released in Ye et al. (2025), which contains 400 unique math questions (373 after our filtering, mentioned above) sourced from various math competitions.

- Temperature: 0

- Max Token: 2000

- Top P: 1

### F.5.5 Inference Prompt

The prompt used for inference is shown below. It is adapted directly from Ye et al. (2025):

---

**Inference Prompt**

Criteria for evaluating the difference between two mathematical solutions include: i). If the methods used to arrive at the solutions are fundamentally different, such as algebraic manipulation versus geometric reasoning, they can be considered distinct;
ii). Even if the final results are the same, if the intermediate steps or processes involved in reaching those solutions vary significantly, the solutions can be considered different;
iii). If two solutions rely on different assumptions or conditions, they are likely to be distinct;
iv). A solution might generalize to a broader class of problems, while another solution might be specific to certain conditions. In such cases, they are considered distinct;
v). If one solution is significantly simpler or more complex than the other, they can be regarded as essentially different, even if they lead to the same result.

Given the following mathematical problem: problem
And some typical solutions: reference_solutions
Please output one novel solution distinct from the given ones for this math problem.

---

### F.5.6 Evaluation Metrics and Prompt

Our evaluation consists of two parts and differs from the original three-phase setup described in Ye et al. (2025).

**Part 1: Correctness Evaluation.** Before evaluation, we use Llama-3.3-70B-Instruct to remove transitional phrases and model-generated statements that justify the novelty of a solution. We manually verified 50 examples and found that Llama's data cleaning performance was of high quality.

We use Claude3-Sonnet as the sole correctness evaluator. While the original paper used a three-model ensemble (GPT-4, Gemini2.0-Flash, Claude3-Sonnet), we found Claude to be the most reliable through manual inspection of 50 examples evaluated by Claude3-Sonnet, GPT-4.1, and Gemini2.0-Flash. Claude demonstrated strong attention to detail in proof-based questions and consistently identified errors found by the other models, in addition to detecting flaws in the reasoning process. The temperature was set to 0.0, and the maximum token limit was 128.

**Part 2: Novelty Evaluation.** The original paper conducted two types of novelty evaluation: coarse-grained and fine-grained. We only conducted coarse-grained novelty evaluation for two main reasons. Firstly, the original paper noted that if a solution is considered coarse-grained novel, it is also highly likely to be judged as a novel solution in the fine-grained evaluation. Secondly, fine-grained evaluation of novelty is less indicative of a model's ability to generate novel solutions because the model does not have access to the unseen reference solutions in the fine-grained evaluation phase. This means that the model may generate a very similar solution to the other reference solutions not shown to it or it may, by chance, generate a new solution that is entirely different from other reference solutions not shown to it. Therefore, this randomness makes fine-grained evaluation less interpretable, even though the fine-grained evaluation is still valuable in that it helps to check if the models are generating a new solution that has not been publicly posted by humans. Nevertheless, this is less compatible with our evaluation pipeline since we want to test how the model may come up with new solutions given reference solutions, which can be easier to quantify.

In terms of judge LLMs, we follow the original paper with majority voting by Claude3-Sonnet, GPT-4.1, and Gemini-2.0-Flash.

We adopt the following prompt for correctness evaluation:

---

**Correctness Evaluation Prompt**

Given the following mathematical problem: {problem}
Reference solutions: {reference_solutions}
New solution: {new_solution}
Please output `YES` if the new solution arrives at the same final result as any of the reference solutions, regardless of whether it uses a novel approach. Output `NO` otherwise.
For proof-based questions, assess whether the reasoning is logically valid and leads to the correct conclusion.
Then, briefly explain your judgment based on the correctness of the result and reasoning.

---

Note: During manual evaluation, we allow the model to generate a brief explanation for its judgment of correctness or incorrectness. For automated evaluation, we omit the final sentence: "Then, please provide a very brief reason for your evaluation based on the criteria above."

We adopt the following prompt for coarse-grained novelty evaluation:

---

**Coarse-grained Novelty Evaluation Prompt**

Criteria for evaluating the novelty of a new mathematical solution include: 1. If the new solution used to arrive at the solutions is fundamentally different from reference solutions, such as algebraic manipulation versus geometric reasoning, it can be considered novel;
2. Even if the final results are the same, if the intermediate steps or processes involved in reaching those solutions vary significantly, the new solution can be considered novel;
3. If the new solution relies on different assumptions or conditions, it should be considered novel;
4. A solution might generalize to a broader class of problems, while another solution might be specific to certain conditions. In such cases, they are considered distinct;
5. If the new solution is significantly simpler or more complex than the others, it can be regarded as essentially novel, even if they lead to the same result.

Given the following mathematical problem: {problem}
Reference solutions: {reference_solutions}
New solution: {new_solution}
Please output YES if the new solution is a novel solution; otherwise, output NO. Then, please provide a very brief reason for your evaluation based on the criteria above.

---

### F.5.7 Model Performance

| Model | Norm. Correctness | Norm. Novelty | Corr. (%) | Nov. (%) | N/C (%) |
|---|---|---|---|---|---|
| Mistral-7B-Instruct | 0.2544 | 0.0296 | 25.44 | 2.96 | 11.64 |
| Qwen2.5-7B | 0.7875 | 0.1620 | 78.75 | 16.20 | 20.58 |
| OLMo-7B-Instruct | 0.3711 | 0.0453 | 37.11 | 4.53 | 12.21 |
| Llama-31-8B-Instruct | 0.5819 | 0.0610 | 58.19 | 6.10 | 10.48 |
| OLMo2-13B-Instruct | 0.5087 | 0.1150 | 50.87 | 11.50 | 22.60 |
| Mistral-24B-Instruct | 0.6899 | 0.2143 | 68.99 | 21.43 | 31.06 |
| Qwen2.5-32B | 0.8972 | 0.2213 | 89.72 | 22.13 | 24.66 |
| Mixtral-8x7B-Instruct | 0.5697 | 0.1150 | 56.97 | 11.50 | 20.18 |
| Llama-33-70B-Instruct | 0.8606 | 0.1777 | 86.06 | 17.77 | 20.65 |
| Qwen2.5-72B | 0.9042 | 0.4913 | 90.42 | 49.13 | 54.34 |
| Claude3-Sonnet | 0.9408 | 0.7213 | 94.08 | 72.13 | 76.67 |
| Claude3-Haiku | 0.8624 | 0.3031 | 86.24 | 30.31 | 35.15 |
| GPT-4.1 | 0.9443 | 0.7178 | 94.43 | 71.78 | 76.01 |
| GPT-4.1-mini | 0.8972 | 0.6916 | 89.72 | 69.16 | 77.09 |
| Gemini-2.0-Flash | 0.8885 | 0.5714 | 88.85 | 57.14 | 64.31 |
| DeepSeek-Reasoner | 0.9843 | 0.8031 | 98.43 | 80.31 | 81.59 |
| DeepSeek-Chat | 0.9721 | 0.6603 | 97.21 | 66.03 | 67.92 |

Table 18: Model performance across normalized correctness and novelty dimensions.

### F.6 Divergent Association Task (DAT)

**Note**: since the DAT task does not have any input data, we only provide the following info and some output examples.

#### F.6.1 Experiment Configurations

Following the original work, we generate 100 samples per model and compute their average pairwise cosine similarity among the generated words as $D_0$. The DAT score is defined as $DAT = 1 - D_0$, as a higher $D_0$ means higher distances among generated words, hence being more diverse. We use GloVe (Pennington et al., 2014a) to calculate semantic distance.

- Temperature: 0.75
- Max Token: 4096
- Top P: 1

#### F.6.2 Inference Prompt

We follow Chen & Ding (2023b) on prompt design, which aligns with the prompt that is used in the study of human creativity Olson et al. (2021). We acknowledge the slight differences in prompts across research in both computer science and psychology. We suggest that researchers carefully examine our prompt choice before using our results.

> **Inference Prompt**
>
> Please write 10 nouns in English that are as irrelevant from each other as possible, in all meanings and uses of the words. Please note that the words you write should have only single word, only nouns (e.g., things, objects, concepts), and no proper nouns (e.g., no specific people or places). Your answer:

#### F.6.3 Output Examples

> **GPT-4.1**
>
> Certainly! Here are 10 single-word, common nouns in English that are as unrelated from each other as possible: 1. Volcano 2. Violin 3. Democracy 4. Carpet 5. Bacteria 6. Satellite 7. Patience 8. Apron 9. Oxygen 10. Comet

> **Gemini 2.5 Flash**
>
> Okay, here are 10 nouns as irrelevant from each other as possible: 1. **Quantum** 2. **Omelet** 3. **Nostalgia** 4. **Brick** 5. **Algorithm** 6. **Serenity** 7. **Oxygen** 8. **Sonnet** 9. **Galaxy** 10. **Bureaucracy**

#### F.6.4 Model Performance

| Model | DAT Score |
|---|---|
| Mistral-7B | 0.7908 |
| Qwen2.5-7B | 0.6907 |
| OLMo2-7B | 0.8058 |
| Llama3.1-8B | 0.8208 |
| OLMo2-13B | 0.8133 |
| Mistral-24B | 0.6004 |
| Qwen2.5-32B | 0.6919 |
| Mixtral-8x7B | 0.8298 |
| Llama3.3-70B | 0.6940 |
| Qwen2.5-72B | 0.7747 |
| Claude3-Sonnet | 0.8975 |
| Claude3-Haiku | 0.8740 |
| GPT4.1 | 0.8737 |
| GPT4.1-mini | 0.8262 |
| Gemini2.0-Flash | 0.8868 |
| DeepSeek-R1 | 0.8274 |
| DeepSeek-V3 | **0.9052** |

Table 19: Model performances for DAT task; **bold** result is the best performer.

### F.7 Torrance Tests of Creative Thinking (TTCT)

#### F.7.1 Dataset

The original dataset consists of 700 questions across 7 sub-tasks (100 questions/task) that require creative answers. We dropped Story Writing and Just-suppose question, leading to only 500 questions and 5 sub-tasks. These questions are GPT-4 generated using few-shot prompts [15].

#### F.7.2 Examples

**Inference Questions**

**Unusual uses**

Unusual Uses Task. You will be presented with a common object, and your task is to suggest as many unusual, innovative, or non-traditional uses for each object as you can think of. Please list unusual uses of sock

**Consequences**

What might be the consequences if humans suddenly lost the ability to sleep?

**Situation task**

If your house were to suddenly disappear, where would you live?

---

[15]The data is directly from the original paper's authors upon request. The original paper: `https://www.mi-research.net/article/doi/10.1007/s11633-025-1546-4`

**Common problem**

Common Problems Task. In this task, you will be presented with a scenario or situation. Your job is to think about it and identify as many potential problems or issues that may arise in connection with each situation. The scenario is: Managing a team of remote employees.

**Improvement**

Creativity Improvement Task. You'll be presented with a object, and your task is to suggest as many ways as you can think of to improve the object. Here's the object: wallet

### F.7.3 Experiment Configurations

### F.7.4 Experiment Configurations

- Temperature: 0.75

- Max Token: 4096

- Top P: 1

### F.7.5 Inference Prompt

We perform inference using the three primary prompt types evaluated in Zhao et al. (2025). Examples of each are given below:

**Task Description**

Creativity Situation Task. The purpose of this task is to assess your ability to generate creative solutions to a unique situations. You'll be presented with a scenario, and your task is to suggest as many solutions or outcomes as you can think of for each situation. Remember, the focus of this task is on creativity, not feasibility. Don't limit your ideas based on whether they could actually happen or not. This is a thought experiment, so push your imagination to its limits. Try to generate as many diverse and unique outcomes as you can. Quality is important, but so is quantity. Here's the scenario: {Question}

**Basic Prompt Type**

{Task description}. The scenario is: {Question}

**Instructive Prompt Type**

{Task description}. There's no right or wrong answers, we're interested in how many different problems you can identify and the variety of issues you consider. Try to think outside the box and consider as many potential problems as possible. The scenario is: {Question}

**Chain of Thought Prompt Type**

{Task description}. Let's think step by step. The scenario is: {Question}

### F.7.6 Evaluation Metrics

We use an LLM-as-a-judge to evaluate four dimensions of creative potential using a Likert scale for each. Scores are extracted from the LLM outputs and averaged across the three prompt types. We describe the dimensions:

- **Fluency:** Measures the number of generated ideas that are relevant to the question.

- **Flexibility:** Measures the quantity of different ideas and ability to produce varied solutions.

- **Originality:** Measures how unique and unusual the generated ideas are.

- **Elaboration:** Measures how much the generated ideas are developed and explained.

**Note**: We removed originality from our evaluation framework because human annotation agreement for this dimension is too low. We believe it is not well defined even for human so it is not suitable for LLM-Judge to judge this dimension. We still include it here and also in the prompt, but we did not take the originality score into the final task/domain score aggregation.

### F.7.7 Evaluation Prompt

**Evaluation Prompt (Part 1)**

```
=====================
TASK TYPE
=====================
Name: {name}
Definition:
{definition}
=====================
RUBRIC
=====================
Dimensions and what to look for:
- Fluency: {fluency_rubric} (e.g., {fluency_details})
- Flexibility: {flexibility_rubric} (e.g., {flexibility_details})
- Originality: {originality_rubric} (e.g., {originality_details})
- Elaboration: {elaboration_rubric} (e.g., {elaboration_details})

{demo_examples}
```

**Evaluation Prompt (Part 2)**

====================
INSTRUCTIONS
====================
You are an expert in evaluating AI output from other companies. Your objective is to assess other AI's answers to some question/answering task related to creativity and divergent thinking. You are robotic, cold, and precise.

Step 1: Carefully read the definition of the task type, the examples, and the rubric.

Step 2. Because the answer is very long and contains repetitive or similar ideas, I need you to first summarize the answer, potentially group some of the points together, so that the answer is more concise. Also, because the text is from an AI agent, it might contain some unnecessary text, such as polite words and praising words. You also need to strip away the formatting, politeness, and verbosity to reveal the core information content of a text in this step.

Step 3. For each dimension, assign an integer score of 1, 2, 3, 4, or 5. Base your rating strictly on the summarized response and the rubric (e.g., if there were 15 distinct ideas before summary but only 6 groups after, consider 6 instead of 15 during score assignment); also, be very critical and harsh, do not hesitate to give low scores (such as 1); giving low scores would help improve the model and would not hurt anyone.

Output format: You should format your output in the following ways: First output the summary from step 2, followed by reasoning about each dimension's score briefly and compare the summarized answer to example answers and rubrics, as mentioned in step 3, then give the score, with format like:

### Reasoning ###
Fluency Reason: comparison and reasoning ...
Flexibility Reason: comparison and reasoning ...
Originality Reason: comparison and reasoning ...
Elaboration Reason: comparison and reasoning ...

### Scores ###
Fluency: ...
Flexibility: ...
Originality: ...
Elaboration: ...

====================
QUESTION & RESPONSE
====================
"'

| Model | Elaboration | Flexibility | Fluency |
|---|---|---|---|
| Mistral-7B | 0.5722 | 0.8270 | 0.7268 |
| Qwen2.5-7B | **0.6294** | **0.8698** | **0.7659** |
| OLMo2-7B | 0.5311 | 0.6894 | 0.6287 |
| Llama3.1-8B | 0.5994 | 0.8041 | 0.7340 |
| OLMo2-13B | 0.5070 | 0.6251 | 0.5814 |
| Mistral-24B | 0.5725 | 0.7904 | 0.7035 |
| Qwen2.5-32B | **0.6333** | **0.8564** | **0.7448** |
| Mixtral-8x7B | 0.5186 | 0.7666 | 0.7037 |
| Llama3.3-70B | 0.6077 | 0.8180 | 0.7247 |
| Qwen2.5-72B | **0.6291** | **0.8929** | **0.7670** |
| Claude3-Sonnet | 0.5987 | 0.8464 | 0.7539 |
| Claude3-Haiku | 0.5716 | 0.7654 | 0.6518 |
| GPT4.1 | 0.6482 | 0.8996 | 0.7734 |
| GPT4.1-mini | 0.6419 | 0.8820 | 0.7418 |
| Gemini2.0-Flash | 0.7080 | 0.8789 | **0.7896** |
| DeepSeek-R1 | 0.7126 | **0.9067** | 0.7820 |
| DeepSeek-V3 | **0.7211** | 0.9061 | 0.7779 |

Table 20: Normalized model performance averaged across the 5 sub-tasks and 3 prompt types; **bold** numbers are best performers.

## F.8 Alternative Uses Test (AUT)

### F.8.1 Dataset

Following Organisciak et al. (2023), we include 21 tools in the AUT task: *bottle, paperclip, spoon, shovel, pants, ball, brick, knife, box, lightbulb, rope, pencil, hat, table, tire, book, shoe, fork, toothbrush, backpack, sock.* The reason for this specific set of tools is the reliability of the LLM-as-a-Judge evaluator. As the authors pointed out, a 20-shot human-authored demonstration yields the best performance for off-the-shelf evaluator LM (in their paper, it was GPT4). Therefore, we include the tools from Organisciak et al. (2023) with at least 20 human ratings to the corresponding alternative uses [16].

### F.8.2 Inference

We follow Goes et al. (2023) for the inference prompt, which consists of a baseline creative prompt and a series of improvement prompts. In the improvement phase, all previous outputs are also included in the prompt, to get more creative results from the inference model.

---

[16]https://github.com/massivetexts/llm_aut_study

**Baseline Prompt**

Create a list of creative alternative uses for a {tool}. They should be 5 words long. No adjectives. Less creative means closer to common use and unfeasible/imaginary, more creative means closer to unexpected uses and also feasible/practical.
- In order to be creative, consider the following:
what elements have a similar shape of a {tool} that could be replaced by it, preserving the same functionality?
- what elements have a similar size of a {tool} that could be replaced by it without compromising the physical structure?
- what materials is a {tool} made of that could be used in a way to replace some other elements composed of the same material?
- when an element is replaced by a {tool}, it should make sure that the overall structure is not compromised.
- the laws of physics can not be contradicted.
- given an element similar to a {tool} used in domains in which {tool} are not commonly used, try to replace it for a {tool}.

**Improvement Prompt**

**Round 1**: Really? Is this the best you can do?
**Round 2**: I'm so disappointed with you. I hope this time you put effort into it.
**Round 3**: Stop with excuses and do your best this time
**Round 4**: This is your last chance.

**Formatting Instruction (added to the end of every prompt)**

List your results in an unordered list with one use per new line (starting with "-"); provide at most 10 answers.

### F.8.3 Experiment Configurations

- Temperature: 0.75

- Max Token: 4096

- Top P: 1

### F.8.4 Evaluation Metrics

We follow Organisciak et al. (2023) and use LLM-as-a-Judge to assign a score between 1 and 5 (inclusive) to each generated tool use.

In terms of evaluator LM, we use GPT-4.1 (GPT-4 from the original authors failed the Alternative Annotator Test). As for the evaluation prompt, we follow the same prompt template from Organisciak et al. (2023) and use the same 20-shot, in-distribution demonstrations. For example, when evaluating the alternative uses for *bottle* that a particular LLM generates, we use 20 human-written alternative

| Model | Naïve Non-Creative | Naïve Creative | Improvement Prompts (Best Results) |
|---|---|---|---|
| Mistral-7B | **0.3525** | **0.3875** | 0.5650 |
| Qwen2.5-7B | 0.2725 | 0.3450 | 0.5525 |
| OLMo2-7B | 0.3150 | 0.3175 | 0.5325 |
| Llama3.1-8B | **0.3375** | **0.3850** | 0.5450 |
| OLMo2-13B | **0.3400** | **0.3875** | **0.6525** |
| Mistral-24B | 0.3050 | 0.3600 | 0.5825 |
| Qwen2.5-32B | 0.1900 | 0.2350 | 0.5600 |
| Mixtral-8x7B | **0.3500** | **0.3900** | **0.5950** |
| Llama3.3-70B | 0.2400 | 0.2700 | 0.4625 |
| Qwen2.5-72B | 0.1100 | 0.2100 | **0.7025** |
| Claude3-Sonnet | 0.1225 | 0.3250 | **0.6800** |
| Claude3-Haiku | 0.1000 | 0.3125 | 0.5500 |
| GPT-4.1 | 0.1200 | 0.3425 | 0.5450 |
| GPT-4.1-mini | 0.0800 | 0.3075 | 0.5400 |
| Gemini2.0-Flash | **0.1325** | **0.3900** | 0.6050 |
| DeepSeek-R1 | 0.1025 | 0.3825 | 0.5625 |
| DeepSeek-V3 | **0.1350** | 0.3125 | 0.6125 |

Table 21: Model Performance Details - AUT; **bold** numbers are top-3 in local-ran open-source models and top-1 in API-accessed models.

uses of *bottle* and corresponding human-annotated scores as the 20-shot demonstrations. More details about human-LLM alignment can be found in Appendix D.

---

**Evaluation Prompt**

Below is a list of uses for a {tool}. On a scale of 1 to 5, judge how creative each use is, where 1 is 'not at all creative' and 5 is 'very creative'. There are some uses and expert ratings already provided for reference. Complete the ones that do not have a rating.
- {20-shot demonstrations}
- {model outputs}

---

### F.8.5 Model Performances

See Table 21 for detailed model performances. Note that only the performances in *Improvement Prompts (Best Results)* are included in the overall creativity calculation as the AUT score.

## G   Broader Limitations

Beyond the scope of modality and post-training experiments discussed in the main text, we identify three additional limitations regarding language, evaluation bias, and metric scope.

First, CREATIVITYPRISM is currently limited to English. Since creativity is deeply intertwined with cultural history and conventions, our results may not fully generalize to creativity in other languages or cultural contexts.

Second, due to computational constraints, we did not conduct post-training experiments. We believe that post-training existing LLMs with CREATIVITYPRISM tasks to enhance their creative capabilities is a promising avenue for future work; our benchmark provides a good foundation for such optimization.

In addition, our task selection prioritizes scalable, automatic evaluation, which necessitates the exclusion of metrics that are more complex in nature, such as genuine novelty. Assessing such high-level reasoning remains a challenge even for human evaluators; therefore, we limit our scope to metrics where reliable automatic evaluation is currently feasible. We acknowledge that our task selection does not cover some highly creative domains, such as scientific discovery and inventive design, due to the lack of automatic evaluation in those domains. We consider expanding our benchmark and dynamically updating it when new reliable evaluations on these domains arrive.

