# OpenReview forum: "CreativityPrism: A Cross-Domain Evaluation Framework for Large Language Model Creativity"
_TMLR — Accepted by TMLR_

### Review · Reviewer_tygV · 2026-03-15

**Summary Of Contributions:**

**Summary**

This paper proposes CreativityPrism, an evaluation framework for assessing the creativity of large language models. The framework integrates eight tasks spanning divergent thinking, creative writing, and logical reasoning, and organizes evaluation metrics into three dimensions: quality, novelty, and diversity. The authors also introduce an evaluation pipeline that combines automatic metrics and LLM-based judges, with additional human annotation used to validate judge reliability. Using this framework, the paper benchmarks 17 state-of-the-art LLMs and analyzes performance differences across creativity domains and dimensions.


**Strengths**

• The paper studies an increasingly relevant topic, namely how to evaluate creativity in large language models in a systematic and scalable way.

• The proposed framework integrates multiple creativity-related tasks from different domains, which helps provide a more comprehensive view of model capabilities.

• The empirical study includes a relatively broad set of models, covering both open-source and proprietary systems.

• The authors attempt to validate LLM-based judges using human annotations and agreement analysis, which improves confidence in the automatic evaluation procedure.

• The paper is generally clear and the experimental setup is well described.


**Weaknesses**

• The main contribution of the work appears to be the integration of existing creativity benchmarks into a unified framework. While this integration is useful, the paper could more clearly articulate what conceptual or methodological advances the framework introduces beyond aggregation.

• The proposed three-dimensional taxonomy of creativity (quality, novelty, diversity) is largely inspired by prior literature, and the paper could further clarify why these dimensions sufficiently capture creativity in LLM outputs.

• The framework currently focuses only on text-based tasks, which somewhat limits the scope of the claim that the evaluation is holistic.

• The aggregation procedure used to compute the overall creativity score may influence model rankings. Additional discussion or analysis of the robustness of this aggregation would strengthen the work.

**Audience:**

Yes

**Audience Explanation:**

The paper studies the evaluation of creativity in large language models, which is an increasingly relevant topic for the machine learning community. Researchers working on language models, evaluation methodologies, and AI creativity would likely find the proposed framework and benchmark results informative.

**Broader Impact Concerns:**

I do not identify significant ethical concerns associated with this work. The paper proposes an evaluation framework for measuring creativity in language models and does not introduce new model capabilities or datasets that raise immediate societal or ethical risks beyond those already associated with large language models in general.

**Claims And Evidence:**

Yes

**Claims Explanation:**

The paper provides empirical evidence through experiments on multiple creativity tasks and evaluates a broad set of state-of-the-art models. The experimental setup, metrics, and evaluation procedures are clearly described, and the main claims are generally supported by the reported results and analyses.

**Requested Changes:**

**Critical for acceptance**

• The paper would benefit from a clearer discussion of the methodological novelty of the framework. Since many of the tasks and metrics are adopted from existing work, it would be helpful for the authors to more explicitly clarify what aspects of CreativityPrism go beyond aggregating existing benchmarks and how the framework advances the evaluation of LLM creativity.

• The justification for the three-dimensional taxonomy (quality, novelty, diversity) could be strengthened. While these dimensions are motivated by prior creativity literature, the paper could better explain why these dimensions are sufficient to capture creativity in LLM outputs and how alternative formulations might affect the evaluation.

• The aggregation procedure used to compute the overall creativity score should be further discussed. In particular, the choice of averaging normalized metrics across tasks and dimensions may influence the final ranking of models, so additional justification or sensitivity analysis would help clarify the robustness of the results.

**Suggestions to strengthen the work**

• The paper could further discuss how the proposed framework compares with evaluating individual creativity benchmarks independently. For example, it would be useful to clarify what new insights CreativityPrism provides that are not observable from existing task-specific benchmarks.

• The scope of the framework is currently limited to text-based tasks. The authors may consider discussing more concretely how the framework could be extended to multimodal creativity evaluation in future work.

• Some additional analysis of the benchmark results could strengthen the empirical contribution. For instance, the paper could further investigate why novelty metrics show weaker correlations with other dimensions and what this implies for training or evaluating creative models.

• Providing clearer guidance on how practitioners should interpret or use the aggregated creativity scores in different applications could improve the practical usefulness of the framework.

---

> ### Author Response · Authors · 2026-04-06
> **Requested Changes [1, 4] Methodological novelty of the framework**
>
> We sincerely thank reviewer tygV for the supportive and very constructive review of our evaluation framework! We will present the following response to the requested changes and weaknesses that the reviewer listed.
>
> ### 1. Methodological novelty of the framework; 4. Compared with individual creativity benchmark
>
> Thanks for raising this concern! We believe our work, although it seems to be a “simple combination” of existing benchmarks, has the following novel and valuable contribution. We will also add them to the corresponding sections of the paper to specify the significance of our evaluation framework
> - **LLM-Judge verification**:
>   - Although an increasing number of benchmarks nowadays adopt the LLM-as-a-Judge paradigm, very few of them rigorously verify the alignment between their LLM-Judge and human judgment and are transparent about their verification process (e.g., releasing their human annotation data), especially in creativity evaluation, where expensive, high-quality human annotation drives more researchers to LLM-Judge evaluation. As mentioned in the paper, 5 out of 8 tasks we collected require LLM-Judge, but 3 out of 5 (TTCT, NeoCoder, CreativeMath) do not even have an annotated judgment released, and one of the remaining two (TTCW AUT) has an LLM-Judge that does not align well enough with human judgment.
>    - Aside from human-annotated data availability, our initial experiment with the TTCT task shows LLM-Judge failing in at least 2 cases: (1) length bias, where judges tend to assign higher scores to verbose responses, which is particularly pronounced for models that generate preambles or closing remarks (e.g., greeting messages, expressions of politeness); (2) redundancy blindness, where judges fail to penalize responses that repeat the same core idea with superficial variation. **We mitigate both by preprocessing model outputs to strip formatting, politeness markers, and verbosity before evaluation (see prompt in Appendix E.7.6).**
> In summary, our effort in standardizing annotation protocols and LLM-Judge alignment verification can serve as a reference for future research in machine creativity and should be considered as an important contribution.
>
> - **Open-source Code**: We also have a concrete plan to release, maintain, and integrate more tasks in LLM (and MMLM) creativity evaluation by building a protocol for new task integration. Our open-sourced data cleaning and task integration pipeline would foster community contributions to creativity evaluation. We also have a website under construction to visualize the results, inform the community the latest changes (new task, new models, etc.), and enable easy task submission.
> - **Models included**: Existing work only focuses on a single domain and studies a limited number of models. Our expansion to cross-domain and cross-dimension benchmarking enables comparison of the SoTA LLM across different task domains, which, as mentioned in the analysis section, shows low correlation among tasks from different domains. This indicates that good performance in one domain does not transfer to others, further motivating such cross-domain analysis.

---

> ### Author Response · Authors · 2026-04-06
> **Requested Changes [2] Justification for the three-dimensional taxonomy**
>
> ### 2. Justification for the three-dimensional taxonomy
>
> Thanks for pointing this out! We acknowledge that we need more theoretical justification for the creativity taxonomy. We will make the following changes to Section 3 - “Three Dimensions of Creativity”:
>
> - **Grounding in existing literatures**: “Our taxonomy is grounded in the most recent version of the TTCT verbal test, which operationalizes creativity along three dimensions: fluency, flexibility, and originality.\footnote{While the original TTCT [1] includes a fourth dimension, elaboration, the current version of the TTCT verbal test as administered by major testing agencies [2,5] consists of three dimensions only, a simplification also reflected in recent psychology literature [3].} These map directly onto our dimensions: fluency corresponds to quality (the ability to produce coherent, well-formed responses), flexibility to diversity (the ability to vary approaches and perspectives), and originality to novelty (the ability to produce responses that deviate from the commonplace) [2]. We are also different from the widely-cited binary taxonomy of usefulness and originality [4] by separating originality into novelty and diversity. We argue this distinction is particularly important for LLM evaluation: diversity captures breadth, i.e., how much a model varies its outputs across responses or within a single generation, while novelty captures depth, i.e., how much a single output deviates from existing human-generated content or conventional solutions (as visualized in figures above). A model can score highly on one while failing on the other (e.g., producing many distinct but individually unoriginal responses, or producing one highly novel response with no variation across prompts), and collapsing them into a single originality score would obscure this. ”
>
> - **Visualized presentation**: We also refer to a visualized framing, [click this to view image](https://ibb.co/QvkMXkmZ), that motivates our choice of dimensions. Consider three spaces: the LLM's output distribution (green), the valid answer space (black), and the training data (blue). Each dimension captures a distinct geometric relationship among these three spaces. These three relationships are geometrically independent: a model can expand its output distribution along one axis without moving along the others. This independence is what motivates treating them as separate dimensions rather than collapsing them.
>    - *Quality* measures how much of the LLM's output distribution falls within the valid answer space, i.e., how well the model produces coherent, appropriate responses.
>    - *Diversity* measures the breadth of the LLM's output distribution within the valid answer space, i.e., how widely the model explores different valid responses across generations.
>    - *Novelty* measures how far the LLM's output distribution extends beyond the training data, i.e., how much the model generates content that deviates from what it has been exposed to.
>
> - **Alternative formulation**: One obvious alternative formulation is the earlier cited “usefulness and originality”. However, merging novelty and diversity into a single originality score obscures meaningful per-model differences. OLMo2-7B ranks 7th on diversity but 16th on novelty (a 9-position gap, with an absolute score difference of .3), indicating a model that generates varied outputs but rarely departs from conventional content. Qwen2.5-32B shows a similar pattern (11th vs. 17th, gap of 6, score difference .269). The reverse also occurs: Claude3-Sonnet ranks 2nd on diversity, but 5th on novelty, and DeepSeek-R1 ranks 1st on diversity but 4th on novelty. These dissociations, i.e., high diversity with low novelty, or vice versa, correspond to qualitatively different creative behaviors that a binary originality score would average away, and are relevant for both model evaluation and development.
>
> [1] TorranceE. P. (1966). Torrance tests of creative thinking: Norms technical manual (Research Edition).Princeton: Personnel Press
>
> [2] Torrance, E. Paul. n.d. “Torrance Tests of Creative Thinking Interpretive Manual.” Bensenville: Scholastic Testing Service. https://www.ststesting.com/gift/TTCT_InterpMOD.2018.pdf
>
> [3] Alabbasi, Ahmed M Abdulla et al. “What do educators need to know about the Torrance Tests of Creative Thinking: A comprehensive review.” Frontiers in psychology vol. 13 1000385. 26 Oct. 2022, doi:10.3389/fpsyg.2022.1000385
>
> [4] Runco, Mark A., and Garrett J. Jaeger. "The standard definition of creativity." Creativity research journal 24.1 (2012): 92-96.
>
> [5] “Torrance Tests of Creative Thinking (TTCT) Training - Mary Frances Early College of Education.” n.d. Accessed April 1, 2026. https://coe.uga.edu/outreach/programs/ttct/.

---

> ### Author Response · Authors · 2026-04-06
> **Requested Changes [3] Aggregation procedure**
>
> ### 3. The aggregation procedure used to compute the overall creativity score should be further discussed.
>
> Thanks for pointing out this very crucial detail! We tried three other methods of score aggregation (after metric normalization), and we will add the following to the appendix of the camera-ready version.
>
> We compare four aggregation methods to assess whether model rankings are robust to the choice of aggregation procedure:
> - **M0 (Used in CreativityPrism)**: normalize metrics -> average within task -> average tasks within domain/dimension (task-equal score weighting)
> - **M1**: normalize metrics -> average all metrics directly within domain/dimension
>   (metric-equal score weighting)
> - **M2**: convert each metric to a rank (1 = best) -> average ranks directly within
>   domain/dimension (metric-equal rank aggregation)
> - **M3**: convert each metric to a rank -> average ranks within task -> average task
>   ranks within domain/dimension (task-equal rank aggregation)
>
> Rankings are computed on the mean score/rank across the three domains (or dimensions), i.e., the "Overall" column in Table 3, where for M0/M1, a higher score = better rank, and for M2/M3, a lower average rank = better. “Max swing” refers to the maximum ranking change among the four aggregating methods.
>
> ---
>
> #### By Domain
>
> | Model | M0 original | M1 metric-eq score | M2 metric-eq rank | M3 task-eq rank | Max swing |
> |---|:---:|:---:|:---:|:---:|:---:|
> | DeepSeek-V3    |  1 |  1 |  1 |  1 | 0 |
> | GPT4.1         |  2 |  3 |  3 |  3 | 1 |
> | DeepSeek-R1    |  3 |  2 |  2 |  2 | 1 |
> | Claude3-Sonnet |  4 |  4 |  5 |  5 | 1 |
> | Gemini2.0-Flash|  5 |  6 |  4 |  4 | 2 |
> | GPT4.1-mini    |  6 |  5 |  6 |  6 | 1 |
> | Claude3-Haiku  |  7 |  8 |  8 |  7 | 1 |
> | Qwen2.5-72B    |  8 |  7 |  7 |  8 | 1 |
> | Mistral-24B    |  9 | 10 | 13 | 11 | **4** |
> | Llama3.3-70B   | 10 |  9 | 10 | 13 | **4** |
> | Mixtral-8x7B   | 11 | 11 | 14 | 12 | 3 |
> | OLMo2-13B      | 12 | 12 | 11 |  9 | 3 |
> | Qwen2.5-32B    | 13 | 14 |  9 | 10 | **5** |
> | Llama3.1-8B    | 14 | 16 | 15 | 14 | 2 |
> | Qwen2.5-7B     | 15 | 13 | 12 | 14 | 3 |
> | Mistral-7B     | 16 | 17 | 16 | 16 | 1 |
> | OLMo2-7B       | 17 | 15 | 17 | 17 | 2 |
>
> **Spearman $\rho$ between methods (domain-level overall rank):**
>
> | | M0 | M1 | M2 | M3 |
> |---|:---:|:---:|:---:|:---:|
> | M0 | — | .973 | .929 | .955 |
> | M1 | | — | .929 | .926 |
> | M2 | | | — | .963 |
> | M3 | | | | — |
>
> ---
>
> #### By Dimension
>
> | Model | M0 original | M1 metric-eq score | M2 metric-eq rank | M3 task-eq rank | Max swing |
> |---|:---:|:---:|:---:|:---:|:---:|
> | DeepSeek-V3    |  1 |  1 |  1 |  1 | 0 |
> | DeepSeek-R1    |  2 |  2 |  2 |  2 | 0 |
> | GPT4.1         |  3 |  3 |  3 |  3 | 0 |
> | Claude3-Sonnet |  4 |  4 |  5 |  5 | 1 |
> | Gemini2.0-Flash|  5 |  6 |  4 |  4 | 2 |
> | GPT4.1-mini    |  6 |  5 |  6 |  6 | 1 |
> | Qwen2.5-72B    |  7 |  7 |  7 |  8 | 1 |
> | Claude3-Haiku  |  8 |  8 |  8 |  7 | 1 |
> | OLMo2-7B       |  9 |  9 | 12 | 13 | **4** |
> | OLMo2-13B      | 10 | 10 | 10 | 10 | 0 |
> | Llama3.3-70B   | 11 | 11 | 11 | 11 | 0 |
> | Mistral-24B    | 12 | 13 | 14 | 12 | 2 |
> | Qwen2.5-32B    | 13 | 12 |  9 |  9 | **4** |
> | Mixtral-8x7B   | 14 | 14 | 15 | 14 | 1 |
> | Mistral-7B     | 15 | 15 | 17 | 16 | 2 |
> | Llama3.1-8B    | 16 | 16 | 16 | 17 | 1 |
> | Qwen2.5-7B     | 17 | 17 | 13 | 15 | **4** |
>
> **Spearman $\rho$ between methods (dimension-level overall rank):**
>
> | | M0 | M1 | M2 | M3 |
> |---|:---:|:---:|:---:|:---:|
> | M0 | — | .995 | .936 | .949 |
> | M1 | | — | .944 | .951 |
> | M2 | | | — | .983 |
> | M3 | | | | — |
>
> ---
>
> ####  Summary
>
> All four methods produce highly consistent rankings, with Spearman $\rho \geq .93$
> across all method pairs for both domain- and dimension-level aggregation. The top 8
> models (all proprietary plus Qwen2.5-72B), and the bottom 2 models are stable across
> all methods. The small number of rank swings (max 5 positions, confined to mid-tier
> open-source models) reflects genuine score proximity in that range rather than
> sensitivity to aggregation choice. These results confirm that the main findings of
> CreativityPrism are robust to the choice of aggregation procedure.

---

> ### Author Response · Authors · 2026-04-06
> **Requested Changes [5] The scope of the framework is currently limited to text-based tasks.**
>
> ### 5. The scope of the framework is currently limited to text-based tasks.
>
> - Many existing multimodal creativity benchmarks have a similar structure as we do, i.e., having metrics spanning quality (e.g., convergent in [1], usefulness and manufacturability in [2]), novelty (e.g., originality in [2]), and diversity (e.g., divergent in [1], [2]). These metrics can be easily integrated into our taxonomy.
> - We also acknowledge that our task selection does not cover some highly creative domains, such as scientific discovery and inventive design, which is mainly because of the lack of automatic evaluation in those domains. We will reframe our paper by changing all the “holistic” and “unified” to “cross-domain” or “multi-domain” (whichever is more appropriate to the context). We will also add this to the limitation section
> - We also have a concrete plan to release our modularized codebase. We plan to release the full evaluation pipeline, task wrappers, prompts, preprocessing scripts, LLM-Judge configurations, and all the human annotations upon publication. We will also establish a protocol for adding new tasks, including but not limited to: task-related data structure, inference and evaluation code interface (e.g., what kinds of functions they need to have), sample inference and evaluation scripts, and developer contact for people to submit their tasks. We also have a leaderboard website under construction to publicly show benchmark results. We believe that all of these combined will promote community contribution to a truly comprehensive and up-to-date creativity assessment.
>
> [1] He, Zicong, Boxuan Zhang, Weihao Liu, Ruixiang Tang, and Lu Cheng. 2025. “What Shapes a Creative Machine Mind? Comprehensively Benchmarking Creativity in Foundation Models.” arXiv [Cs.AI]. arXiv. https://doi.org/10.48550/arXiv.2510.04009.
>
> [2] Xue, K., Li, C., Ou, Z., Zhang, G., Lu, K., Lyu, S., Zhu, Y., Zong, P., Ding, J., Liu, X., Chen, Q., Qin, W., Shen, Y., & Cen, J. (2026). CreBench: Human-Aligned Creativity Evaluation from Idea to Process to Product. Proceedings of the AAAI Conference on Artificial Intelligence, 40(32), 27441-27449. https://doi.org/10.1609/aaai.v40i32.39962

---

> ### Author Response · Authors · 2026-04-06
> **Requested Changes [6] Additional Analysis on low/weak correlations among Novelty metrics**
>
> ### 6. Additional Analysis on low/weak correlations among Novelty metrics
> We thank the reviewers for raising this point! Here are some changes we will add to the corresponding sections:
>
> - **Choice of Three Dimensions** First, we clarify that our three dimensions are not derived from correlations among metrics; instead, they are defined by the goal each metric is designed to measure. A weak inter-metric correlation within a dimension, therefore, does not indicate a poorly constructed dimension; it indicates that different metrics operationalize the same high-level goal through different mechanisms, each with its own strengths and limitations. This is especially pronounced for novelty, which is most difficult to measure because it requires access to a reference corpus or solution set that is often incomplete, private, or domain-specific. For example, the Creativity Index is sensitive to the unique private training data each model has seen and to each model's paraphrasing ability; NeoCoder Divergence@0 is significantly affected by a model's constraint-following ability, which we will discuss more below. The weak correlation among these novelty metrics is therefore expected and should be considered as a contribution rather than a weakness, since they all have limitations and can be complementary to each other. Only when putting together can we evaluate the full landscape of novelty/diversity/quality of each LLM. [We will also add this discussion to Section 3 and the Appendix]
>
> - **Negative correlations between NeoCoder Divergence@0 and other metrics**. NeoCoder Divergence@0 measures the proportion of coding techniques in a model's solution absent from the human solution set [3]. A model that fails to solve a problem correctly will, by definition, avoid using standard human techniques, not because it is creative, but because its incorrect output lacks the structural properties of valid solutions. Nonsensical outputs would score perfectly on this metric. This is distinct from other LLM-judged novelty metrics, which score the quality of a particular output rather than performing a binary classification against a reference set. The NeoCoder paper itself addresses this by introducing denial prompting to isolate genuine divergence from failure; our use of the @0 baseline captures the unconditional setting where this confound is most visible. Notably, CreativeMath avoids this issue by only counting a solution as novel after it is verified correct. We will add this discussion to the paper as an illustration of why cross-task novelty correlations should be interpreted with care.
>
> - **Novelty requires external references**. More broadly, novelty is the only dimension that inherently asks "novel compared to what?", and the answer differs across tasks: compared to prior human-written corpora (L-Uniqueness, C-Short Surprises), to known human solutions (NeoCoder Divergence@0, C-Math Novelty), or to common conceptual associations (AUT Score). Quality asks "is this output well-formed and coherent?", which is a property intrinsic to the output. Diversity asks "how much does this output vary from other outputs?", which is measurable within the model's own generation distribution. This reference heterogeneity means novelty metrics measure a family of related but operationally distinct notions of deviation from a norm rather than a single latent construct. A model scoring high on one novelty metric may score low on another for interpretable reasons: AUT Score reflects unusualness of free associations in a single pass, while C-Math Novelty demands solutions that are both correct and absent from the human solution set, which are two very different demands. Therefore, as mentioned earlier, having weak/negative correlation among performances in novelty metrics is not a weakness but a strength of our proposed framework. Since all the metrics have their limitations, our framework unifies them together; our result also further validates that none of them on its own is well-representative for novelty. Hence, we recommend testing them all using our benchmark to reveal the full landscape of creativity of the target LLM being evaluated.
>
> [3] Yining Lu, Dixuan Wang, Tianjian Li, Dongwei Jiang, Sanjeev Khudanpur, Meng Jiang, and Daniel Khashabi. 2025. Benchmarking Language Model Creativity: A Case Study on Code Generation. In Proceedings of the 2025 Conference of the Nations of the Americas Chapter of the Association for Computational Linguistics: Human Language Technologies (Volume 1: Long Papers), pages 2776–2794, Albuquerque, New Mexico. Association for Computational Linguistics.

---

> ### Author Response · Authors · 2026-04-06
> **Requested Changes [7] clearer guidance for practitioners**
>
> Thanks for pointing this out! We advise LLM developers or researchers to 1) check the task description and example, 2) select the dimension and task domain that is closely related to their topic of interest, and 3) only use the model performance on that subset (domain & dimension) as a reference for result interpretation. [We will also add this part to the limitation section]

---

### Review · Reviewer_ocsC · 2026-03-23

**Summary Of Contributions:**

The paper introduces CreativityPrism, an evaluation framework for assessing creativity in large language models (LLMs). The framework unifies 8 tasks across 3 domains—divergent thinking, creative writing, and logical reasoning—and organizes evaluation into three core dimensions: quality, novelty, and diversity.

The contributions are:
- Proposes a taxonomy of creativity (quality, novelty, diversity) and maps 17 metrics into these dimensions.
- Designs a scalable evaluation pipeline, leveraging LLM-as-a-judge, validated via human annotations and the Alternative Annotator Test.
- Benchmarks 17 state-of-the-art LLMs (open and proprietary), providing a cross-domain, multi-dimensional analysis of creativity.
- Identifies systematic findings, including:
- - Strong performance gaps between proprietary and open models (especially in quality and reasoning tasks).
- - Weak cross-domain and cross-dimension correlations.

The weaknesses are:
- Heavy reliance on automatic metrics and LLM judges, which may still encode bias.
- Task selection is constrained by existing benchmarks, limiting novelty of tasks themselves.
- Some dimensions (especially novelty) remain ill-defined and inconsistent across tasks.

**Audience:**

Yes

**Audience Explanation:**

The paper addresses a timely and underexplored problem that evaluating LLM creativity in a principled, scalable way.

It contributes a benchmarking framework which is directly relevant to LLM evaluation research, alignment assessment and creative AI applications.

**Broader Impact Concerns:**

Risk of over-reliance on LLM judges, which may encode systemic biases or favor certain stylistic outputs.

**Claims And Evidence:**

Yes

**Claims Explanation:**

The claims are generally supported by systematic empirical evaluation and validation procedures.
Some claims are partially limited by dependence on proxy metrics for creativity.
Moreover, variability and weak correlation of novelty metrics, which weakens interpretability.

**Requested Changes:**

I recommend to add some changes to make the results more robust and convincing. The recommendations are as follows:

1. Clarify construct validity of “creativity”: Provide stronger theoretical grounding or justification for the chosen three dimensions.
2. Address inconsistency of novelty metrics: The paper itself shows weak/negative correlations; discuss implications more rigorously.
3. Expand limitations of LLM-as-a-judge: Include deeper analysis of bias, failure modes, and sensitivity to judge model choice.

---

> ### Author Response · Authors · 2026-04-06
> **Requested Changes [1] - Clarify the construct validity of “creativity”**
>
> We sincerely thank reviewer ocsC for the supportive and very constructive review of our evaluation framework. We will present the following response to the requested changes and weaknesses that the reviewer listed.
>
> ### 1. Clarify the construct validity of “creativity”: Provide stronger theoretical grounding or justification for the chosen three dimensions.
> Thanks for pointing this out! We acknowledge that we need more theoretical justification for the creativity taxonomy. We will make the following changes to Section 3 - “Three Dimensions of Creativity”:
> - **Grounding in existing literatures**: “Our taxonomy is grounded in the most recent version of the TTCT verbal test, which operationalizes creativity along three dimensions: fluency, flexibility, and originality.\footnote{While the original TTCT [1] includes a fourth dimension, elaboration, the current version of the TTCT verbal test as administered by major testing agencies [2,5] consists of three dimensions only, a simplification also reflected in recent psychology literature [3].} These map directly onto our dimensions: fluency corresponds to quality (the ability to produce coherent, well-formed responses), flexibility to diversity (the ability to vary approaches and perspectives), and originality to novelty (the ability to produce responses that deviate from the commonplace) [2]. We are also different from the widely-cited binary taxonomy of usefulness and originality [4] by separating originality into novelty and diversity. We argue this distinction is particularly important for LLM evaluation: diversity captures breadth, i.e., how much a model varies its outputs across responses or within a single generation, while novelty captures depth, i.e., how much a single output deviates from existing human-generated content or conventional solutions (as visualized in figures above). A model can score highly on one while failing on the other (e.g., producing many distinct but individually unoriginal responses, or producing one highly novel response with no variation across prompts), and collapsing them into a single originality score would obscure this. ”
> - **Visualized presentation**: We also refer to a visualized framing, [click this to view image](https://ibb.co/QvkMXkmZ), that motivates our choice of dimensions. Consider three spaces: the LLM's output distribution (green), the valid answer space (black), and the training data (blue). Each dimension captures a distinct geometric relationship among these three spaces. These three relationships are geometrically independent: a model can expand its output distribution along one axis without moving along the others. This independence is what motivates treating them as separate dimensions rather than collapsing them.
>    - *Quality* measures how much of the LLM's output distribution falls within the valid answer space, i.e., how well the model produces coherent, appropriate responses.
>    - *Diversity* measures the breadth of the LLM's output distribution within the valid answer space, i.e., how widely the model explores different valid responses across generations.
>    - *Novelty* measures how far the LLM's output distribution extends beyond the training data, i.e., how much the model generates content that deviates from what it has been exposed to.
>
> [1] TorranceE. P. (1966). Torrance tests of creative thinking: Norms technical manual (Research Edition).Princeton: Personnel Press
>
> [2] Torrance, E. Paul. n.d. “Torrance Tests of Creative Thinking Interpretive Manual.” Bensenville: Scholastic Testing Service. https://www.ststesting.com/gift/TTCT_InterpMOD.2018.pdf
>
> [3] Alabbasi, Ahmed M Abdulla et al. “What do educators need to know about the Torrance Tests of Creative Thinking: A comprehensive review.” Frontiers in psychology vol. 13 1000385. 26 Oct. 2022, doi:10.3389/fpsyg.2022.1000385
>
> [4] Runco, Mark A., and Garrett J. Jaeger. "The standard definition of creativity." Creativity research journal 24.1 (2012): 92-96.
>
> [5] “Torrance Tests of Creative Thinking (TTCT) Training - Mary Frances Early College of Education.” n.d. Accessed April 1, 2026. https://coe.uga.edu/outreach/programs/ttct/.

---

> ### Author Response · Authors · 2026-04-06
> **Requested Changes [2] Address inconsistency of novelty metrics**
>
> ### 2. Address inconsistency of novelty metrics: The paper itself shows weak/negative correlations; discuss implications more rigorously.
>
> We thank the reviewers for raising this point! We want to clarify our choice of three dimensions as well as our interpretation of LLM performances (especially the negative correlations) in novelty dimension. (we will also add this discussion to Section 3 and Appendix)
>
> - **Choice of Three Dimensions**: First, we clarify that our three dimensions are not derived from correlations among metrics; instead, they are defined by the goal each metric is designed to measure. A weak inter-metric correlation within a dimension, therefore, does not indicate a poorly constructed dimension; it indicates that different metrics operationalize the same high-level goal through different mechanisms, each with its own strengths and limitations. This is especially pronounced for novelty, which is most difficult to measure because it requires access to a reference corpus or solution set that is often incomplete, private, or domain-specific. For example, the Creativity Index is sensitive to the unique private training data each model has seen and to each model's paraphrasing ability; NeoCoder Divergence@0 is significantly affected by a model's constraint-following ability, which we will discuss more below. The weak correlation among these novelty metrics is therefore expected and should be considered as a contribution rather than a weakness, since they all have limitations and can be complementary to each other. Only when putting together can we evaluate the full landscape of novelty/diversity/quality of each LLM.
>
> - **Negative correlations between NeoCoder Divergence@0 and other metrics**. NeoCoder Divergence@0 measures the proportion of coding techniques in a model's solution absent from the human solution set [3]. A model that fails to solve a problem correctly will, by definition, avoid using standard human techniques, not because it is creative, but because its incorrect output lacks the structural properties of valid solutions. Nonsensical outputs would score perfectly on this metric. This is distinct from other LLM-judged novelty metrics, which score the quality of a particular output rather than performing a binary classification against a reference set. The NeoCoder paper itself addresses this by introducing denial prompting to isolate genuine divergence from failure; our use of the @0 baseline captures the unconditional setting where this confound is most visible. Notably, CreativeMath avoids this issue by only counting a solution as novel after it is verified correct. We will add this discussion to the paper as an illustration of why cross-task novelty correlations should be interpreted with care.
>
> - **Novelty requires external references**. More broadly, novelty is the only dimension that inherently asks "novel compared to what?", and the answer differs across tasks: compared to prior human-written corpora (L-Uniqueness, C-Short Surprises), to known human solutions (NeoCoder Divergence@0, C-Math Novelty), or to common conceptual associations (AUT Score). Quality asks "is this output well-formed and coherent?", which is a property intrinsic to the output. Diversity asks "how much does this output vary from other outputs?", which is measurable within the model's own generation distribution. This reference heterogeneity means novelty metrics measure a family of related but operationally distinct notions of deviation from a norm rather than a single latent construct. A model scoring high on one novelty metric may score low on another for interpretable reasons: AUT Score reflects the unusualness of free associations in a single pass, while C-Math Novelty demands solutions that are both correct and absent from the human solution set, which are two very different demands. Therefore, as mentioned earlier, having weak/negative correlation among performances in novelty metrics is not a weakness but a strength of our proposed framework. Since all the metrics have their limitations, our framework unifies them together; our result also further validates that none of them on its own is well-representative for novelty. Hence, we recommend testing them all using our benchmark to reveal the full landscape of creativity of the target LLM being evaluated.
>
> [3] Yining Lu, Dixuan Wang, Tianjian Li, Dongwei Jiang, Sanjeev Khudanpur, Meng Jiang, and Daniel Khashabi. 2025. Benchmarking Language Model Creativity: A Case Study on Code Generation. In Proceedings of the 2025 Conference of the Nations of the Americas Chapter of the Association for Computational Linguistics: Human Language Technologies (Volume 1: Long Papers), pages 2776–2794, Albuquerque, New Mexico. Association for Computational Linguistics.

---

> ### Author Response · Authors · 2026-04-06
> **Requested Changes [3] Expand limitations of LLM-as-a-judge**
>
> ### 3. "Heavy reliance on automatic metrics and LLM judges, which may still encode bias. Expand limitations of LLM-as-a-judge: Include deeper analysis of bias, failure modes, and sensitivity to judge model choice"
>
> - We acknowledge that sensitivity to judge model choice is a genuine limitation. However, rigorous cross-judge evaluation would require re-tuning prompts for each judge's backbone, since prompt design is not fully transferable across models, making a direct inter-judge comparison methodologically unreliable within the scope of this revision. Instead, we ground our reliability claims in two ways: (1) human inter-annotator agreement on our validation subsets confirms that the evaluated constructs are well-defined and consistently interpretable, providing a stable human reference that our judge is calibrated against; and (2) the Alternative Annotator Test [2] validates our specific judge-prompt configuration against these human annotations within a statistically justified error margin (\alpha = .05), rather than making claims about judge-agnostic reliability. We acknowledge that our validation is inherently specific to our current judge-prompt setup. A promising direction for future work is adopting a multi-LLM judge jury system [1], where consensus across multiple independently-prompted judge backbones replaces reliance on any single model, providing robustness to individual judge biases without requiring prompt-matched comparisons. We will add both the limitations and this future direction to the camera-ready version.
> - In our initial experiments with TTCT task, we followed the original papers’ setup and observed that **LLM-Judge show two failure modes**: (1) length bias, where judges tend to assign higher scores to verbose responses, which is particularly pronounced for models that generate preambles or closing remarks (e.g., greeting messages, expressions of politeness); (2) redundancy blindness, where judges fail to penalize responses that repeat the same core idea with superficial variation. **We mitigate both by preprocessing model outputs to strip formatting, politeness markers, and verbosity before evaluation (see task specific prompt in Appendix E.7.6).** For other tasks with no explicit length requirements, we impose different ways to parse the output so that they don’t get extraordinarily long, e.g., for AUT, we only keep the results from the first ten lines; for CreativeShort, we require the generation model to insert [START] and [END] label before and after the story.
> - We will also add these changes to acknowledge the limitation brought by using LLM-Judge
>   - **Intro**: “For every evaluation metric that requires LLM-as-a-Judge, we collect human judgments from well-trained researchers or domain experts, followed by Alternative Annotator Test [2] to verify our LLM-Judge setup ~~can indeed achieve the same quality as well-trained human annotators or domain experts~~ [has a much closer alignment to high-quality human judgement than previous automatic evaluations].”
>   - **Section 4.2 LLM-Judge Reliability**: “This method adopts a leave-one-out strategy to validate the LLM as a reliable substitute[substitute with acceptable error margin]. \footnote{We follow the Alternative Annotator Test [2], a statistical procedure that tests whether an LLM judge is an acceptable substitute for human annotators within a pre-specified error margin at significance level \alpha = .05}
>   - **Limitation Section** (add): Despite our efforts to verify LLM-judge reliability, limitations remain: annotation sample sizes are relatively small due to resource constraints, judge backbone choice introduces potential bias that we do not fully characterize, and LLM judges may carry systematic biases in assessing certain types of creative content. We acknowledge these as open challenges and encourage future work to address them as the LLM-as-a-judge methodology matures.
>
> [1] Verga, Pat, Sebastian Hofstatter, Sophia Althammer, Yixuan Su, Aleksandra Piktus, Arkady Arkhangorodsky, Minjie Xu, Naomi White, and Patrick Lewis. 2024. “Replacing Judges with Juries: Evaluating LLM Generations with a Panel of Diverse Models.” arXiv [Cs.CL]. arXiv. https://doi.org/10.48550/arXiv.2404.18796.
>
> [2] Nitay Calderon, Roi Reichart, and Rotem Dror. 2025. The Alternative Annotator Test for LLM-as-a-Judge: How to Statistically Justify Replacing Human Annotators with LLMs. In Proceedings of the 63rd Annual Meeting of the Association for Computational Linguistics (Volume 1: Long Papers), pages 16051–16081, Vienna, Austria. Association for Computational Linguistics.

---

> ### Author Response · Authors · 2026-04-06
> **Limitation - Task selection is constrained by existing benchmarks, limiting the novelty of tasks themselves**
>
> ### Task selection is constrained by existing benchmarks, limiting the novelty of tasks themselves
>
> Thanks for raising this concern. We believe our work, although it seems to be a “simple combination” of existing benchmarks, has the following novel and valuable contribution:
>
> - LLM-Judge verification: although an increasing number of benchmarks nowadays adopt the LLM-as-a-Judge paradigm, very few of them rigorously verify the alignment between their LLM-Judge and human judgment and are transparent about their verification process (e.g., releasing their human annotations data), especially in creativity evaluation, where expensive, high-quality human annotation drives more researchers to LLM-Judge evaluation. As mentioned in the paper, 5 out of 8 tasks we collected require LLM-Judge, but 3 out of 5 (TTCT, NeoCoder, CreativeMath) do not even have an annotated judgment released, and one of the remaining two (TTCW, AUT) has an LLM-Judge that does not align well enough with human judgment. Aside from human-annotated data availability, as mentioned in Request Changes [3], our initial experiment with the TTCT task shows LLM-Judge failing in at least 2 cases (long / duplicated output text), to which we designed mitigating methods. In summary, our effort in standardizing annotation protocols and LLM-Judge alignment verification can serve as a reference for future research in machine creativity and should be considered as an important contribution.
>
> - We have a concrete plan to release, maintain, and integrate more tasks in LLM (and MMLM) creativity evaluation by building a protocol for new task integration. Our open-sourced data cleaning and task integration pipeline would foster community contributions to creativity evaluation. We also have a website under construction to visualize the results, inform the community the latest changes (new task, new models, etc.), and enable easy task submission.
>
> - Existing work only focuses on a single domain and studies a limited number of models. Our expansion to cross-domain and cross-dimension benchmarking enables comparison of the SoTA LLM across different task domains, which, as mentioned in the analysis section, shows low correlation among tasks from different domains. This indicates that good performance in one domain does not transfer to others, further motivating such cross-domain analysis.

---

### Review · Reviewer_UBBj · 2026-03-24

**Summary Of Contributions:**

This paper introduces CreativityPrism, a benchmark intended to provide a broader and more scalable evaluation of LLM creativity across multiple tasks and domains. The framework combines eight tasks spanning divergent thinking, creative writing, and logical reasoning, and organizes seventeen task-specific metrics into three higher-level dimensions: quality, novelty, and diversity. The authors also benchmark 17 proprietary and open-source LLMs, and analyze differences across model families, domains, and creativity dimensions.

The paper’s main strengths are its ambition, breadth, and practical orientation. Compared with many prior works that focus on a single task or narrow notion of creativity, this benchmark attempts to consolidate several existing evaluation settings into a unified framework. I also appreciate the authors’ effort to validate LLM-based judges against human annotations rather than relying on automatic evaluation without verification. The empirical benchmarking across many models is useful, and the analysis suggesting uneven performance across domains is interesting.

At the same time, several aspects are less convincing. The proposed taxonomy of quality, novelty, and diversity is intuitive, but in my view not fully justified theoretically, and some of the dimensions group together rather heterogeneous constructs. The claim of a “holistic” evaluation framework also feels somewhat overstated, since important creativity domains such as scientific discovery, design/invention, humor, or interactive creativity are not represented. In addition, some of the higher-level conclusions, especially those based on correlation analyses across models, would benefit from stronger statistical support and more cautious interpretation.

**Audience:**

Yes

**Audience Explanation:**

Yes. I believe this paper would be of interest to at least part of the TMLR audience, especially researchers working on LLM evaluation, benchmark design, automatic assessment of open-ended generation, and computational creativity. The question of how to evaluate creativity in language models is timely and important, and the paper’s attempt to consolidate multiple tasks into a broader evaluation framework is relevant even if some of the framing and conclusions would benefit from refinement.

**Broader Impact Concerns:**

I do not see major direct broader-impact concerns beyond those already mentioned in the manuscript and common to evaluation benchmarks.

**Claims And Evidence:**

Yes

**Claims Explanation:**

Partly yes. The paper provides a substantial empirical benchmark across multiple tasks, domains, and models, and I appreciate the authors’ effort to validate several automatic judges against human annotations. The descriptive results are generally clear and support the main empirical observation that model performance varies across tasks and that different domains of creativity are not perfectly aligned.

That said, I found some of the higher-level claims stronger than the evidence currently supports. In particular, the proposed taxonomy of quality, novelty, and diversity is only partially justified, and the correlation-based arguments about transfer across domains/dimensions would benefit from stronger statistical support and more cautious interpretation. So while the evidence is broadly suggestive and often convincing at the benchmark level, some conceptual and inferential claims should be toned down or better supported.

**Requested Changes:**

The benchmark is ambitious, timely, and likely to be useful to the community, but I think several revisions would be important to strengthen the conceptual framing, support some of the empirical claims more rigorously, and improve reproducibility.


Requested changes

1. Clarify and better justify the proposed creativity taxonomy

The proposed dimensions of quality, novelty, and diversity provide a useful organizational scaffold, but their theoretical justification remains underdeveloped. The literature more commonly centers on constructs such as originality and usefulness/appropriateness, with fluency, flexibility, and elaboration also being standard in divergent thinking. As currently written, the taxonomy feels more like a pragmatic benchmark organization than a strongly grounded theory of creativity.

Such statements should better be justified:
‘’A common belief among these work is the balanced view of quality (e.g., elaboration, usefulness), novelty (e.g., originality, synthesis), and diversity (e.g., flexibility) when it comes to different dimensions of creativity.’’
‘’This taxonomy is inspired by previous work in psychological creativity evaluation (Alabbasi et al., 2022; Runco & Jaeger, 2012)’’
The paper would benefit from framing quality, novelty, and diversity more explicitly as a practical organizational taxonomy for this benchmark, rather than implying that they constitute a generally established or theoretically settled decomposition of creativity.

2. Acknowledge more clearly that the three dimensions aggregate heterogeneous constructs

At present, each dimension groups together rather different kinds of measures:
- quality mixes correctness, elaboration, fluency, and narrative effectiveness;
- novelty mixes uncommonness, different solution strategies, corpus rarity, and local sentence surprisal;
- diversity mixes semantic spread, flexibility, and lexical diversity.
This does not invalidate the benchmark, but it does mean that the paper should be more cautious about presenting these as fully unified dimensions. I encourage the authors to acknowledge this heterogeneity explicitly, from the beginning.
For instance:
The definition of diversity as evaluating “how much the LLM-generated content differs in different passes” does not seem to fit all included tasks. In DAT, for example, the 10 nouns are produced in a single pass, and the metric reflects semantic spread within one response rather than variation across multiple generations. This further suggests that the diversity dimension groups together different underlying notions (e.g., flexibility, semantic dispersion, lexical variation, cross-sample variability), and the paper should define this dimension more carefully, as the present sentence is misleading.

3. Tone down or better qualify the “holistic/comprehensive” framing

The benchmark is broader than many prior works, but the paper’s framing sometimes overstates the coverage. The current benchmark covers only a subset of creativity domains, with strong emphasis on divergent thinking, narrative generation, and constrained reasoning. Important domains such as scientific hypothesis generation/discovery, inventive design, and other forms of open-ended creative problem solving are not represented.

I suggest softening claims such as “holistic” or “comprehensive” unless they are more clearly bounded, and adding a limitation noting that conclusions may not generalize to all important forms of LLM creativity, while specifying what are the main lacking/unrepresented domains.

4. Strengthen the statistical support for the correlation-based claims

The correlation analysis is interesting, but it is currently largely descriptive. Since the conclusions rely on Pearson correlations computed across a relatively small number of models, I would encourage the authors to report statistical uncertainty for these correlations, for example:
- confidence intervals,
- significance tests with correction for multiple comparisons,
- robustness checks using rank-based measures such as Spearman’s rho or Kendall’s tau.

This is especially important given the number of pairwise comparisons and the potential non-independence among models from the same family.

More specifically, to support claims such as “models performing well in one domain or in one dimension of creativity do not necessarily perform similarly well in another”, I would encourage the authors to go beyond descriptive heatmaps and add at least one more formal analysis, for example:
- comparing within-domain versus between-domain correlations using bootstrap or permutation testing, and/or
- reporting uncertainty on correlations among aggregated domain-level scores.

5. Support the “domain-specific differences” claims with explicit statistical tests

Claims such as “logical reasoning and creative writing see a notably larger gap than divergent thinking” should be accompanied by clear quantitative support in the main results section, not only descriptive comparisons. If the authors wish to make such claims, I recommend adding appropriate tests and reporting the relevant effect sizes / uncertainty.

6. Clarify the generation protocol and sampling strategy across tasks

The paper reports dataset sizes and, for most tasks, implies one generation per item per model. However, the number of stochastic samples per input is not always stated in a single unified way across tasks. Since creativity evaluations can be sensitive to sampling variance, it would help to explicitly summarize, for each task:
- the number of generations per prompt/question,
- whether results are based on a single sample or multiple samples per item,

Since creativity-related metrics can be sensitive to sampling variance, I think repeated generations should be included rather than relying on a single sample per prompt/question. In a benchmark of this kind, where data collection is relatively accessible and automated, evaluating each task multiple times per LLM would provide a more reliable estimate of model performance and make the reported comparisons more robust.

Relatedly, I think the paper would be stronger if it discussed the role of temperature more explicitly. At minimum, it should clearly state how temperature was set for each task/model; ideally, the authors could also comment on sensitivity to temperature, since this can substantially affect creativity-related outputs.
See these papers:
Peeperkorn, M., Kouwenhoven, T., Brown, D., & Jordanous, A. (2024). Is temperature the creativity parameter of large language models?. arXiv preprint arXiv:2405.00492.
Bellemare-Pepin, A., Lespinasse, F., Thölke, P., Harel, Y., Mathewson, K., Olson, J. A., ... & Jerbi, K. (2026). Divergent creativity in humans and large language models. Scientific Reports, 16(1), 1279.

7. Clarify the human annotation / expertise claims for LLM-judge validation

The effort to validate LLM judges against human annotations is appreciated and is a strength of the paper. However, in several places annotators are described as “researchers” rather than domain experts, and it is not always clear what qualifies them to judge the creativity of the outputs. I recommend clarifying:
- the expertise/background of annotators,
- how they were trained,
- why they are appropriate judges for each task.

This would make the LLM-judge validation section more convincing.

8. Expand the limitations around LLM-based evaluation

I would strongly encourage the authors to add a more explicit limitation about continued reliance on LLM judges. Despite the careful validation effort, the framework still relies substantially on judge models. The reported inter-annotator agreement is often in the moderate range (around Fleiss’ kappa ≈ 0.4), some constructs remain inherently subjective, annotation sample sizes are relatively small for some tasks, and the choice of judge backbone varies across tasks. In addition, LLM judges may encode stylistic or benchmark-specific biases.
I think the paper should frame the approach as reducing rather than fully resolving the challenges of scalable creativity evaluation.

9. Make the reproducibility / release plan more concrete

The paper emphasizes scalability and positions CreativityPrism as a community-usable benchmark, but the reproducibility and release plan could be stated more concretely. At present, the manuscript briefly notes that the code and dataset will be published upon publication. For a benchmark paper, it would be helpful to clarify exactly what will be released and how the authors intend to support community adoption, for example:
- full evaluation pipeline,
- task wrappers,
- prompts,
- preprocessing scripts,
- Judge-LLM configurations,
- annotation resources,
- documentation for adding new models or tasks.

A clearer reproducibility statement would strengthen the paper and make its scalability claims more convincing.

10. Add a limitation regarding possible contamination / access to prior content

The paper notes that later model cutoff dates may make some models more competitive on novelty-like metrics. I think this should be acknowledged more explicitly as a limitation: some models may have had differential access to relevant prior content, which may advantage them on metrics that reward difference from earlier corpora or solution sets.

Additional comments / improvements

11. Clarify the DAT instruction and ensure exact task fidelity
The DAT prompt wording shown in the paper appears not to match the standard task instruction exactly. Since task phrasing can affect results, I recommend verifying and clearly stating whether the original DAT instruction was used verbatim or adapted.

12. Improve figure/table presentation of the main benchmark results
I found Table 3 somewhat dense. A more visual summary figure of the main benchmark results might improve readability and make cross-model/domain comparisons easier to interpret.

13. Clarify figure captions and analysis details
For correlation figures, please specify clearly in the figure or caption which correlation measure is used (e.g., Pearson). This is a small point, but it would improve clarity.

14. Validation of output length
I was not fully convinced that length effects are controlled for in the creative writing evaluations. Although some tasks impose explicit length constraints (e.g., TTCW minimum length, short-story maximum length), several metrics used in the benchmark may still be sensitive to verbosity or response length, including elaboration, fluency, lexical diversity, and corpus-overlap-based measures. It would strengthen the paper to report basic length statistics by model/task and discuss or test whether key results are robust to response length.

15. Quality evaluation of creative writing tasks
The statement that “without quality, random words and sentences would be very creative in terms of novelty and diversity, but do not convey any meaning” is intuitive, but it also highlights an important conceptual issue: the notion of quality is not operationalized in a fully uniform way across tasks. In particular, for some creative writing tasks, quality does not appear to be assessed for each individual generation as directly as it is in settings such as code execution or math correctness. I would encourage the authors to clarify how quality is defined and measured across tasks, and to acknowledge where this dimension is more indirect or task-specific than the current wording suggests. In the present state, the cited sentence is misleading.

16. Minor wording / style issues (minor)
There are a few places where the wording could be smoothed. For example, the sentence:
“...designed to capture the complex nature of LLM creativity by tasks in three distinct domains...”
reads awkwardly. A formulation such as “through tasks spanning three domains” or “across three domains” would be clearer.
Section 3:
‘’we evaluation of LLMs’’ seems to be missing a word.
‘’different dimensions of cognitive capability’’
I am not sure that “dimensions of cognitive capability” is the right wording here, since the benchmark evaluates model outputs rather than cognition per se.

---

> ### Author Response · Authors · 2026-04-06
> **Requested Changes [1-2]**
>
> We sincerely thank reviewer UBBj for the well-rounded and very constructive review of our evaluation framework. We will present the following response to every requested change in the order that the reviewer listed.
>
> ### 1. Clarify and better justify the proposed creativity taxonomy
> Thanks for pointing this out! We acknowledge that we need more theoretical justification for the creativity taxonomy. We will make the following changes to Section 3 - “Three Dimensions of Creativity”:
> - **Grounding in existing literatures**: “Our taxonomy is grounded in the most recent version of the TTCT verbal test, which operationalizes creativity along three dimensions: fluency, flexibility, and originality.\footnote{While the original TTCT [1] includes a fourth dimension, elaboration, the current version of the TTCT verbal test as administered by major testing agencies [2,5] consists of three dimensions only, a simplification also reflected in recent psychology literature [3].} These map directly onto our dimensions: fluency corresponds to quality (the ability to produce coherent, well-formed responses), flexibility to diversity (the ability to vary approaches and perspectives), and originality to novelty (the ability to produce responses that deviate from the commonplace) [2]. We are also different from the widely-cited binary taxonomy of usefulness and originality [4] by separating originality into novelty and diversity. We argue this distinction is particularly important for LLM evaluation: diversity captures breadth, i.e., how much a model varies its outputs across responses or within a single generation, while novelty captures depth, i.e., how much a single output deviates from existing human-generated content or conventional solutions (as visualized in figures above). A model can score highly on one while failing on the other (e.g., producing many distinct but individually unoriginal responses, or producing one highly novel response with no variation across prompts), and collapsing them into a single originality score would obscure this. ”
> - **Visualized presentation**: We also refer to a visualized framing, [click this to view image](https://ibb.co/QvkMXkmZ), that motivates our choice of dimensions. Consider three spaces: the LLM's output distribution (green), the valid answer space (black), and the training data (blue). Each dimension captures a distinct geometric relationship among these three spaces. These three relationships are geometrically independent: a model can expand its output distribution along one axis without moving along the others. This independence is what motivates treating them as separate dimensions rather than collapsing them.
>    - *Quality* measures how much of the LLM's output distribution falls within the valid answer space, i.e., how well the model produces coherent, appropriate responses.
>    - *Diversity* measures the breadth of the LLM's output distribution within the valid answer space, i.e., how widely the model explores different valid responses across generations.
>    - *Novelty* measures how far the LLM's output distribution extends beyond the training data, i.e., how much the model generates content that deviates from what it has been exposed to.
>
> [1] TorranceE. P. (1966). Torrance tests of creative thinking: Norms technical manual (Research Edition).Princeton: Personnel Press
>
> [2] Torrance, E. Paul. n.d. “Torrance Tests of Creative Thinking Interpretive Manual.” Bensenville: Scholastic Testing Service. https://www.ststesting.com/gift/TTCT_InterpMOD.2018.pdf
>
> [3] Alabbasi, Ahmed M Abdulla et al. “What do educators need to know about the Torrance Tests of Creative Thinking: A comprehensive review.” Frontiers in psychology vol. 13 1000385. 26 Oct. 2022, doi:10.3389/fpsyg.2022.1000385
>
> [4] Runco, Mark A., and Garrett J. Jaeger. "The standard definition of creativity." Creativity research journal 24.1 (2012): 92-96.
>
> [5] “Torrance Tests of Creative Thinking (TTCT) Training - Mary Frances Early College of Education.” n.d. Accessed April 1, 2026. https://coe.uga.edu/outreach/programs/ttct/.
>
>
> ### 2. Three dimensions aggregate heterogeneous categories.
> Thanks for pointing this out! We acknowledge that, as the review has mentioned, each of our dimensions includes metrics that measure different behaviors of LLMs. We believe this is because, even under the same taxonomy, different domains and tasks still require different evaluation metrics for creative content; for example, in math problem solving, quality means the correctness of the problem solution, while in creative writing, quality means the fluency and coherence of the storyline, or correctly conveying the central message. We believe that having heterogeneous metrics is complementary to each other, making our evaluation framework cover a wider range of perspectives on creativity.
> [We will also add the above text to Section 3.]

---

> ### Author Response · Authors · 2026-04-06
> **Request Changes [3-5]**
>
> ### 3. “holistic/comprehensive” framing:
> Thanks for pointing this out! We acknowledge that our task selection does not cover some highly creative domains, such as scientific discovery and inventive design, due to lack of automatic evaluation in those domains. We will reframe our paper by changing all the “holistic” and “unified” to “cross-domain” or “multi-domain” (whichever is more appropriate to the context). [We will also add this to the limitation section.]
>
>
> ### 4. Statistical support for correlation-based claims
> (comparing the within-domain and cross-domain performance correlation):
> Thanks for bringing this up! Follow the suggestion, we did the following experiment, and we will add these to the appendix and refer to them in the corresponding sections in the main text.
>
> To show that within-domain correlation is significantly larger than cross-domain ones, we computed Pearson correlations for all 136 pairwise combinations of the 17 evaluation metrics ($N=17$ models), classifying each pair as within-domain (both metrics from the same domain; $n=44$ pairs) or between-domain (metrics from different domains; $n=92$ pairs). A permutation test was used to assess whether within-domain pairs show systematically higher correlations: all 136 $r$-values were pooled and randomly reassigned to within/between groups across 10,000 shuffles, with the $p$-value defined as the proportion of shuffles yielding an absolute mean difference at least as large as observed. Bootstrap 95% confidence intervals on the mean difference
> were computed from 10,000 resamples (seed=42). The test result shown in the table below can serve as a statistical support for our claim in Section 5.2: “We find a strong correlation in the models’ performance on metrics coming from the same task.”
>
> | Group | $n$ pairs | Mean $r$ | Median $r$ |
> |---|:---:|:---:|:---:|
> | Within-domain  | 44 | .494 | .486 |
> | Between-domain | 92 | .341 | .350 |
> | **Difference** |    | **+.153** | |
> | 95% CI (bootstrap) | | [.057, .248] | |
> | Permutation $p$ (two-tailed) | | <.001 | |
>
> *Based on 10,000 permutation shuffles and 10,000 bootstrap resamples (seed=42).*
>
>
>
> ### 5. Statistical support on “domain-specific differences” claims
> (i.e., the gap between close and open models on different domains):
> Thanks for bringing this up. Follow the suggestion, we did the following statistical test, and we will add these to the appendix and refer to them in the corresponding sections in the main text.
>
> To formally test whether the performance gap between open-source and proprietary models varies across creativity domains, we conducted Welch's independent-samples $t$-tests (because of differences in size and variance) on per-domain scores (open-source: $n=10$; proprietary: $n=7$), with simple average; we also applied Bonferroni correction across $k=3$ domain comparisons. Effect sizes are reported as Cohen's $d$ using the pooled standard deviation. Bootstrap 95% confidence intervals on the mean gap were computed from 10,000 resamples (seed=42). Results have shown a significant difference between the mean performance score of open versus proprietary models in all three domains, with creative writing and logical reasoning having a notably larger gap in terms of Cohen’s $d$. The test result in the table below support our claim in Section 5.1: “Among the three domains, logical reasoning and creative writing see a notably larger gap than divergent thinking.”
>
> | Domain | Open ($n$=10) | Prop. ($n$=7) | Gap | 95% CI (bootstrap) | $t$ | $p$ (Bonf.) | Cohen's $d$ |
> |---|:---:|:---:|:---:|:---:|:---:|:---:|:---:|
> | Creative Writing   | .320 | .500 | +.179 | [.123, .236] | 5.747 | <.001 \*\*\* | 2.772 |
> | Divergent Thinking | .664 | .731 | +.067 | [.036, .097] | 3.956 | .004 \*\*    | 1.930 |
> | Logical Reasoning  | .406 | .648 | +.242 | [.174, .310] | 6.493 | <.001 \*\*\* | 3.023 |
>
> *\*\* p < .01, \*\*\* p < .001 after Bonferroni correction (k=3).*

---

> ### Author Response · Authors · 2026-04-06
> **Requested Changes [6-7]**
>
> ### 6. Specify the generation protocol and sampling strategy.
> - First, we will acknowledge in the main text that we only ran all experiments once and will add 4 more runs for each task before the camera-ready version is due; we will also add that temperature is set to 0.7 for all tasks in the main text.
> - **Regarding multiple runs with the same setting**: We agree with the reviewer that it would be better to have multiple runs with the same temperature and to compare performances across temperatures to ensure benchmark robustness. We present the following partial results, from AUT and TTCW tasks, with all models except Claude-3.7-Sonnet (no API access anymore), DeepSeek Models (same reason), and will run full experiments, i.e., 5 total runs for each model. The performance visualizations are in the link below, with error bars showing the min, max, and average of task performances. Based on these partial results, we believe the model performances are stable across 5 runs. We also want to point out that there are many other factors, e.g., prompt style, max length, etc., that could also impact the results. We will leave the analysis of those factors to future work and focus on a baseline study now.
>   - AUT Performance: https://ibb.co/67Wmw0Bs
>   - TTCW Performance: https://ibb.co/cXZn1fdN
> - Regarding the effect of temperature on results: Prior work has shown mixed conclusions when it comes to temperature versus creativity [9, 10, 11]. One of our early experiments that involves CreativeShortStory and CreativityIndex, with Qwen72B and Olmo13B, shows that temperature has little influence on the performance of CreativeShortStory tasks, while models perform slightly better in CreativityIndex with higher temperature settings. With such mixed results, we decided to fix the temperature for all experiments and focus on building an evaluation framework at that time. To make the analysis more rounded, we will run a further analysis of the effect of temperature on creativity performance before the camera-ready deadline.
>   - Temperature varying experiment result: https://ibb.co/WWm43K9j  (Q = Qwen2.5-72B, O = Olmo2-13B)
>
> ### 7. Clarify the human annotator's background, qualifications, and training strategy
> We will make the following changes:
> - **LLM-as-a-Judge Reliability Section (Section 4.2)**: “Annotators are all researchers familiar with the task.” => “In total, 10 annotators participate in the LLM-Judge verification, all are Ph.D. students or faculty in the field of computer science in U.S. institutes (\footnote: TTCW has its own expert annotation, so we do not collect annotation for it. Details in Appendix E.). Annotator training details can be found in Appendix E.”
> - **Appendix E**: Training steps: “Each annotator starts by going through no more than 3 example datapoints with the leading author available for any clarification questions either in-person or virtually. After confirming the annotation is done correctly, the author will leave the annotator to work on the annotations”.
>
> [9] Peeperkorn, Max, Tom Kouwenhoven, Dan Brown, and Anna Jordanous. 2024. “Is Temperature the Creativity Parameter of Large Language Models?” In ICCC, 226–35.
>
> [10] Lu, Li-Chun, Shou-Jen Chen, Tsung-Min Pai, Chan-Hung Yu, Hung-Yi Lee, and Shao-Hua Sun. 2024. “LLM Discussion: Enhancing the Creativity of Large Language Models via Discussion Framework and Role-Play.” In First Conference on Language Modeling. https://openreview.net/forum?id=ybaK4asBT2
>
> [11] Zhao, Yunpu, Rui Zhang, Wenyi Li, and Ling Li. 2025. “Assessing and Understanding Creativity in Large Language Models.” Machine Intelligence Research, April, 1–20.

---

> ### Author Response · Authors · 2026-04-06
> **Requested Changes [8-13]**
>
> ### 8. Framing of LLM-based evaluation
> We acknowledge the limitations in our LLM-based evaluation settings and will make the following changes:
>
> - **Intro**: “For every evaluation metric that requires LLM-as-a-Judge, we collect human judgments from well-trained researchers or domain experts, followed by Alternative Annotator Test (Calderon et al., 2025) to verify our LLM-Judge setup can ~~indeed achieve the same quality as well-trained human annotators or domain experts~~ [has a much closer alignment to high-quality human judgement than previous automatic evaluations].”
> - **Section 4.2 LLM-Judge Reliability**: “This method adopts a leave-one-out strategy to validate the LLM as a ~~reliable substitute~~ [substitute with acceptable error margin]. \footnote{We follow the Alternative Annotator Test [5], a statistical procedure that tests whether an LLM judge is an acceptable substitute for human annotators within a pre-specified error margin at significance level \alpha = .05}
> - **Limitation Section** (add): Despite our efforts to verify LLM-judge reliability, limitations remain: annotation sample sizes are relatively small due to resource constraints, judge backbone choice introduces potential bias that we do not fully characterize, and LLM judges may carry systematic biases in assessing certain types of creative content. We acknowledge these as open challenges and encourage future work to address them as the LLM-as-a-judge methodology matures.
>
>
> ### 9. Adding Contamination issue to the Limitation section:
>
> We will add the following sentence to the limitation section: Given the inevitable data contamination problem, models with a later release date may have advantages on metrics that favor different outputs compared to existing content (e.g., CreativityIndex, CreativeMath, etc). This is an inherent limitation that needs to be aware of.
>
>
> ### 10. Reproducibility plan:
> We will add the following section in the Appendix about the benchmark release plan:
>
> We plan to release the full evaluation pipeline, task wrappers, prompts, preprocessing scripts, LLM-Judge configurations, and all the human annotations upon publication. We will also establish a protocol for adding new tasks, including but not limited to: task-related data structure, inference and evaluation code interface (e.g., what kinds of functions they need to have), sample inference and evaluation scripts, and developer contact for people to submit their tasks. We also have a leaderboard website under construction to publicly show benchmark results. We believe that all of these combined will promote community contribution to a truly comprehensive and up-to-date creativity assessment.
>
> ### 11. DAT prompt different from the original one:
> Thanks for pointing this out! We will add the following in Appendix E.6.2 (DAT Inference Prompt):
>
> We follow [1] on prompt design, which aligns with the prompt that is used in the study of human creativity [2]. We acknowledge the slight differences in prompts across research in both computer science and psychology. We suggest that researchers carefully examine our prompt choice before using our results.
>
> ### 12. Figure table representation (table 3 too dense):
> Thanks for pointing it out! Figures 9- 15 are bar charts that visualize the results in Table 3. We will add a reference to them in the main text!
>
> ### 13. Clarify figure captions and analysis details:
> We will add the clarification in the caption that we use Pearson’s correlation for all analyses.
>
>
> [1] Honghua Chen and Nai Ding. 2023. Probing the “Creativity” of Large Language Models: Can models produce divergent semantic associations?. In Findings of the Association for Computational Linguistics: EMNLP 2023, pages 12881–12888, Singapore. Association for Computational Linguistics.
>
> [2] J.A. Olson, J. Nahas, D. Chmoulevitch, S.J. Cropper, & M.E. Webb, Naming unrelated words predicts creativity, Proc. Natl. Acad. Sci. U.S.A. 118 (25) e2022340118, https://doi.org/10.1073/pnas.2022340118 (2021).
>
> [5] Nitay Calderon, Roi Reichart, and Rotem Dror. 2025. The Alternative Annotator Test for LLM-as-a-Judge: How to Statistically Justify Replacing Human Annotators with LLMs. In Proceedings of the 63rd Annual Meeting of the Association for Computational Linguistics (Volume 1: Long Papers), pages 16051–16081, Vienna, Austria. Association for Computational Linguistics.

---

> ### Author Response · Authors · 2026-04-06
> **Requested Changes [14-16]**
>
> ### 14. Validation of output length
> - Thanks for pointing this out! Here are the model output distribution length (simple split by space) for each task. For all tasks and all LLMs, the output max token is set to 4096 (except for CreativityIndex, which is set to 288). We believe this distribution confirms that 1) no model has a significant advantage compared to other tasks. 2) For tasks with length constraints (CreativityIndex, CreativeShortStory, TTCW), the length requirements are met correctly.
>   - Length Distribution (TTCW): https://ibb.co/M5RFW7vk
>   - Length Distribution (Other Tasks): https://ibb.co/Qjxmjv6f
>   - Length Distribution by models: https://ibb.co/dsTb8SMB
>
> - In our initial experiments with TTCT task, we followed the original papers’ setup and observed that LLM-Judge show two failure modes: (1) length bias, where judges tend to assign higher scores to verbose responses, which is particularly pronounced for models that generate preambles or closing remarks (e.g., greeting messages, expressions of politeness); (2) redundancy blindness, where judges fail to penalize responses that repeat the same core idea with superficial variation. **We mitigate both by preprocessing model outputs to strip formatting, politeness markers, and verbosity before evaluation (see task specific prompt in Appendix E.7.6)**. For other tasks with no explicit length requirements, we impose different ways to parse the output so that they don’t get extraordinarily long, e.g., for AUT, we only keep the results from the first ten lines; for CreativeShort, we require the generation model to insert [START] and [END] label before and after the story.
>
> ### 15. Quality evaluation of creative writing tasks:
> addressed in point 2 (Three dimensions aggregate heterogeneous categories.)
>
> ### 16. Other edits
> - “...designed to ~~capture~~ [uncover] the ~~complex nature~~ [complexity] of LLM creativity by tasks in three distinct domains.”
> - Formulations such as “through tasks spanning three domains” and “across three domains” will all change to “across three domains”.
> - Section 3: “~~we~~ [our] evaluation of LLMs”
> - “With this categorization, we evaluation of LLMs is not limited into one dimension, ~~but in terms of different dimensions of cognitive capability,~~ even if the tasks come from different domains.”
> - We will also add a note saying that all correlations in Figure 4 are Pearson Correlation (we have that in Figure 3, but not in Figure 4).

---

### Decision · Action_Editor_JZne · 2026-05-31

**Recommendation:** Accept as is

**Additional Comments:**

For the final version, I encourage the authors to incorporate the promised revisions carefully. In particular, the limitations section should explicitly state that most results are based on limited repeated-run evidence, that stochastic sampling robustness has not yet been comprehensively established, and that human annotators were generally task-familiar CS researchers rather than creativity-domain experts. The authors should also verify instruction adherence for constrained tasks, including minimum or maximum length requirements, and clearly describe how noncompliant outputs are handled. The discussion of LLM-as-a-judge evaluation should emphasize that the validation supports the specific judge-prompt configurations used in the paper, not judge-agnostic reliability. The paper should avoid overstating the framework as fully “holistic” or “comprehensive,” since it does not cover many forms of creativity such as scientific discovery, invention/design, humor, multimodal generation, or interactive creative collaboration. Finally, the authors should provide practical guidance on how users should interpret aggregate scores versus domain- or dimension-specific scores, since different applications may require different notions of creativity.

**Audience:**

Yes

**Audience Explanation:**

The paper is likely to be of interest to at least some members of the TMLR audience. Evaluation of LLM creativity is a timely and underexplored problem, and the paper offers a useful consolidation of tasks, metrics, model comparisons, and judge-validation procedures. Researchers working on LLM evaluation, open-ended generation, benchmark design, computational creativity, and automatic assessment of subjective outputs are likely to find the framework and empirical results informative.

**Claims And Evidence:**

Yes

**Claims Explanation:**

Summary: The paper introduces CreativityPrism, a multi-domain evaluation framework for assessing creativity in large language models. The framework combines eight tasks across divergent thinking, creative writing, and logical reasoning, and organizes the evaluation around three dimensions: quality, novelty, and diversity. The paper evaluates 17 proprietary and open-source LLMs and studies performance variation across tasks, domains, and creativity dimensions. A central empirical finding is that performance on one creativity domain or dimension does not reliably transfer to others, motivating a broader evaluation framework rather than relying on a single task or metric.

Overall, I find that the main claims of the paper are supported by sufficiently accurate and clear evidence. The reviewers agreed that the paper provides a systematic empirical evaluation, covers a broad set of models and tasks, and makes a useful effort to validate LLM-as-a-judge components against human annotations. The reviews also raised important concerns about the theoretical grounding of the proposed taxonomy, the heterogeneity of the quality/novelty/diversity dimensions, reliance on LLM judges, aggregation choices, stochastic sampling robustness, instruction adherence, and the limited scope of text-only creativity tasks. The authors responded constructively to these concerns by clarifying the taxonomy, adding statistical tests for correlation and domain-gap claims, analyzing robustness of score aggregation, expanding limitations around LLM-based evaluation and contamination, and committing to a clearer release plan.

Some limitations remain, especially incomplete repeated-run robustness across the full benchmark and the fact that annotators are task-familiar researchers rather than domain experts in creativity assessment. However, these limitations do not undermine the central benchmark-level claims when they are clearly acknowledged.